# Outlier-Robust Wasserstein DRO

**Sloan Nietert**
Cornell University
nietert@cs.cornell.edu

**Ziv Goldfeld**
Cornell University
goldfeld@cornell.edu

**Soroosh Shafiee**
Cornell University
shafiee@cornell.edu

## Abstract

Distributionally robust optimization (DRO) is an effective approach for data-driven decision-making in the presence of uncertainty. Geometric uncertainty due to sampling or localized perturbations of data points is captured by Wasserstein DRO (WDRO), which seeks to learn a model that performs uniformly well over a Wasserstein ball centered around the observed data distribution. However, WDRO fails to account for non-geometric perturbations such as adversarial outliers, which can greatly distort the Wasserstein distance measurement and impede the learned model. We address this gap by proposing a novel outlier-robust WDRO framework for decision-making under both geometric (Wasserstein) perturbations and non-geometric (total variation (TV)) contamination that allows an $\varepsilon$-fraction of data to be arbitrarily corrupted. We design an uncertainty set using a certain robust Wasserstein ball that accounts for both perturbation types and derive minimax optimal excess risk bounds for this procedure that explicitly capture the Wasserstein and TV risks. We prove a strong duality result that enables tractable convex reformulations and efficient computation of our outlier-robust WDRO problem. When the loss function depends only on low-dimensional features of the data, we eliminate certain dimension dependencies from the risk bounds that are unavoidable in the general setting. Finally, we present experiments validating our theory on standard regression and classification tasks.

## 1 Introduction

The safety and effectiveness of various operations rely on making informed, data-driven decisions in uncertain environments. Distributionally robust optimization (DRO) has emerged as a powerful framework for decision-making in the presence of uncertainties. In particular, Wasserstein DRO (WDRO) captures uncertainties of geometric nature, e.g., due to sampling or localized (adversarial) perturbations of the data points. The WDRO problem is a two-player zero-sum game between a learner (decision-maker), who chooses a decision $\theta \in \Theta$, and Nature (adversary), who chooses a distribution $\nu$ from an ambiguity set defined as the $p$-Wasserstein ball of a prescribed radius around the observed data distribution $\tilde{\mu}$. Namely, WDRO is given by[1]

$$\inf_{\theta \in \Theta} \sup_{\nu:\, \mathsf{W}_p(\nu, \tilde{\mu}) \leq \rho} \mathbb{E}_{Z \sim \nu}[\ell(\theta, Z)], \tag{1}$$

whose solution $\hat{\theta} \in \Theta$ is chosen to minimize risk over the Wasserstein ball with respect to (w.r.t.) the loss function $\ell$. WDRO has received considerable attention in many fields, including machine learning [6, 22, 45, 48, 59], estimation and filtering [36, 37, 46], and chance constraint programming [12, 55].

In many practical scenarios, the observed data may be contaminated by non-geometric perturbations, such as adversarial outliers. Unfortunately, the WDRO problem from (1) is not suited for handling this

---

[1]Here, $\mathsf{W}_p(\mu, \nu) \coloneqq \inf_{\pi \in \Pi(\mu, \nu)} \left( \int \|x - y\|^p d\pi(x, y) \right)^{1/p}$ is the $p$-Wasserstein metric between $\mu$ and $\nu$, where $\Pi(\mu, \nu)$ is the set of all their couplings.

37th Conference on Neural Information Processing Systems (NeurIPS 2023).

issue, as even a small fraction of outliers can greatly distort the $W_p$ measurement and impede decision-making. In this work, we address this gap by proposing a novel outlier-robust WDRO framework that can learn well-performing decisions even in the presence of outliers. We couple it with a comprehensive theory of excess risk bounds, statistical guarantees, and computationally-tractable reformulations, as well as supporting numerical results.

## 1.1 Contributions

We consider a scenario where the observed data distribution $\tilde{\mu}$ is subject to both geometric (Wasserstein) perturbations and non-geometric (total variation (TV)) contamination, which allows an $\varepsilon$-fraction of data to be arbitrarily corrupted. Namely, if $\mu$ is the true (unknown) data distribution, then the Wasserstein perturbation maps it to some $\mu'$ with $W_p(\mu', \mu) \leq \rho$, and the TV contamination step further produces $\tilde{\mu}$ with $\|\tilde{\mu} - \mu'\|_{\mathsf{TV}} \leq \varepsilon$ (e.g., in the special case of the Huber model [28], $\tilde{\mu} = (1-\varepsilon)\mu' + \varepsilon\alpha$ where $\alpha$ is an arbitrary noise distribution). To enable robust decision-making under this model, we replace the Wasserstein ambiguity set in (1) with a ball w.r.t. the recently proposed outlier-robust Wasserstein distance $W_p^\varepsilon$ [38, 39]. The $W_p^\varepsilon$ distance (see (2) ahead) filters out the $\varepsilon$-fraction of mass from the contaminated distribution that contributed most to the transportation cost, and then measures the $W_p$ distance

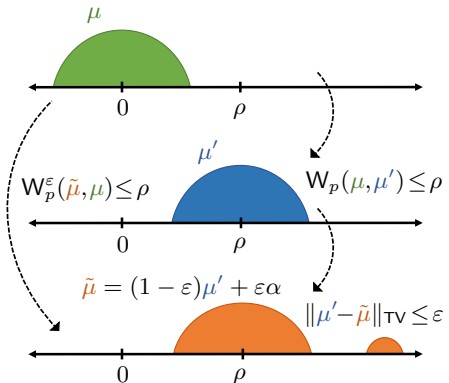

**Figure 1:** A visual depiction of a clean measure $\mu \in \mathcal{P}(\mathbb{R})$ and a corrupted observation $\tilde{\mu} \in \mathcal{P}(\mathbb{R})$ satisfying $W_p^\varepsilon(\tilde{\mu}, \mu) \leq \rho$.

post-filtering. To obtain well-performing solutions for our WDRO problem, the $W_p^\varepsilon$ ball is intersected with a set that encodes standard moment assumptions on the uncorrupted data distribution, which are necessary for meaningful outlier-robust estimation guarantees.

We establish minimax optimal excess risk bounds for the decision $\hat{\theta}$ that solves the proposed outlier-robust WDRO problem. The bounds control the gap $\mathbb{E}[\ell(\hat{\theta}, Z)] - \mathbb{E}[\ell(\theta_\star, Z)]$, where $Z \sim \mu$ follows the true data distribution and $\theta_\star = \arg\min_\theta \mathbb{E}[\ell(\theta, Z)]$ is the optimal decision, subject to regularity properties of $\ell_\star = \ell(\theta_\star, \cdot)$. In turn, our bounds imply that the learner can make effective decisions using outlier-robust WDRO based on the contaminated observation $\tilde{\mu}$, so long as $\ell_\star$ has low variational complexity. The bounds capture this complexity using the Lipschitz or Sobolev seminorms of $\ell_\star$ and clarify the distinct effect of each perturbation (Wasserstein versus TV) on the quality of the learned $\hat{\theta}$ solution. We further establish their minimax optimality when $p = 1$, by providing a matching lower bound in the setting when an adversary picks a class of Lipschitz functions over which the learner must perform uniformly well. The excess risk bounds become looser as the data dimension $d$ grows. We show that this degradation is alleviated when the loss function depends on the data only through $k$-dimensional affine features, by providing risk bounds that adapt to $k$ instead of $d$.

We then move to study the computational side of the problem, which may initially appear intractable due to non-convexity of the constraint set. We resolve this via a cheap preprocessing step that computes a coarse robust estimate of the mean [34] and replaces the original constraint set (that involves the true mean) with a version centered around the estimate. We adapt our excess risk bounds to this formulation and then prove a strong duality theorem. The dual form is reminiscent of the one for classical WDRO with adaptations reflecting the constraint to the clean distribution family and the partial transportation under $W_p^\varepsilon$. Under additional convexity conditions on the loss, we further derive an efficiently-computable, finite-dimensional, convex reformulation. The optimization results are also adapted to the setting with low-dimensional features. Using the developed machinery, we present experiments that validate our theory on simple regression/classification tasks and demonstrate the superiority of the proposed approach over classical WRDO, when the observed data is contaminated.

## 1.2 Related Work

**Distributionally robust optimization.** The Wasserstein distance has emerged as a powerful tool for modeling uncertainty in the data generating distribution. It was first used to construct an ambiguity set around the empirical distribution in [40]. Recent advancements in convex reformulations and

approximations of the WDRO problem, as discussed in [8, 20, 35], have brought notable computational advantages. Additionally, WDRO is linked to various forms of variation [2, 9, 19, 43] and Lipschitz [7, 11, 44] regularization, which contribute to its success in practice. Robust generalization guarantees can also be provided by WDRO via measure concentration argument or transportation inequalities [18, 30, 31, 51, 53, 54]. Several works have raised concerns regarding the sensitivity of standard DRO to outliers [24, 27, 58]. An attempt to address this was proposed in [56] using a refined risk function based on a family of $f$-divergences. This formulation aims to prevent DRO from overfitting to potential outliers but is not robust to geometric perturbations. Further, their risk bounds require a moment condition to hold uniformly over $\Theta$, in contrast to our bounds that depend only on $\theta_\star$. We are able to address these limitations by setting a WDRO framework based on partial transportation. While partial OT has been previously used in the context of DRO problems, it was introduced to address stochastic programs with side information in [17] rather than to account for outlier robustness. Another closely related line of work is presented in [4, 5], where the ambiguity set is constructed using an $f$-divergence to mitigate statistical errors and the Prokhorov distance to handle outlier data. The proposed model is both computationally efficient and statistically reliable. However, they have not investigated its minimax optimality or robustness against the Huber contamination model, which we aim to do in this paper. Additionally, a best-case favorable analysis approach has been proposed in [29] to address outlier data. This approach is an alternative to the worst-case distributionally robust method. However, it requires solving a non-convex optimization problem, significantly impacting its scalability, and is not accompanied by any proof of minimax optimality.

**Robust statistics.** The problem of learning from data under TV $\varepsilon$-corruptions dates back to [28]. Over the years, various robust and sample-efficient estimators, particularly for mean and scale parameters, have been developed in the robust statistics community; see [41] for a comprehensive survey. The theoretical computer science community, on the other hand, has focused on developing computationally efficient estimators that achieve optimal estimation rates in high dimensions [13, 16]. Relatedly, the probably approximate correct (PAC) learning framework has been well-studied in similar models [1, 10]. Recently, [58] developed a unified robust estimation framework based on minimum distance estimation that gives sharp population-limit and promising finite-sample guarantees for mean and covariance estimation, as well linear regression. Their analysis centers on a generalized resilience quantity, which is essential to our work. We are unaware of any results in the settings above which extend to combined TV and $\mathsf{W}_p$ corruptions. Finally, our analysis relies on the outlier-robust Wasserstein distance from [38, 39], which was shown to yield an optimal minimum distance estimate for robust distribution estimation under $\mathsf{W}_p$ loss.

## 2 Preliminaries

**Notation.** Consider a closed, non-empty set $\mathcal{Z} \subseteq \mathbb{R}^d$ equipped with the Euclidean norm $\|\cdot\|$. A continuously differentiable function $f : \mathcal{Z} \to \mathbb{R}$ is called $\alpha$-smooth if $\|\nabla f(z) - \nabla f(z')\| \leq \alpha\|z - z'\|$, for all $z, z' \in \mathcal{Z}$. The perspective function of a lower semi-continuous (l.s.c.) and convex function $f$ is $P_f(x, \lambda) := \lambda f(x/\lambda)$ for $\lambda > 0$, with $P_f(x, \lambda) = \lim_{\lambda \to 0} \lambda f(x/\lambda)$ when $\lambda = 0$. The convex conjugate of $f$ is $f^*(y) := \sup_{x \in \mathbb{R}^d} y^\top x - f(x)$. We denote by $\chi_{\mathcal{Z}}$ the indicator function of $\mathcal{Z}$, that is, $\chi_{\mathcal{Z}}(z) = 0$ if $z \in \mathcal{Z}$ and $\chi_{\mathcal{Z}}(z) = \infty$ otherwise. The convex conjugate of $\chi_{\mathcal{Z}}$, denoted by $\chi_{\mathcal{Z}}^*$, is termed as the support function of $\mathcal{Z}$.

We use $\mathcal{M}(\mathcal{Z})$ for the set of signed Radon measures on $\mathcal{Z}$ equipped with the TV norm $\|\mu\|_{\mathsf{TV}} := \frac{1}{2}|\mu|(\mathcal{Z})$, and write $\mu \leq \nu$ for set-wise inequality. The class of Borel probability measures on $\mathcal{Z}$ is denoted by $\mathcal{P}(\mathcal{Z})$. We write $\mathbb{E}_\mu[f(Z)]$ for expectation of $f(Z)$ with $Z \sim \mu$; when clear from the context, the random variable is dropped and we write $\mathbb{E}_\mu[f]$. Let $\Sigma_\mu$ denote the covariance matrix of $\mu \in \mathcal{P}_2(\mathcal{Z})$. Define $\mathcal{P}_p(\mathcal{Z}) := \{\mu \in \mathcal{P}(\mathcal{Z}) : \inf_{z_0 \in \mathcal{Z}} \mathbb{E}_\mu[\|Z - z_0\|^p] < \infty\}$. The push-forward of $f$ through $\mu \in \mathcal{P}(\mathcal{Z})$ is $f_\# \mu(\cdot) := \mu(f^{-1}(\cdot))$, and, for $\mathcal{A} \subseteq \mathcal{P}(\mathcal{Z})$, write $f_\# \mathcal{A} := \{f_\# \mu : \mu \in \mathcal{A}\}$. The $p$th order homogeneous Sobolev (semi)norm of continuously differentiable $f : \mathcal{Z} \to \mathbb{R}$ w.r.t. $\mu$ is $\|f\|_{\dot{H}^{1,p}(\mu)} := \mathbb{E}_\mu[\|\nabla f\|^p]^{1/p}$. The set of integers up to $n \in \mathbb{N}$ is denote by $[n]$; we also use the shorthand $[x]_+ = \max\{x, 0\}$. We write $\lesssim, \gtrsim, \asymp$ for inequalities/equality up to absolute constants.

**Classical and outlier-robust Wasserstein distances.** For $p \in [1, \infty)$, the $p$-*Wasserstein distance* between $\mu, \nu \in \mathcal{P}_p(\mathcal{Z})$ is $\mathsf{W}_p(\mu, \nu) := \inf_{\pi \in \Pi(\mu, \nu)} \left(\mathbb{E}_\pi[\|X - Y\|^p]\right)^{1/p}$, where $\Pi(\mu, \nu) := \{\pi \in \mathcal{P}(\mathcal{Z}^2) : \pi(\cdot \times \mathcal{Z}) = \mu, \pi(\mathcal{Z} \times \cdot) = \nu\}$ is the set of all their couplings. Some basic properties of

$W_p$ are (see, e.g., [42, 52]): (i) $W_p$ is a metric on $\mathcal{P}_p(\mathcal{Z})$; (ii) the distance is monotone in the order, i.e., $W_p \leq W_q$ for $p \leq q$; and (iii) $W_p$ metrizes weak convergence plus convergence of $p$th moments: $W_p(\mu_n, \mu) \to 0$ if and only if $\mu_n \overset{w}{\to} \mu$ and $\int \|x\|^p d\mu_n(x) \to \int \|x\|^p d\mu(x)$.

To handle corrupted data, we employ the $\varepsilon$-*outlier-robust $p$-Wasserstein distance*[2], defined by

$$W_p^\varepsilon(\mu, \nu) \coloneqq \inf_{\substack{\mu' \in \mathcal{P}(\mathbb{R}^d) \\ \|\mu' - \mu\|_{\mathsf{TV}} \leq \varepsilon}} W_p(\mu', \nu) = \inf_{\substack{\nu' \in \mathcal{P}(\mathbb{R}^d) \\ \|\nu' - \nu\|_{\mathsf{TV}} \leq \varepsilon}} W_p(\mu, \nu'). \tag{2}$$

The second equality is a useful consequence of Lemma 4 in [39] (see Appendix A for details, along with an interpretation of $W_p^\varepsilon$ as a partial OT distance).

**Robust statistics.**    Resilience is a standard sufficient condition for population-limit robust statistics bounds [49, 58]. The *$p$-Wasserstein resilience* of a measure $\mu \in \mathcal{P}(\mathcal{Z})$ is defined by

$$\tau_p(\mu, \varepsilon) \coloneqq \sup_{\mu' \leq \frac{1}{1-\varepsilon} \mu} \sup_{\mu' \leq \frac{1}{1-\varepsilon} \mu} W_p(\mu', \mu),$$

and that of a family $\mathcal{G} \subseteq \mathcal{P}(\mathbb{R})$ by $\tau_p(\mathcal{G}, \varepsilon) \coloneqq \sup_{\mu \in \mathcal{G}} \tau_p(\mu, \varepsilon)$. The relation between $W_p$ resilience and robust estimation is formalized in the following proposition.

**Proposition 1** (Robust estimation under $W_p$ resilience [39]). *Fix $0 \leq \varepsilon \leq 0.49$. For any clean distribution $\mu \in \mathcal{G} \subseteq \mathcal{P}(\mathcal{Z})$ and corrupted measure $\tilde{\mu} \in \mathcal{P}(\mathcal{Z})$ such that $W_p^\varepsilon(\tilde{\mu}, \mu) \leq \rho$, the minimum distance estimate $\hat{\mu} = \arg\min_{\nu \in \mathcal{G}} W_p^\varepsilon(\nu, \tilde{\mu})$ satisfies $W_p(\hat{\mu}, \mu) \lesssim \rho + \tau_p(\mathcal{G}, 2\varepsilon)$.*[3]

Throughout, we focus on the bounded covariance class $\mathcal{G}_{\mathrm{cov}} \coloneqq \{\mu \in \mathcal{P}(\mathcal{Z}) : \Sigma_\mu \preceq I_d\}$.

**Proposition 2** ($W_p$ resilience bound for $\mathcal{G}_{\mathrm{cov}}$ [39]). *Fixing $0 \leq \varepsilon \leq 0.99$ and $1 \leq p \leq 2$, we have $\tau_p(\mathcal{G}_{\mathrm{cov}}, \varepsilon) \lesssim \sqrt{d}\, \varepsilon^{1/p - 1/2}$.*

# 3   Outlier-robust WDRO

We perform stochastic optimization with respect to an unknown data distribution $\mu$, given access only to a corrupted version $\tilde{\mu}$. We allow both localized Wasserstein perturbations, that map $\mu$ to some $\mu'$ with $W_p(\mu, \mu') \leq \rho$, and TV $\varepsilon$-contamination that takes $\mu'$ to $\tilde{\mu}$ with $\|\tilde{\mu} - \mu'\|_{\mathsf{TV}} \leq \varepsilon$. Equivalently, both perturbations are captured by $W_p^\varepsilon(\tilde{\mu}, \mu) \leq \rho$.[4] To simplify notation, we henceforth suppress the dependence of the loss function $\ell$ on the model parameters $\theta \in \Theta$, writing $\ell$ for $\ell(\theta, \cdot)$ for a specific function and $\mathcal{L} = \{\ell(\theta, \cdot)\}_{\theta \in \Theta}$ for the whole class. Our full model is as follows.

**Setting A:** Fix a $p$-Wasserstein radius $\rho \geq 0$ and TV contamination level $\varepsilon \in [0, 0.49]$. Let $\mathcal{L} \subseteq \mathbb{R}^{\mathcal{Z}}$ be a family of real-valued loss functions on $\mathcal{Z}$, such that each $\ell \in \mathcal{L}$ is l.s.c. with $\sup_{z \in \mathcal{Z}} \frac{\ell(z)}{1 + \|z\|^p} < \infty$, and fix a class $\mathcal{G} \subseteq \mathcal{P}_p(\mathcal{Z})$ encoding distributional assumptions. We consider the following model:

   (i)  Nature selects a distribution $\mu \in \mathcal{G}$, unknown to the learner;
   (ii) The learner observes a corrupted measure $\tilde{\mu} \in \mathcal{P}(\mathcal{Z})$ such that $W_p^\varepsilon(\tilde{\mu}, \mu) \leq \rho$;
   (iii) The learner selects a decision $\hat{\ell} \in \mathcal{L}$ and suffers excess risk $\mathbb{E}_\mu[\hat{\ell}] - \inf_{\ell \in \mathcal{L}} \mathbb{E}_\mu[\ell]$.

We seek a decision-making procedure for the learner which provides strong excess risk guarantees when $\ell_\star = \arg\min_{\ell \in \mathcal{L}} \mathbb{E}_\mu[\ell]$ is appropriately "simple." To achieve this, we introduce the *$\varepsilon$-outlier-robust $p$-Wasserstein DRO problem*:

$$\inf_{\ell \in \mathcal{L}} \sup_{\nu \in \mathcal{G}: W_p^\varepsilon(\tilde{\mu}, \nu) \leq \rho} \mathbb{E}_\nu[\ell]. \tag{OR-WDRO}$$

## 3.1   Excess Risk Bounds

We quantify the excess risk of decisions made using OR-WDRO for the two most popular choices of order, $p = 1, 2$. Proofs are provided in Supplement B.

---

[2] While not a metric, $W_p^\varepsilon$ is symmetric and satisfies an approximate triangle inequality ([39], Proposition 3).

[3] If a minimizer does not exist for either problem, an infimizing sequence will achieve the same guarantee.

[4] We defer explicit modeling of sampling to Section 3.2 but note that the following results immediately transfer to the $n$-sample setting so long as $\rho$ is taken to be larger than $W_p(\hat{\mu}_n, \mu)$ with high probability.

**Theorem 1** (OR-WDRO risk bound). *Under Setting A, let $\hat{\ell}$ minimize* (OR-WDRO). *Then, writing $c = 2(1-\varepsilon)^{-1/p}$, the excess risk is bounded by*

$$\mathbb{E}_\mu[\hat{\ell}] - \mathbb{E}_\mu[\ell_\star] \leq \begin{cases} \|\ell_\star\|_{\mathrm{Lip}}\big(c\rho + 2\tau_1(\mathcal{G}, 2\varepsilon)\big), & p = 1, \ell_\star \text{ Lipschitz} \\ \|\ell_\star\|_{\dot{H}^{1,2}(\mu)}\big(c\rho + 2\tau_2(\mathcal{G}, 2\varepsilon)\big) + \frac{1}{2}\alpha\big(c\rho + 2\tau_2(\mathcal{G}, 2\varepsilon)\big)^2, & p = 2, \ell_\star \text{ } \alpha\text{-smooth} \end{cases}.$$

Note that $c = O(1)$ since $\varepsilon \leq 0.49$. These bounds imply that the learner can make effective decisions when $\ell_\star$ has low variational complexity[5]. In contrast, there are simple regression settings with TV corruption that drive the excess risk of standard WDRO to infinity. Our proof derives both results as a special case of a general bound in terms of the $\mathsf{W}_p$ regularizer, defined by $\mathcal{R}_{\mu,p}(\rho; \ell) := \sup_{\nu' \in \mathcal{P}(\mathcal{Z}): \mathsf{W}_p(\nu',\nu) \leq \rho} \mathbb{E}_{\nu'}[\ell] - \mathbb{E}_\nu[\ell]$. Introduced in [18], this quantity appears implicitly throughout the WDRO literature. In particular, for each $\ell \in \mathcal{L}$, we derive the following bound:

$$\mathbb{E}_\mu[\hat{\ell}] - \mathbb{E}_\mu[\ell] \leq \mathcal{R}_{\mu,p}\big(\underbrace{c\rho}_{\mathsf{W}_p} + \underbrace{2\tau_p(\mathcal{G}, 2\varepsilon)}_{\mathrm{TV}}; \ell\big), \tag{3}$$

whose radius reveals the effect of each perturbation (viz. Wasserstein versus TV) on the quality of the decision. The first bound of the theorem follows by plugging in $p = 1$ and controlling $\mathcal{R}_{\mu,1}$ via Kantorovich duality. The second bound uses $p = 2$ and controls $\mathcal{R}_{\mu,2}$ by replacing $\ell$ with its Taylor expansion about $Z \sim \mu$. We now instantiate Theorem 1 for the bounded covariance class $\mathcal{G}_{\mathrm{cov}}$.

**Corollary 1** (Risk bounds for $\mathcal{G}_{\mathrm{cov}}$). *Under the setting of Theorem 1 with $\mathcal{G} \subseteq \mathcal{G}_{\mathrm{cov}}$, we have*

$$\mathbb{E}_\mu[\hat{\ell}] - \mathbb{E}_\mu[\ell_\star] \lesssim \begin{cases} \|\ell_\star\|_{\mathrm{Lip}}\big(\rho + \sqrt{d\varepsilon}\big), & p = 1, \ell_\star \text{ Lipschitz} \\ \|\ell_\star\|_{\dot{H}^{1,2}(\mu)}(\rho + \sqrt{d}) + \alpha(\rho^2 + d), & p = 2, \ell_\star \text{ } \alpha\text{-smooth} \end{cases}.$$

Since $\mathcal{G}_{\mathrm{cov}}$ encodes second moment constraints, $\tau_2(\mathcal{G}_{\mathrm{cov}}, \varepsilon) \asymp d$ is independent of $\varepsilon$. Therefore, the first bound is preferable as $\varepsilon \to 0$ if $\|\ell_\star\|_{\dot{H}^{1,2}(\mu)} \approx \|\ell_\star\|_{\mathrm{Lip}}$, while the second is better when $\varepsilon = \Omega(1)$ and $\|\ell_\star\|_{\dot{H}^{1,2}(\mu)} \ll \|\ell_\star\|_{\mathrm{Lip}}$.[6] Distinct trade-offs are observed under stronger tail bounds like sub-Gaussianity, i.e., for $\mathcal{G}_{\mathrm{subG}} := \{\mu \in \mathcal{P}(\mathcal{Z}) : \mathbb{E}_\mu[e^{(\theta^\top(Z - \mathbb{E}[Z])^2}] \leq 2, \forall \theta \in \mathbb{S}^{d-1}\}$.

**Corollary 2** (Risk bounds for $\mathcal{G}_{\mathrm{subG}}$). *Under the setting of Theorem 1 with $\mathcal{G} \subseteq \mathcal{G}_{\mathrm{subG}}$, the excess risk $\mathbb{E}_\mu[\hat{\ell}] - \mathbb{E}_\mu[\ell_\star]$ is bounded up to constants by*

$$\begin{cases} \|\ell_\star\|_{\mathrm{Lip}}\Big(\rho + \sqrt{d + \log \frac{1}{\varepsilon}}\,\varepsilon\Big), & p = 1, \ell_\star \text{ Lipschitz} \\ \|\ell_\star\|_{\dot{H}^{1,2}(\mu)}\Big(\rho + \sqrt{(d + \log \frac{1}{\varepsilon})\varepsilon}\Big) + \alpha\Big(\rho^2 + \big(d + \log \frac{1}{\varepsilon}\big)\varepsilon\Big), & p = 2, \ell_\star \text{ } \alpha\text{-smooth} \end{cases}.$$

**Remark 1** (Comparison to MDE under $\mathsf{W}_p^\varepsilon$). We note that the excess risk $\mathcal{R}_{\mu,p}\big(c\rho + 2\tau_p(\mathcal{G}, 2\varepsilon); \ell_\star\big)$ from (3) can alternatively be obtained by performing standard $p$-WDRO with an expanded radius $c\rho + 2\tau_1(\mathcal{G}, 2\varepsilon)$ around the minimum distance estimate $\hat{\mu} = \mathrm{argmin}_{\nu \in \mathcal{G}} \mathsf{W}_1^\varepsilon(\tilde{\mu}, \nu)$. However, obtaining $\hat{\mu}$ is an expensive preprocessing step, and we are unaware of any efficient algorithms for such MDE in the finite-sample setting. In Supplement D, we explore recentering WDRO around a tractable estimate obtained from iterative filtering [15], but find the resulting risk to be highly suboptimal. Furthermore, the improvements to our risk bounds under low-dimensional structure, which are derived in Section 4, do not extend to decisions obtained from these alternative procedures.

We now show that Theorem 1 cannot be improved in general. In particular, the first bound is minimax optimal over Lipschitz loss families when $\mu \in \mathcal{G}_{\mathrm{cov}}$.

**Proposition 3** (Lower bound). *Fix $\mathcal{Z} = \mathbb{R}^d$ and $\varepsilon \in [0, 0.49]$. For any $L \geq 0$, there exists a family $\mathcal{L} \subseteq \mathrm{Lip}_L(\mathbb{R}^d)$, independent of $\varepsilon$, such that for any decision rule $\mathsf{D} : \mathcal{P}(\mathcal{Z}) \to \mathcal{L}$ there exists a pair $(\mu, \tilde{\mu}) \in \mathcal{G}_{\mathrm{cov}} \times \mathcal{P}(\mathcal{Z})$ with $\mathsf{W}_1^\varepsilon(\tilde{\mu}, \mu) \leq \rho$ satisfying $\mathbb{E}_\mu[\mathsf{D}(\tilde{\mu})] - \inf_{\ell \in \mathcal{L}} \mathbb{E}_\mu[\ell] \gtrsim L\big(\rho + \sqrt{d\varepsilon}\big)$.*

Each family $\mathcal{L}$ encodes a multivariate regression problem. Our proof combines a one-dimensional lower bound of [49] for linear regression with lower bounds of [39] for robust estimation under $\mathsf{W}_1$.

---

[5]The same bounds hold up to $\varepsilon$ additive slack if $\ell_\star$ is only $\varepsilon$-approximately optimal for (OR-WDRO).

[6]Under $\mathsf{W}_p^\varepsilon$ perturbations, one may perform outlier-robust WDRO using any $p' \in [1, p]$, which may be advantageous in terms of the TV component of the excess risk.

## 3.2 Statistical Guarantees

We next formalize a finite-sample model and adapt our excess risk bounds to it.

**Setting B:** Fix $\rho, \varepsilon, \mathcal{L}, \mathcal{G}$ as in Setting A, and let $Z_1, \ldots, Z_n$ be identically and independently distributed (i.i.d.) according to $\mu \in \mathcal{G}$, with empirical measure $\hat{\mu}_n = \frac{1}{n} \sum_{i=1}^{n} \delta_{Z_i}$. Upon observing these clean samples, Nature applies a $\mathsf{W}_p$ perturbation of size $\rho_0$, producing $\{Z_i'\}_{i=1}^n$ with empirical measure $\mu_n'$ such that $\mathsf{W}_p(\hat{\mu}_n, \mu_n') \leq \rho_0$. Finally, Nature corrupts up to $\lfloor \varepsilon n \rfloor$ samples to obtain $\{\tilde{Z}_i\}_{i=1}^n$ with empirical measure $\tilde{\mu}_n$ such that $\|\tilde{\mu}_n - \mu'\|_{\mathsf{TV}} = \frac{1}{n} \sum_{i=1}^{n} \mathbb{1}\{\tilde{Z}_i \neq Z_i\} \leq \varepsilon$. Equivalently, the final dataset satisfies $\mathsf{W}_p^\varepsilon(\tilde{\mu}_n, \hat{\mu}_n) \leq \rho_0$.[7] The learner is now tasked with selecting $\hat{\ell} \in \mathcal{L}$ given $\tilde{\mu}_n$.

The results from Section 3 apply whenever $\rho \geq \rho_0 + \mathsf{W}_p(\mu, \hat{\mu}_n)$. In particular, we obtain the following corollary as an immediate consequence of Theorem 1 and Theorem 3.1 of [32].

**Corollary 3** (Finite-sample risk bounds). *Under Setting B, fix $\hat{\ell} \in \mathcal{L}$ minimizing (OR-WDRO) centered at $\tilde{\mu} = \tilde{\mu}_n$ with $\rho \geq \rho_0 + 100 \, \mathbb{E}[\mathsf{W}_p(\mu, \hat{\mu}_n)]$. Then the excess risk bounds of Theorem 1 hold with probability at least 0.99. If $\mathcal{G} \in \{\mathcal{G}_{\mathrm{cov}}, \mathcal{G}_{\mathrm{subG}}\}$, $p = 1$, and $d \geq 3$, or if $\mathcal{G} = \mathcal{G}_{\mathrm{subG}}$, $p = 2$, and $d \geq 5$, then $\mathbb{E}[\mathsf{W}_p(\mu, \hat{\mu}_n)] \lesssim \sqrt{d} n^{-1/d}$.*

**Remark 2** (Smaller radius). In the classic WDRO setting with $\rho_0 = \varepsilon = 0$, the radius $\rho$ can be taken significantly smaller than $n^{-1/d}$ if $\mathcal{L}$ and $\mu$ are sufficiently well-behaved. For example, [18] proves that $\rho = \widetilde{O}(n^{-1/2})$ gives meaningful risk bounds when $\mu$ satisfies a $T_2$ transportation inequality.[8] While this high-level condition may be hard to verify in practice, Supplement E shows that this improvement can be lifted to an instance of our outlier-robust WDRO problem.

## 3.3 Tractable Reformulations and Computation

For computation, we restrict to $\mu \in \mathcal{G}_{\mathrm{cov}}$. Initially, (OR-WDRO) may appear intractable, since $\mathcal{G}_{\mathrm{cov}}$ is non-convex when viewed as a subset of the cone $\mathcal{M}_+(\mathcal{Z})$. Moreover, enforcing membership to this class is non-trivial. To remedy these issues, we use a cheap preprocessing step to obtain a robust estimate $z_0 \in \mathcal{Z}$ of the mean $\mathbb{E}_\mu[Z]$, and we optimize over the modified class $\mathcal{G}_2(\sigma, z_0) := \{\nu \in \mathcal{P}(\mathcal{Z}) : \mathbb{E}_\nu[\|Z - z_0\|^2] \leq \sigma^2\}$, with $\sigma \gtrsim \|z_0 - \mathbb{E}_\mu[Z]\| + \sqrt{d}$ taken so that $\mu \in \mathcal{G}_2(\sigma, z_0)$. Finally, for technical reasons, we switch to the one-sided robust distance $\mathsf{W}_p^\varepsilon(\mu \| \nu) := \inf_{\mu' \in \mathcal{P}(\mathbb{R}^d) : \mu' \leq \frac{1}{1-\varepsilon} \mu} \mathsf{W}_p(\mu', \nu)$. Altogether, we arrive at the modified DRO problem

$$\inf_{\ell \in \mathcal{L}} \sup_{\nu \in \mathcal{G}_2(\sigma, z_0) : \mathsf{W}_p^\varepsilon(\tilde{\mu}_n \| \nu) \leq \rho} \mathbb{E}_\nu[\ell], \tag{4}$$

which, as stated next, admits risk bounds matching Corollary 1 up to empirical approximation error.

**Proposition 4** (Risk bound for modified problem). *Consider Setting B with $\mathcal{G} \subseteq \mathcal{G}_{\mathrm{cov}}$. Fix $z_0 \in \mathcal{Z}$ such that $\|z_0 - \mathbb{E}_\mu[Z]\| \leq E = O(\rho_0 + \sqrt{d})$, and suppose that $\mathsf{W}_p(\hat{\mu}_n, \mu) \leq \delta$. Take $\hat{\ell}$ minimizing (4) with $\rho = (\rho_0 + \delta)(1 - \varepsilon)^{-1/p} + \tau_p(\mathcal{G}_{\mathrm{cov}}, \varepsilon)$ and $\sigma = \sqrt{d} + E$. We then have*

$$\mathbb{E}_\mu[\hat{\ell}] - \mathbb{E}_\mu[\ell_\star] \lesssim \begin{cases} \|\ell_\star\|_{\mathrm{Lip}} (\rho_0 + \sqrt{d}\varepsilon + \delta), & p = 1, \ell_\star \text{ Lipschitz} \\ \|\ell_\star\|_{\dot{H}^{1,2}(\mu)} (\rho_0 + \sqrt{d} + \delta) + \alpha(\rho_0 + \sqrt{d} + \delta)^2, & p = 2, \ell_\star \text{ } \alpha\text{-smooth} \end{cases}.$$

Parameters $\rho, \sigma$ are taken so that $\mu \in \mathcal{G}_2(\sigma, z_0)$ and $\mathsf{W}_p^\varepsilon(\tilde{\mu}_n \| \mu) \leq \rho$. Noting this, the proof mirrors that of Theorem 1, using a $\mathsf{W}_p$ resilience bound for $\mathcal{G}_2(\sigma, z_0)$. To ensure $\mathsf{W}_p(\hat{\mu}_n, \mu) \leq \delta$ with decent probability, one should take $\delta$ to be an upper bound on $\sup_{\nu \in \mathcal{G}} \mathbb{E}[\mathsf{W}_p(\hat{\nu}_n, \nu)]$. When $p = 2$, this quantity is only finite if $\mathcal{Z}$ is bounded or if $\mathcal{G}$ encodes stronger tail bounds than $\mathcal{G}_{\mathrm{cov}}$ (see, e.g., [32]).

For efficient computation, we must specify a robust mean estimation algorithm to obtain $z_0$ and a procedure for solving (4). The former is achieved by taking a coordinate-wise trimmed mean.

**Proposition 5** (Coarse robust mean estimation). *Consider Setting B with $\mathcal{G} \subseteq \mathcal{G}_{\mathrm{cov}}$ and $\varepsilon \leq 1/3$. For $n = \Omega(\log(d))$, there is a trimmed mean procedure, which applied coordinate-wise to $\{\tilde{Z}_i\}_{i=1}^n$, returns $z_0 \in \mathbb{R}^d$ with $\|z_0 - \mathbb{E}_\mu[Z]\| \lesssim \sqrt{d} + \rho_0$ with probability at least 0.99, in time $\tilde{O}(d)$.*

---

[7]In general, $\{\tilde{Z}_i\}_{i=1}^n$ may be any measurable function of $\{Z_i\}_{i=1}^n$ and independent randomness such that $\mathsf{W}_p^\varepsilon(\tilde{\mu}_n, \hat{\mu}_n) \leq \rho_0$.

[8]We say that $\mu \in T_2(\tau)$ if $\mathsf{W}_2(\nu, \mu) \leq \sqrt{\tau \mathsf{H}(\nu \| \mu)}$, for all $\nu \in \mathcal{P}_2(\mathcal{Z})$, where $\mathsf{H}(\nu \| \mu)$ is relative entropy.

More sophisticated methods, e.g., iterative filtering [15], achieve dimension-free estimation guarantees at the cost of additional sample and computational complexity. We will return to these techniques in Section 4, but overlook them for now since they do not impact worst-case excess risk bounds.

We next show that that the inner maximization problem of (4) can be simplified to a minimization problem involving only two scalars provided the following assumption holds.

**Assumption 1** (Slater condition I). Given the distribution $\tilde{\mu}_n$ and the fixed point $z_0$, there exists $\nu_0 \in \mathcal{P}(\mathcal{Z})$ such that $\mathsf{W}_p^\varepsilon(\tilde{\mu}_n \| \nu_0) < \rho$ and $\mathbb{E}_{\nu_0}[\|Z - z_0\|^2] < \sigma^2$. Additionally, we require $\rho > 0$.

Notice that Assumption 1 indeed holds for $\nu_0 = \mu$ as applied in Proposition 4.

**Proposition 6** (Strong duality). *Under Assumption 1, for any $\ell \in \mathcal{L}$ and $z_0 \in \mathbb{R}^d$, we have*

$$\sup_{\substack{\nu \in \mathcal{G}_2(\sigma, z_0): \\ \mathsf{W}_p^\varepsilon(\tilde{\mu}_n \| \nu) \le \rho}} \mathbb{E}_\nu[\ell] = \inf_{\substack{\lambda_1, \lambda_2 \in \mathbb{R}_+ \\ \alpha \in \mathbb{R}}} \lambda_1 \sigma^2 + \lambda_2 \rho^p + \alpha + \frac{1}{1 - \varepsilon} \mathbb{E}_{\tilde{\mu}_n} \left[ \bar{\ell}(\cdot \,; \lambda_1, \lambda_2, \alpha) \right], \qquad (5)$$

*where $\bar{\ell}(z; \lambda_1, \lambda_2, \alpha) := \sup_{\xi \in \mathcal{Z}} \left[ \ell(\xi) - \lambda_1 \|\xi - z_0\|^2 - \lambda_2 \|\xi - z\|^p - \alpha \right]_+$.*

The minimization problem over $(\lambda_1, \lambda_2, \alpha)$ is an instance of stochastic convex optimization, where the expectation of the implicit function $\bar{\ell}$ is taken w.r.t. the contaminated empirical measure $\tilde{\mu}_n$. In contrast, the dual reformulation for classical WDRO only involves $\lambda_2$ and takes the expectation of the implicit function $\underline{\ell}(z; \lambda_2) := \sup_{\xi \in \mathcal{Z}} \ell(\xi) - \lambda_2 \|\xi - z\|^p$ w.r.t. $\tilde{\mu}_n$. The additional $\lambda_1$ variable above is introduced to account for the clean family $\mathcal{G}_2(\sigma, z_0)$, and the use of partial transportation under $\mathsf{W}_p^\varepsilon$ results in the introduction of the operator $[\cdot]_+$ and the decision variable $\alpha$.

**Remark 3** (Connection to conditional value at risk (CVaR)). The CVaR of a Borel measurable loss function $\ell$ acting on a random vector $Z \sim \mu \in \mathcal{P}(\mathcal{Z})$ with risk level $\varepsilon \in (0, 1)$ is defined as

$$\mathrm{CVaR}_{1-\varepsilon, \mu}[\ell(Z)] = \inf_{\alpha \in \mathbb{R}} \alpha + \frac{1}{1 - \varepsilon} \mathbb{E}_{Z \sim \mu} \left[ [\ell(Z) - \alpha]_+ \right].$$

CVaR is also known as expected shortfall and is equivalent to the conditional expectation of $\ell(Z)$, given that it is above an $\varepsilon$ threshold. This concept is often used in finance to evaluate the market risk of a portfolio. With this definition, the result of Proposition 6 can be written as

$$\sup_{\substack{\nu \in \mathcal{G}_2(\sigma, z_0): \\ \mathsf{W}_p^\varepsilon(\tilde{\mu}_n \| \nu) \le \rho}} \mathbb{E}_\nu[\ell] = \inf_{\lambda_1, \lambda_2 \in \mathbb{R}_+} \lambda_1 \sigma^2 + \lambda_2 \rho^p + \mathrm{CVaR}_{1-\varepsilon, \tilde{\mu}_n} \left[ \sup_{\xi \in \mathcal{Z}} \ell(\xi) - \lambda_1 \|\xi - z_0\|^2 - \lambda_2 \|\xi - Z\|^p \right].$$

When $\varepsilon \to 0$ and $\sigma \to \infty$, whence CVaR reduces to expected value and the constrained class $\mathcal{G}_2(\sigma, z_0)$ expands to $\mathcal{P}(\mathcal{Z})$, the dual formulation above reduces to that of classical WDRO [8, 21].

Evaluating $\bar{\ell}$ requires solving a maximization problem, which could be in itself challenging. To overcome this, we impose additional convexity assumptions, which are standard for WDRO [35, 43].

**Assumption 2** (Convexity condition). The loss $\ell$ is a pointwise maximum of finitely many concave functions, i.e., $\ell(\xi) = \max_{j \in [J]} \ell_j(\xi)$, for some $J \in \mathbb{N}$, where $\ell_j$ is real-valued, l.s.c., and concave[9]. The set $\mathcal{Z}$ is closed and convex. The atoms of $\tilde{\mu}_n$ are in the relative interior of $\mathcal{Z}$.

**Theorem 2** (Convex reformulation). *Under Assumption 1, for any $\ell \in \mathcal{L}$ satisfying Assumption 2 and $z_0 \in \mathbb{R}^d$, we have*

$$\sup_{\substack{\nu \in \mathcal{G}_q(\sigma, z_0): \\ \mathsf{W}_p^\varepsilon(\tilde{\mu}_n \| \nu) \le \rho}} \mathbb{E}_\nu[\ell] = \begin{cases} \inf & \lambda_1 \sigma^2 + \lambda_2 \rho^p + \alpha + \frac{1}{n(1-\varepsilon)} \sum_{i \in [n]} s_i \\ \text{s.t.} & \alpha \in \mathbb{R}, \lambda_1, \lambda_2 \in \mathbb{R}_+, s, \tau_{ij} \in \mathbb{R}_+^n, \zeta_{ij}^\ell, \zeta_{ij}^\mathcal{G}, \zeta_{ij}^\mathsf{W}, \zeta_{ij}^\mathcal{Z} \in \mathbb{R}^d, \forall i \in [n], \forall j \in [J] \\ & s_i \ge (-\ell_j)^*(\zeta_{ij}^\ell) + z_0^\top \zeta_{ij}^\mathcal{G} + \tau_{ij} \\ & \qquad + \tilde{Z}_i^\top \zeta_{ij}^\mathsf{W} + P_h(\zeta_{ij}^\mathsf{W}, \lambda_2) + \chi_\mathcal{Z}^*(\zeta_{ij}^\mathcal{Z}) - \alpha, \quad \forall i \in [n], \forall j \in [J] \\ & \zeta_{ij}^\ell + \zeta_{ij}^\mathcal{G} + \zeta_{ij}^\mathsf{W} + \zeta_{ij}^\mathcal{Z} = 0, \ \|\zeta_{ij}^\mathcal{G}\|^2 \le \lambda_1 \tau_{ij}, \qquad \forall i \in [n], \forall j \in [J], \end{cases}$$

*where $P_h$ is the perspective function (i.e., $P_h(\zeta, \lambda) = \lambda h(\zeta / \lambda)$) of*

---

[9]Generally, any continuous function can be approximated arbitrarily well by a maximum of finitely many concave functions. However, the number of functions needed may be arbitrarily large in general. Fortunately, some losses like the $\ell_\infty$-norm $\|z\|_\infty = \max_{i \in [d], a \in \{\pm 1\}} \sigma z_i$ require only $\mathrm{poly}(d)$ pieces.

$$h(\zeta) := \begin{cases} \chi_{\{z \in \mathbb{R}^d : \|z\| \leq 1\}}(\zeta), & p = 1 \\ \frac{(p-1)^{p-1}}{p^p} \|\zeta\|^{\frac{p}{p-1}}, & p > 1. \end{cases} \tag{6}$$

The minimization problem in Theorem 2 is a finite-dimensional convex program. In Section 5, we use this result in conjunction with Proposition 5 to efficiently perform outlier-robust WDRO.

We conclude this section by characterizing the worst-case distribution, i.e., the optimal adversarial strategy, for our outlier-robust WDRO problem. To that end, we need the primal formulation below.

**Theorem 3** (Worst-case distribution). *Under Assumption 1, for any $\ell \in \mathcal{L}$ satisfying Assumption 2 and $z_0 \in \mathbb{R}^d$, we have*

$$\sup_{\substack{\nu \in \mathcal{G}_q(\sigma, z_0): \\ \mathsf{W}_p^\varepsilon(\tilde{\mu}_n \| \nu) \leq \rho}} \mathbb{E}_\nu[\ell] = \begin{cases} \max & -\sum_{(i,j) \in [n] \times [J]} P_{-\ell_j}(\xi_{ij}, q_{ij}) \\ \text{s.t.} & q_{ij} \in \mathbb{R}_+, \, \xi_{ij} \in q_{ij} \cdot \mathcal{Z} & \forall i \in [n], \forall j \in [J] \\ & \sum_{j \in [J]} q_{ij} \leq \frac{1}{n(1-\varepsilon)} & \forall i \in [n] \\ & \sum_{(i,j) \in [n] \times [J]} q_{ij} = 1 \\ & \sum_{(i,j) \in [n] \times [J]} P_{\|\cdot\|^p}(\xi_{ij} - q_{ij}\tilde{Z}_i, q_{ij}) \leq \rho \\ & \sum_{(i,j) \in [n] \times [J]} P_{\|\cdot\|^2}(\xi_{ij} - q_{ij}z_0, q_{ij}) \leq \sigma^2 \end{cases}$$

*The discrete distribution $\nu^\star = \sum_{(i,j) \in \mathcal{Q}} q_{ij}^\star \delta_{\xi_{ij}^\star / q_{ij}^\star}$ achieves the worst-case expectation on the left-hand side, where $(q_{ij}^\star, \xi_{ij}^\star)_{(i,j) \in [n] \times [J]}$ are optimizers of the maximization problem on the right and $\mathcal{Q} := \{(i,j) \in [n] \times [J] : q_{ij}^\star > 0\}$.*

The maximization problem from Theorem 3 is the conjugate dual of the minimization in Theorem 2. Subsequently, we propose a systematic approach for constructing a discrete distribution based on a solution derived from the maximization problem that achieves the worst-case expected loss.

**Remark 4** (Comparison to WDRO worst-case distribution). Recall that our robust WDRO approach reduces to the classic WDRO approach as $\varepsilon = 0$ and $\sigma \to \infty$. Consequently, this implies that the constraints $\sum_{j \in [J]} q_{ij} \leq 1/(n(1-\varepsilon))$ and $\sum_{(i,j) \in [n] \times [J]} P_{\|\cdot\|^2}(\xi_{ij} - q_{ij}z_0, q_{ij}) \leq \sigma^2$ can be dropped under this specific choice of $\varepsilon$ and $\sigma$. As a result, our construction simplifies to the approach presented in [35, Theorem 4.4] for WDRO problems.

**Remark 5** (Parameter tuning). In practice, $\varepsilon$, $\rho_0$, and the relevant tail bound may be unknown. Thus, in Appendix F, we consider learning under Setting B with $\mathcal{G} = \mathcal{G}_{\text{cov}}(\sigma)$ for potentially unknown $\varepsilon$, $\sigma$, and $\rho_0$. First, we observe that knowledge of upper bounds on these parameters is sufficient to attain risk bounds scaling in terms of said upper bounds. This approach avoids meticulous parameter tuning but may result in suboptimal risk. To efficiently match our risk bounds with known parameters, we show that it is necessary and sufficient to know $\rho_0$ and at least one of $\varepsilon$ or $\sigma$ (up to constant factors).

## 4 Low-Dimensional Features

While Proposition 3 shows that the excess risk bounds from Theorem 1 cannot be improved in general, finer guarantees can be derived when the optimal loss function depends only on $k$-dimensional affine features of the data. Defining $\mathcal{G}^{(k)}$ as the union of the projections $\{U_\# \mu : \mu \in \mathcal{G}\}$ over $U \in \mathbb{R}^{k \times d}$ with $UU^\top = I_k$[10], we improve the excess risk bound of Theorem 1 for this setting.

**Theorem 4** (Excess risk bound). *Under Setting A, let $\hat{\ell}$ minimize* (OR-WDRO)*, and assume that $\ell_\star = \underline{\ell} \circ A$ for an affine map $A : \mathbb{R}^d \to \mathbb{R}^k$ and some $\underline{\ell} : \mathbb{R}^k \to \mathbb{R}$. Writing $c = 2(1-\varepsilon)^{-1/p}$, we have*

$$\mathbb{E}_\mu[\hat{\ell}] - \mathbb{E}_\mu[\ell_\star] \leq \begin{cases} \|\ell_\star\|_{\text{Lip}}\big(c\rho + 2\tau_1(\mathcal{G}^{(k)}, 2\varepsilon)\big), & p = 1, \ell_\star \text{ Lipschitz} \\ \|\ell_\star\|_{\dot{H}^{1,2}(\mu)}\big(c\rho + 2\tau_2(\mathcal{G}^{(k)}, 2\varepsilon)\big) + \frac{1}{2}\alpha\big(c\rho + 2\tau_2(\mathcal{G}^{(k)}, 2\varepsilon)\big)^2, & p = 2, \ell_\star \text{ } \alpha\text{-smooth} \end{cases}.$$

This dependence on $\mathcal{G}^{(k)}$ rather than $\mathcal{G} = \mathcal{G}^{(d)}$ is a substantial improvement when $k \ll d$.

---

[10]If $\mathcal{G}$ is closed under isometries, like $\mathcal{G}_{\text{cov}}$, then $\mathcal{G}^{(k)} = \{U_\# \mu : \mu \in \mathcal{G}\}$ for any such $U$.

**Corollary 4** (Risk bounds for $\mathcal{G}_{\mathrm{cov}}$). *Under the setting of Theorem 4 with $\mathcal{G} \subseteq \mathcal{G}_{\mathrm{cov}}$, we have*

$$\mathbb{E}_\mu[\hat{\ell}] - \mathbb{E}_\mu[\ell_\star] \lesssim \begin{cases} \|\ell_\star\|_{\mathrm{Lip}}(\rho + \sqrt{k}\varepsilon), & p = 1, \ell_\star \text{ Lipschitz} \\ \|\ell_\star\|_{\dot{H}^{1,2}(\mu)}(\rho + \sqrt{k}) + \alpha(\rho^2 + k), & p = 2, \ell_\star \text{ } \alpha\text{-smooth.} \end{cases}$$

We again have a matching lower bound for the Lipschitz setting, this time using $k$-variate regression.

**Proposition 7** (Lower bound). *Fix $\mathcal{Z} = \mathbb{R}^d$ and $\varepsilon \in [0, 0.49]$. For any $L \geq 0$, there exists a family $\mathcal{L} \subseteq \mathrm{Lip}_L(\mathbb{R}^d)$, independent of $\varepsilon$, such that each $\ell \in \mathcal{L}$ decomposes as $\ell = \underline{\ell} \circ A$ for $A \in \mathbb{R}^{k \times d}$ and $\underline{\ell} : \mathbb{R}^k \to \mathbb{R}$, and such that for any decision rule $\mathsf{D} : \mathcal{P}(\mathcal{Z}) \to \mathcal{L}$ there exists a pair $(\mu, \tilde{\mu}) \in \mathcal{G}_{\mathrm{cov}} \times \mathcal{P}(\mathcal{Z})$ with $\mathsf{W}_1^\varepsilon(\tilde{\mu}, \mu) \leq \rho$ satisfying $\mathbb{E}_\mu[\mathsf{D}(\tilde{\mu})] - \inf_{\ell \in \mathcal{L}} \mathbb{E}_\mu[\ell] \gtrsim L(\rho + \sqrt{k}\varepsilon)$.*

For computation, we turn to a slightly modified $n$-sample contamination model. Our analysis for the low-dimensional case only supports additive TV corruptions (sometimes called Huber contamination).

**Setting B′:** Fix $\rho, \varepsilon, \mathcal{L}, \mathcal{G}$ as in Setting A, and fix $m = \lceil (1 - \varepsilon)n \rceil$ for some $n \in \mathbb{N}$. Let $Z_1, \ldots, Z_m$ be drawn i.i.d. from $\mu \in \mathcal{G}$, with empirical measure $\hat{\mu}_m = \frac{1}{m} \sum_{i=1}^m \delta_{Z_i}$. Upon observing these clean samples, Nature applies a $\mathsf{W}_p$ perturbation of size $\rho_0$, producing $\{Z_i'\}_{i=1}^m$ with empirical measure $\mu_m'$ such that $\mathsf{W}_p(\hat{\mu}_m, \mu_m') \leq \rho_0$. Finally, Nature adds $\lfloor \varepsilon n \rfloor$ samples to obtain $\{\tilde{Z}_i\}_{i=1}^n$ with empirical measure $\tilde{\mu}_n$ such that $\mu_m' \leq \frac{1}{1-\varepsilon}\tilde{\mu}_n$. Equivalently, the final dataset satisfies $\mathsf{W}_p^\varepsilon(\tilde{\mu}_n \| \hat{\mu}_m) \leq \rho_0$.

As before, we modify (OR-WDRO) using a centered alternative to $\mathcal{G}_{\mathrm{cov}}$. Defining $\mathcal{G}_{\mathrm{cov}}(\sigma, z_0) := \{\mu \in \mathcal{P}(\mathcal{Z}) : \mathbb{E}_\mu[(Z - z_0)(Z - z_0)^\top] \preceq \sigma^2 I_d\}$, we consider the outlier-robust WDRO problem

$$\inf_{\ell \in \mathcal{L}} \sup_{\nu \in \mathcal{G}_{\mathrm{cov}}(\sigma, z_0) : \mathsf{W}_p^\varepsilon(\tilde{\mu}_n \| \nu) \leq \rho} \mathbb{E}_\nu[\ell]. \tag{7}$$

To start, we provide a corresponding risk bound which matches Corollary 4 when $k = O(1)$.

**Proposition 8** (Risk bound for modified problem). *Consider Setting B′ with $\mathcal{G} \subseteq \mathcal{G}_{\mathrm{cov}}$, and assume $\ell_\star = \underline{\ell} \circ A$ for affine $A : \mathbb{R}^d \to \mathbb{R}^k$ and $\underline{\ell} : \mathbb{R}^k \to \mathbb{R}$. Fix $z_0 \in \mathcal{Z}$ such that $\|z_0 - \mathbb{E}_\mu[Z]\| \leq E = O(\rho_0 + 1)$, and assume $\mathsf{W}_p(\hat{\mu}_m, \mu) \leq \delta$. If $\hat{\ell}$ minimizes (7) with $\rho = \rho_0 + \delta$ and $\sigma = 1 + E$, then*

$$\mathbb{E}_\mu[\hat{\ell}] - \mathbb{E}_\mu[\ell_\star] \lesssim \begin{cases} \|\ell_\star\|_{\mathrm{Lip}}(\sqrt{k}\rho_0 + \sqrt{k}\varepsilon + \delta), & p = 1, \ell_\star \text{ Lipschitz} \\ \|\ell_\star\|_{\dot{H}^{1,2}(\mu)}(\sqrt{k}\rho_0 + \sqrt{k} + \delta) + \alpha(\sqrt{k}\rho_0 + \sqrt{k} + \delta)^2, & p = 2, \ell_\star \text{ } \alpha\text{-smooth} \end{cases}.$$

Here, the stronger requirement for the robust mean estimate, the restriction to additive contamination, and the need to optimize over the centered $\mathcal{G}_{\mathrm{cov}}$ class rather than $\mathcal{G}_2$ all stem from the fact that the resilience term $\tau_p((\mathcal{G}_{\mathrm{cov}})_k, \varepsilon)$ scales with $\sqrt{k}$ rather than $\sqrt{d}$. Fortunately, efficient computation is still possible. First, we employ iterative filtering [15] for dimension-free robust mean estimation.

**Proposition 9** (Refined robust mean estimation). *Consider Setting B or B′ with $\mathcal{G} = \mathcal{G}_{\mathrm{cov}}$ and $\varepsilon \leq 1/12$. For $n = \tilde{\Omega}(d)$, there exists an iterative filtering algorithm which takes $\tilde{\mu}_n$ as input, runs in time $\tilde{O}(nd^2)$, and outputs $z_0 \in \mathbb{R}^d$ such that $\|z_0 - \mathbb{E}_\mu[Z]\| \lesssim \rho_0 + 1$ with probability at least 0.99.*

The analysis requires care when $p = 1$, since $\mathsf{W}_1$ perturbations can arbitrarily increase the initial covariance bound. Fortunately, this increase can be controlled by trimming out a few samples.

Next, we show that computing the inner worst-case expectation in (7) can be simplified into a minimization problem involving only a scalar and a positive semidefinite matrix provided the following assumption holds (which is indeed the case in the setting of Proposition 8).

**Assumption 3** (Slater condition II). *Given the distribution $\tilde{\mu}_n$ and fixed point $z_0$, there exists $\nu_0 \in \mathcal{P}(\mathcal{Z})$ such that $\mathsf{W}_p^\varepsilon(\tilde{\mu}_n \| \nu_0) < \rho$ and $\mathbb{E}_{\nu_0}[(Z - z_0)(Z - z_0)^\top] \prec \sigma^2 I_d$. Further, we require $\rho > 0$.*

**Proposition 10** (Strong duality). *Under Assumption 3, for any $\ell \in \mathcal{L}$ and $z_0 \in \mathbb{R}^d$, we have*

$$\sup_{\substack{\nu \in \mathcal{G}_{\mathrm{cov}}(\sigma, z_0): \\ \mathsf{W}_p^\varepsilon(\tilde{\mu}_n \| \nu) \leq \rho}} \mathbb{E}_\nu[\ell] = \inf_{\substack{\Lambda_1 \in \mathbb{Q}_+^d \\ \lambda_2 \in \mathbb{R}_+, \alpha \in \mathbb{R}}} - z_0^\top \Lambda_1 z_0 + \sigma^2 \operatorname{Tr}[\Lambda_1] + \lambda_2 \rho^p + \alpha + \frac{1}{1 - \varepsilon} \mathbb{E}_{\tilde{\mu}_n}[\overline{\ell}(\cdot; \Lambda_1, \lambda_2, \alpha)],$$

*where $\overline{\ell}(z; \Lambda_1, \lambda_2, \alpha) := \sup_{\xi \in \mathcal{Z}} [\ell(\xi) - \xi^\top \Lambda_1 \xi + 2\xi^\top \Lambda_1 z_0 - \lambda_2 \|\xi - z\|^p - \alpha]_+$.*

The minimization problem over the variables $(\Lambda_1, \lambda_2, \alpha)$ belongs to the class of stochastic convex optimization problems. As before, we show that under the convexity condition from Assumption 2 we obtain a tractable reformulation that does not involve an extra optimization problem for evaluating $\overline{\ell}$.

**Theorem 5** (Convex reformulation)**.** *Under Assumption 3, for any $\ell \in \mathcal{L}$ satisfying Assumption 2 and $z_0 \in \mathbb{R}^d$, we have*

$$\sup_{\substack{\nu \in \mathcal{G}_{\mathrm{cov}}(\sigma, z_0): \\ \mathsf{W}_p^\varepsilon(\tilde{\mu}_n \| \nu) \leq \rho}} \mathbb{E}_\nu[\ell] = \begin{cases} \inf & -z_0^\top \Lambda_1 z_0 + \sigma^2 \operatorname{Tr}[\Lambda_1] + \lambda_2 \rho^p + \alpha + \frac{1}{n(1-\varepsilon)} \sum_{i \in [n]} s_i \\ \mathrm{s.t.} & \alpha \in \mathbb{R}, \ \Lambda_1 \in \mathbb{Q}_+^d, \ \lambda_2 \in \mathbb{R}_+, \ s \in \mathbb{R}_+^n, \\ & \tau_{ij} \in \mathbb{R}_+, \ \zeta_{ij}^\ell, \zeta_{ij}^{\mathcal{G}}, \zeta_{ij}^{\mathsf{W}}, \zeta_{ij}^{\mathcal{Z}} \in \mathbb{R}^d, \quad \forall i \in [n], \forall j \in [J] \\ & s_i \geq (-\ell_j)^*(\zeta_{ij}^\ell) + \tau_{ij} + \tilde{Z}_i^\top \zeta_{ij}^{\mathsf{W}} \\ & \qquad + P_h(\zeta_{ij}^{\mathsf{W}}, \lambda_2) + \chi_{\mathcal{Z}}^*(\zeta_{ij}^{\mathcal{Z}}) - \alpha, \quad \forall i \in [n], \forall j \in [J] \\ & \zeta_{ij}^\ell + \zeta_{ij}^{\mathcal{G}} + \zeta_{ij}^{\mathsf{W}} + \zeta_{ij}^{\mathcal{Z}} = 2\Lambda_1 z_0, (\zeta_{ij}^{\mathcal{G}})^\top \Lambda_1^{-1} \zeta_{ij}^{\mathcal{G}} \leq 4\tau_{ij}, \quad \forall i \in [n], \forall j \in [J], \end{cases}$$

*where $P_h$ is the perspective function of $h$ defined in* (6).

## 5 Experiments

Lastly, we implement our tractable reformulations and validate their excess risk bounds. Fixing $\mathcal{Z} = \mathcal{X} \times \mathcal{Y} = \mathbb{R}^{d-1} \times \mathbb{R}$, we focus on linear regression with the mean absolute deviation loss, i.e., $\mathcal{L} = \{\ell_\theta(x, y) = |\theta^\top x - y| : \theta \in \mathbb{R}^d\}$. The experiments below were run in 80 minutes on an M1 MacBook Air with 16GB RAM. See Supplement G for full details and additional experiments treating classification and multivariate regression. Code is available at `https://github.com/sbnietert/outlier-robust-WDRO`.

Fix $\mathcal{Z} = \mathbb{R}^d$ for $d \geq 2$, $\rho = \varepsilon = 0.1$, and $\theta_\star \in \mathbb{S}^{d-2}$. Taking $X \sim \mathcal{N}(0, I_{d-1})$, we fix clean data $(X, \theta_\star^\top X) \sim \mu$ and corrupted data $(\tilde{X}, \tilde{Y}) \sim \tilde{\mu}$ such that $(\tilde{X}, \tilde{Y}) = (X, \theta_\star^\top X + \rho)$ with probability $1 - \varepsilon$ and $(\tilde{X}, \tilde{Y}) = (CX, -C^2\theta_\star^\top X)$ with probability $\varepsilon$, so that $\mathsf{W}_p^{\varepsilon_0}(\tilde{\mu} \| \mu) \leq \rho$. In Figure 2 (top), we fix $d = 10, C = 8$ and compare the excess risk $\mathbb{E}_\mu[\ell_{\hat{\theta}}] - \mathbb{E}_\mu[\ell_{\theta_\star}]$ of standard WDRO and outlier-robust WDRO with $\mathcal{G} = \mathcal{G}_2$, as described by Proposition 4 and implemented via Theorem 2. The results are averaged over $T = 20$ runs for sample size $n \in \{10, 20, 50, 75, 100\}$. We run outlier-robust WDRO with corruption fraction $\hat{\varepsilon} \in \{0, \varepsilon, 2\varepsilon\}$, achieving low excess risk when $\hat{\varepsilon} \geq \varepsilon$ as predicted. In Figure 2 (bottom), to highlight the Section 4 improve-

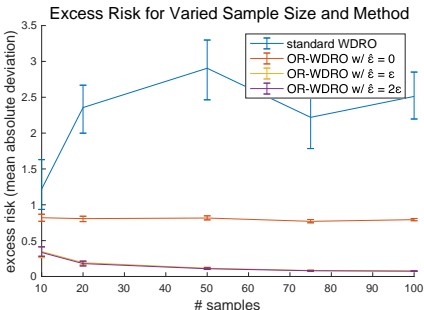

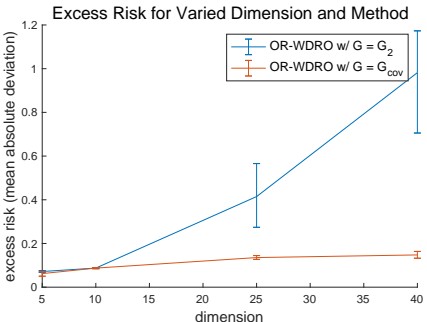

**Figure 2:** Excess risk of standard WDRO and several forms of outlier-robust WDRO for linear regression under $\mathsf{W}_p$ and TV corruptions, with varied sample size and dimension.

ments due to low-dimensional structure, we fix $n = 20, C = 100$ and compare the excess risk of outlier-robust WDRO with $\mathcal{G} = \mathcal{G}_2$ to that with $\mathcal{G} = \mathcal{G}_{\mathrm{cov}}$, as described by Proposition 8 and implemented via Theorem 5. We average over $T = 10$ runs and present results for dimension $d \in \{5, 10, 25, 40\}$. Confidence bands in both plots depict the top and bottom 10% quantiles among 100 bootstrapped means from the $T$ runs. Implementations were performed in MATLAB using the YALMIP toolbox [33] and the Gurobi and SeDuMi solvers [23, 50].

## 6 Concluding Remarks

In this work, we have introduced a novel framework for outlier-robust WDRO that allows for both geometric and non-geometric perturbations of the data distribution, as captured by $\mathsf{W}_p$ and TV, respectively. We provided minimax optimal excess risk bounds and strong duality results, with the latter enabling efficient computation via convex reformulations. There are numerous directions for future work, including refined statistical guarantees for $\rho \ll \rho_0 + n^{-1/d}$ and convex reformulations for distribution families beyond $\mathcal{G}_{\mathrm{cov}}$. Overall, our approach enables principled, data-driven decision-making in realistic scenarios where observations may be subject to adversarial contamination.

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

## A Preliminary Results on Robust OT and Wasserstein DRO

We first recall properties of the robust Wasserstein distance $W_p^\varepsilon$ which will be used throughout the supplement. To start, we show that our definition coincides with another based on partial OT, considered in [39]. In what follows, we fix $p \geq 1$, write $c\mathcal{P}(\mathcal{Z}) := \{c\mu : \mu \in \mathcal{P}(\mathcal{Z})\}$, and, for $\mu, \nu \in c\mathcal{P}(\mathcal{Z})$, we define $\Pi(\mu, \nu) := c\Pi(\mu/c, \nu/c)$ and $W_p(\mu, \nu)^p := cW_p(\mu/c, \nu/c)$.

**Lemma 1** ($W_p^\varepsilon$ as partial OT). *Fix $\varepsilon \in [0, 1]$. For any $\mu, \nu \in \mathcal{P}(\mathcal{Z})$, we have*

$$W_p^\varepsilon(\mu, \nu) := \inf_{\substack{\mu' \in \mathcal{P}(\mathcal{Z}) \\ \|\mu' - \mu\|_{\mathsf{TV}} \leq \varepsilon}} W_p(\mu', \nu) = \inf_{\substack{\nu' \in \mathcal{P}(\mathcal{Z}) \\ \|\nu' - \nu\|_{\mathsf{TV}} \leq \varepsilon}} W_p(\mu, \nu') = \inf_{\substack{\mu', \nu' \in (1-\varepsilon)\mathcal{P}(\mathcal{Z}) \\ \mu' \leq \mu, \, \nu' \leq \nu}} W_p(\mu', \nu').$$

*Proof.* We write $\underline{W}_p^\varepsilon(\mu, \nu)$ for the rightmost expression; this is definition of $W_p^\varepsilon$ considered in [39]. We first show that $\underline{W}_p^\varepsilon(\mu, \nu) \leq W_p^\varepsilon(\mu, \nu)$. Fix any $\mu'$ feasible for the $W_p^\varepsilon$ problem. Then, by the approximate triangle inequality for $\underline{W}_p^\varepsilon$ (Proposition 3 of [39]), we have

$$\underline{W}_p^\varepsilon(\mu, \nu) \leq \underline{W}_p^\varepsilon(\mu, \mu') + W_p(\mu', \nu) \leq W_p(\mu', \nu).$$

Indeed, writing $c := (\mu \wedge \mu')(\mathcal{Z}) \geq 1 - \varepsilon$, the last inequality uses that $\underline{W}_p^\varepsilon(\mu, \mu') \leq W_p(\frac{1-\varepsilon}{c}\mu \wedge \mu', \frac{1-\varepsilon}{c}\mu \wedge \mu') = 0$. Infimizing over feasible $\mu'$ gives that $\underline{W}_p^\varepsilon(\mu, \nu) \leq W_p^\varepsilon(\mu, \nu)$.

For the other direction, take any $\mu', \nu'$ feasible for the $\underline{W}_p^\varepsilon$ problem. Let $\pi \in \Pi(\mu', \nu')$ be any optimal coupling for the $W_p(\mu', \nu')$ problem, and write $\mu'' = \mu' + (\nu - \nu') \in \mathcal{P}(\mathcal{Z})$. Defining the coupling $\pi' = \pi + (\mathrm{Id}, \mathrm{Id})_{\#}(\nu - \nu') \in \Pi(\mu'', \nu)$, we compute

$$W_p(\mu'', \nu)^p \leq \int_{\mathcal{Z} \times \mathcal{Z}} \|x - y\|^p \, d\pi'(x, y) = \int_{\mathcal{Z} \times \mathcal{Z}} \|x - y\|^p \, d\pi(x, y) = W_p(\mu', \nu').$$

By construction, $\|\mu'' - \mu\|_{\mathsf{TV}} \leq \varepsilon$, and so $W_p^\varepsilon(\mu, \nu) \leq W_p(\mu', \nu')$. Infimizing over feasible $\mu', \nu'$ gives that $W_p^\varepsilon(\mu, \nu) \leq \underline{W}_p^\varepsilon(\mu, \nu)$. $\square$

We thus inherit several results for $\underline{W}_p$ given in [39].

**Lemma 2** (Approximate triangle inequality [39]). *If $\mu, \nu, \kappa \in \mathcal{P}(\mathcal{Z})$ and $\varepsilon_1, \varepsilon_2 \in [0, 1]$, then*

$$W_p^{\varepsilon_1 + \varepsilon_2}(\mu, \nu) \leq W_p^{\varepsilon_1}(\mu, \kappa) + W_p^{\varepsilon_2}(\kappa, \nu).$$

**Lemma 3** ($W_p^\varepsilon$ modulus of continuity, [39], Lemma 3). *For any $\mathcal{G} \subseteq \mathcal{P}(\mathcal{Z})$, we have*

$$\sup_{\substack{\alpha, \beta \in \mathcal{G} \\ W_p^\varepsilon(\alpha, \beta) \leq \rho}} W_p(\alpha, \beta) \leq (1 - \varepsilon)^{-1/p}\rho + 2\tau_p(\mathcal{G}, \varepsilon).$$

**Lemma 4** (One-sided vs. two-sided $W_p^\varepsilon$). *For $\mu, \nu \in \mathcal{P}(\mathcal{Z})$, we have*

$$W_p^\varepsilon(\mu\|\nu) \leq (1 - \varepsilon)^{-1/p}W_p^\varepsilon(\mu, \nu) + \tau_p(\nu, \varepsilon).$$

*Proof.* Fix any $\mu', \nu' \in (1 - \varepsilon)\mathcal{P}(\mathcal{Z})$ with $\mu' \leq \mu$ and $\nu' \leq \nu$. By design, we have

$$\begin{aligned}
W_p^\varepsilon(\mu\|\nu) &\leq W_p\left(\tfrac{1}{1-\varepsilon}\mu', \nu\right) \\
&\leq W_p\left(\tfrac{1}{1-\varepsilon}\mu', \tfrac{1}{1-\varepsilon}\nu'\right) + W_p\left(\tfrac{1}{1-\varepsilon}\nu', \nu\right) \\
&= (1 - \varepsilon)^{-1/p}W_p(\mu', \nu') + \tau_p(\nu, \varepsilon)
\end{aligned}$$

Infimizing over $\mu'$ and $\nu'$ and applying Lemma 1 gives the lemma. $\square$

Next, we specify explicit constants for the $W_p$ resilience of $\mathcal{G}_{\mathrm{cov}}$. Our analysis goes through the related notion of mean resilience [49], defined by $\tau(\mu, \varepsilon) = \sup_{\mu' \in \mathcal{P}(\mathcal{Z}): \mu' \leq \frac{1}{1-\varepsilon}\mu} \|\mathbb{E}_{\mu'}[Z] - \mathbb{E}_\mu[Z]\|$. We say that $Z \sim \mu \in \mathcal{P}(\mathcal{Z})$ is $(\tau_0, \varepsilon)$-resilient in mean or under $W_p$ if $\mu \in \tau(\tau_0, \varepsilon)$ or $\mu \in \tau_p(\tau_0, \varepsilon)$.

**Lemma 5** ($W_p$ resilience for $\mathcal{G}_2$ and $\mathcal{G}_{\mathrm{cov}}$). *Fix $\varepsilon \in [0, 1)$ and $\sigma \geq 0$. For $1 \leq p \leq 2$, we have* $\tau_p(\mathcal{G}_2(\sigma), \varepsilon) \leq 4\sigma\varepsilon^{1/p - 1/2}(1 - \varepsilon)^{-1/p}$. *Moreover, we have* $\tau_p(\mathcal{G}_{\mathrm{cov}}(\sigma), \varepsilon) \leq \tau_p(\mathcal{G}_2(\sqrt{d}\sigma), \varepsilon)$.

*Proof.* Fix any $Z \sim \mu \in \mathcal{G}_2(\sigma, z_0)$. By definition, we have $\mathbb{E}[(\|Z - z_0\|^p)^{2/p}] = \mathbb{E}[\|Z - z_0\|^2] \leq \sigma^2 = (\sigma^p)^{2/p}$. Thus, standard bounds (e.g., Lemma E.2 of [58]) give that $\|Z - z_0\|^p$ is $(\sigma^p \varepsilon^{1-p/2}(1-\varepsilon)^{-1}, \varepsilon)$-resilient in mean. By Lemma 7 of [39], we thus have that $Z$ is $(2\sigma\varepsilon^{1/p-1/2}(1-\varepsilon)^{-1/p} + 2\varepsilon^{1/p}\sigma, \varepsilon)$-resilient under $\mathsf{W}_p$. This gives the first result. For the second, we observe that for $Z \sim \mu \in \mathcal{G}_{\mathrm{cov}}(\sigma)$, we have $\mathbb{E}[\|Z - \mathbb{E}[Z]\|^2] = \mathrm{tr}(\Sigma_\mu) \leq \sqrt{d}\sigma$. $\qquad\square$

Lastly, we turn to the $\mathsf{W}_p$ regularizer.

**Lemma 6** (Controlling $\mathsf{W}_p$ regularizer, [18], Lemmas 1 and 2)**.** *For any $\ell \in \mathcal{L}$, we have $\mathcal{R}_{\nu,1}(\rho; \ell) \leq \rho\|\ell\|_{\mathrm{Lip}}$, with equality if $\ell$ is convex. For $\alpha$-smooth $\ell$, we have $|\mathcal{R}_{\nu,2}(\rho; \ell) - \rho\|\ell\|_{\dot{H}^{1,2}(\nu)}| \leq \frac{1}{2}\alpha\rho^2$.*

The factor of 1/2 under smoothness was not present in Lemma 2 of [18], since the proof only used that $|\ell(z) - \ell(z_0) - \nabla(z_0)^\top(z - z_0)| \leq \alpha\|z - z_0\|^2$, instead of the tight upper bound of $\frac{\alpha}{2}\|z - z_0\|^2$.

This quantity naturally bounds the excess risk of standard WDRO.

**Lemma 7** (WDRO excess risk bound)**.** *Under Setting A with $\varepsilon = 0$, the standard WDRO estimate $\hat{\ell} = \mathrm{argmin}_{\ell \in \mathcal{L}} \sup_{\nu \in \mathcal{P}(\mathcal{Z}): \mathsf{W}_p(\tilde{\mu}, \nu) \leq \rho} \mathbb{E}_\nu[\ell]$ satisfies $\mathbb{E}_\mu[\hat{\ell}] - \mathbb{E}_\mu[\ell_\star] \leq \mathcal{R}_{\mu,p}(2\rho; \ell_\star)$.*

*Proof.* We bound

$$
\begin{aligned}
\mathbb{E}_\mu[\hat{\ell}] - \mathbb{E}_\mu[\ell_\star] &\leq \sup_{\substack{\nu \in \mathcal{P}(\mathcal{Z}) \\ \mathsf{W}_p(\nu, \tilde{\mu}) \leq \rho}} \mathbb{E}_\nu[\hat{\ell}] - \mathbb{E}_\mu[\ell_\star] && (\mathsf{W}_p(\mu, \tilde{\mu}) \leq \rho) \\
&\leq \sup_{\substack{\nu \in \mathcal{P}(\mathcal{Z}) \\ \mathsf{W}_p(\nu, \tilde{\mu}) \leq \rho}} \mathbb{E}_\nu[\ell_\star] - \mathbb{E}_\mu[\ell_\star] && \text{(optimality of } \hat{\ell}) \\
&\leq \sup_{\substack{\nu \in \mathcal{P}(\mathcal{Z}) \\ \mathsf{W}_p(\nu, \mu) \leq 2\rho}} \mathbb{E}_\nu[\ell_\star] - \mathbb{E}_\mu[\ell_\star] && (\mathsf{W}_p \text{ triangle inequality}) \\
&= \mathcal{R}_{\mu,p}(2\rho; \ell_\star),
\end{aligned}
$$

as desired. $\qquad\square$

Note that this bound does not incorporate the distributional assumptions encoded by $\mathcal{G}$.

# B  Proofs for Section 3

## B.1  Proof of Theorem 1

We compute

$$
\begin{aligned}
\mathbb{E}_\mu[\hat{\ell}] - \mathbb{E}_\mu[\ell] &\leq \sup_{\substack{\nu \in \mathcal{G} \\ \mathsf{W}_p^\varepsilon(\tilde{\mu}, \nu) \leq \rho}} \mathbb{E}_\nu[\hat{\ell}] - \mathbb{E}_\mu[\ell] && (\mu \in \mathcal{G}, \mathsf{W}_p^\varepsilon(\tilde{\mu}, \mu) \leq \rho) \\
&\leq \sup_{\substack{\nu \in \mathcal{G} \\ \mathsf{W}_p^\varepsilon(\tilde{\mu}, \nu) \leq \rho}} \mathbb{E}_\nu[\ell] - \mathbb{E}_\mu[\ell] && (\hat{\ell} \text{ optimal for (OR-WDRO))} \\
&\leq \sup_{\substack{\nu \in \mathcal{G} \\ \mathsf{W}_p^{2\varepsilon}(\nu, \mu) \leq 2\rho}} \mathbb{E}_\nu[\ell] - \mathbb{E}_\mu[\ell] && \text{(Lemma 2)} \\
&\leq \sup_{\substack{\nu \in \mathcal{G} \\ \mathsf{W}_p(\nu, \mu) \leq c\rho + 2\tau_p(\mathcal{G}, 2\varepsilon)}} \mathbb{E}_\nu[\ell] - \mathbb{E}_\mu[\ell] && \text{(Lemma 3)} \\
&\leq \sup_{\substack{\nu \in \mathcal{P}(\mathcal{Z}) \\ \mathsf{W}_p(\nu, \mu) \leq c\rho + 2\tau_p(\mathcal{G}, 2\varepsilon)}} \mathbb{E}_\nu[\ell] - \mathbb{E}_\mu[\ell] && (\mathcal{G} \subseteq \mathcal{P}(\mathcal{Z})) \\
&= \mathcal{R}_{\mu,p}\big(c\rho + 2\tau_p(\mathcal{G}, 2\varepsilon); \ell\big).
\end{aligned}
$$

Combining this bound with Lemma 6 gives the theorem. $\qquad\square$

## B.2 Proof of Corollary 1

The corollary follows as an immediate consequence of Theorem 1 and the resilience bounds of Proposition 2.

## B.3 Proof of Corollary 2

The corollary follows as an immediate consequence of Theorem 1 and the resilience bound $\tau_p(\mathcal{G}_{\mathrm{subG}}, \varepsilon) \lesssim \sqrt{d + p + \log \frac{1}{\varepsilon}}\, \varepsilon^{1/p}$ established in [39, Theorem 2].

## B.4 Proof of Proposition 3

For ease of presentation, suppose $d = 2m$ is even. Consider $\mathbb{R}^d$ as $\mathbb{R}^m \times \mathbb{R}^m$, fix $w \in \mathbb{R}^m$ with $\|w\| = \rho$, and let $\mathcal{L}$ consist of the following loss functions:

$$\ell_{+,0}(x, y) := L\|x + y\|$$
$$\ell_{-,0}(x, y) := L\|x - y\|$$
$$\ell_{+,1}(x, y) := L\|x + y - w\|$$
$$\ell_{-,1}(x, y) := L\|x - y + w\|.$$

Fixing corrupted measure $\tilde{\mu} = \delta_0$, we consider the following candidates for the clean measure $\mu$:

$$\mu_{+,0} := (1 - \varepsilon)\delta_0 + \varepsilon(\mathrm{Id}, -\mathrm{Id})_{\#}\mathcal{N}(0, I_d/\varepsilon)$$
$$\mu_{-,0} := (1 - \varepsilon)\delta_0 + \varepsilon(\mathrm{Id}, +\mathrm{Id})_{\#}\mathcal{N}(0, I_d/\varepsilon)$$
$$\mu_{+,1} := (1 - \varepsilon)\delta_{(0,w)} + \varepsilon(\mathrm{Id}, -\mathrm{Id} + w)_{\#}\mathcal{N}(0, I_d/\varepsilon)$$
$$\mu_{-,1} := (1 - \varepsilon)\delta_{(0,w)} + \varepsilon(\mathrm{Id}, \mathrm{Id} + w))_{\#}\mathcal{N}(0, I_d/\varepsilon),$$

where $\mathrm{Id} : x \mapsto x$ is the identity map. By design, $\mathsf{W}_1^\varepsilon(\tilde{\mu}\|\mu_{+,0}), \mathsf{W}_1^\varepsilon(\tilde{\mu}\|\mu_{-,0}), \mathsf{W}_1^\varepsilon(\tilde{\mu}\|\mu_{+,1})$, and $\mathsf{W}_1^\varepsilon(\tilde{\mu}\|\mu_{-,1})$ are all at most $\rho$ and $\mu_{+,0}, \mu_{-,0}, \mu_{+,1}, \mu_{-,1} \in \mathcal{G}_{\mathrm{cov}}$. Moreover,

$$\mathbb{E}_{\mu_{+,0}}[\ell_{+,0}] = \mathbb{E}_{\mu_{-,0}}[\ell_{-,0}] = \mathbb{E}_{\mu_{+,1}}[(\ell_{+,1}) = \mathbb{E}_{\mu_{-,1}}[\ell_{-,1}] = 0$$

$$\mathbb{E}_{\mu_{+,0}}[\ell_{-,0}] = \mathbb{E}_{\mu_{-,0}}[\ell_{+,0}] = \mathbb{E}_{\mu_{+,1}}[\ell_{-,1}] = \mathbb{E}_{\mu_{-,1}}[\ell_{+,1}] = 2L\varepsilon\, \mathbb{E}_{Z \sim \mathcal{N}(0, I_d/\varepsilon)}[\|Z\|] \gtrsim L\sqrt{d\varepsilon}$$

$$\mathbb{E}_{\mu_{+,0}}[\ell_{+,1}] = \mathbb{E}_{\mu_{+,1}}[\ell_{+,0}] = \mathbb{E}_{\mu_{-,0}}[\ell_{-,1}] = \mathbb{E}_{\mu_{-,1}}[\ell_{-,0}] = L\|w\| = L\rho.$$

Thus, for any $\hat{\ell} = \mathsf{D}(\tilde{\mu}) \in \mathcal{L}$, there exists $\mu \in \{\mu_{+,0}, \mu_{-,0}, \mu_{+,1}, \mu_{-,1}\}$ such that

$$\mathbb{E}_\mu[\hat{\ell}] - \inf_{\ell \in \mathcal{L}} \mathbb{E}_\mu[\ell] = \mathbb{E}_\mu[\hat{\ell}] \gtrsim L\max\{\rho, \sqrt{d\varepsilon}\} \asymp L(\rho + \sqrt{d\varepsilon}). \qquad \square$$

## B.5 Proof of Proposition 4

Since $\mu \in \mathcal{G}_{\mathrm{cov}}$, we have

$$\mathbb{E}_\mu\big[\|Z - z_0\|^2\big]^{\frac{1}{2}} \leq \mathbb{E}_\mu[\|Z - \mathbb{E}_\mu[Z]\|^2]^{\frac{1}{2}} + \|\mathbb{E}_\mu[Z] - z_0\|$$
$$= \mathrm{tr}(\Sigma_\mu)^{\frac{1}{2}} + \|\mathbb{E}_\mu[Z] - z_0\|$$
$$\leq \sqrt{d} + \|\mathbb{E}_\mu[Z] - z_0\| \leq \sigma.$$

Consequently, we have $\mu \in \mathcal{G}_2(\sigma, z_0)$. Next, we bound

$$\mathsf{W}_p^\varepsilon(\tilde{\mu}_n\|\mu) := \inf_{\substack{\nu \in \mathcal{P}(\mathcal{Z}) \\ \nu \leq \frac{1}{1-\varepsilon}\tilde{\mu}_n}} \mathsf{W}_p(\nu, \mu)$$

$$\leq \inf_{\substack{\nu, \mu' \in \mathcal{P}(\mathcal{Z}) \\ \nu \leq \frac{1}{1-\varepsilon}\tilde{\mu}_n \\ \mu' \leq \frac{1}{1-\varepsilon}\mu}} \mathsf{W}_p(\nu, \mu') + \mathsf{W}_p(\mu', \mu)$$

$$\leq \inf_{\substack{\nu, \mu' \in \mathcal{P}(\mathcal{Z}) \\ \nu \leq \frac{1}{1-\varepsilon}\tilde{\mu}_n \\ \mu' \leq \frac{1}{1-\varepsilon}\mu}} \mathsf{W}_p(\nu, \mu') + \tau_p(\mu, \varepsilon) \qquad \text{(definition of } \tau_p\text{)}$$

$$= (1 - \varepsilon)^{-\frac{1}{p}} \mathsf{W}_p^\varepsilon(\tilde{\mu}_n, \mu) + \tau_p(\mu, \varepsilon) \qquad \text{(Lemma 1)}$$

$$\leq (1 - \varepsilon)^{-\frac{1}{p}} (\rho_0 + \delta) + \tau_p(\mathcal{G}_{\mathrm{cov}}, \varepsilon) = \rho.$$

Writing $\mathcal{G}' = \mathcal{G}_2(\sigma, z_0)$ and mirroring the proof of Theorem 1, we have for each $\ell \in \mathcal{L}$ that

$$\mathbb{E}_\mu[\hat{\ell}] - \mathbb{E}_\mu[\ell] \leq \sup_{\substack{\nu \in \mathcal{G}' \\ \mathsf{W}_p^\varepsilon(\tilde{\mu}_n \| \nu) \leq \rho}} \mathbb{E}_\nu[\hat{\ell}] - \mathbb{E}_\mu[\ell]$$

$$\leq \sup_{\substack{\nu \in \mathcal{G}' \\ \mathsf{W}_p^\varepsilon(\tilde{\mu}_n \| \nu) \leq \rho}} \mathbb{E}_\nu[\ell] - \mathbb{E}_\mu[\ell] \qquad (\hat{\ell} \text{ optimal for } (4))$$

$$\leq \sup_{\substack{\nu \in \mathcal{G}' \\ \mathsf{W}_p^{2\varepsilon}(\nu, \mu) \leq 2\rho}} \mathbb{E}_\nu[\ell] - \mathbb{E}_\mu[\ell] \qquad \text{(Lemma 2)}$$

$$\leq \sup_{\substack{\nu \in \mathcal{G}' \\ \mathsf{W}_p(\nu, \mu) \leq c\rho + 2\tau_p(\mathcal{G}', 2\varepsilon)}} \mathbb{E}_\nu[\ell] - \mathbb{E}_\mu[\ell] \qquad \text{(Lemma 3)}$$

$$\leq \sup_{\substack{\nu \in \mathcal{P}(\mathcal{Z}) \\ \mathsf{W}_p(\nu, \mu) \leq c\rho + 2\tau_p(\mathcal{G}', 2\varepsilon)}} \mathbb{E}_\nu[\ell] - \mathbb{E}_\mu[\ell]$$

$$= \mathcal{R}_{\mu,p}\big(c\rho + 2\tau_p(\mathcal{G}', 2\varepsilon); \ell\big)$$

$$\leq \mathcal{R}_{\mu,p}\big(c\rho + 8\sigma(2\varepsilon)^{\frac{1}{p} - \frac{1}{2}}(1 - 2\varepsilon)^{-\frac{1}{p}}; \ell\big). \qquad \text{(Lemma 5)}$$

When $p = 1$, we bound the regularizer radius by

$$c\rho + 8\sigma\sqrt{2\varepsilon}(1 - 2\varepsilon)^{-1} \lesssim \rho_0 + \delta + \tau_1(\mathcal{G}_{\mathrm{cov}}, \varepsilon) + (\sqrt{d} + \rho_0)\sqrt{\varepsilon} \lesssim \rho_0 + \delta + \sqrt{d}\varepsilon,$$

using Lemma 5. Similarly, when $p = 2$, we bound the radius by

$$c\rho + 8\sigma(1 - 2\varepsilon)^{-\frac{1}{2}} \lesssim \rho_0 + \delta + \tau_2(\mathcal{G}_{\mathrm{cov}}, \varepsilon) + (\sqrt{d} + \rho_0) \lesssim \rho_0 + \delta + \sqrt{d}.$$

Taking $\ell = \ell_\star$ and applying Lemma 7 gives the proposition.

### B.6 Proof of Proposition 5

To start, we fix $d = 1$. Given $0 \leq \gamma < 1/2$ and $\nu \in \mathcal{P}(\mathbb{R})$ with cumulative distribution function (CDF) $F_\nu$, define the $\gamma$-trimming $\mathsf{T}_\gamma(\nu) \in \mathcal{P}(\mathbb{R})$ as the law of $F_\nu^{-1}(U)$, where $U \sim \mathrm{Unif}([\gamma, 1 - \gamma])$, and let $m_\gamma(\nu) := \mathbb{E}_{\mathsf{T}_\gamma(\nu)}[Z]$ denote the $\gamma$-trimmed mean. If $\nu = \frac{1}{|A|} \sum_{a \in A} \delta_a$ is uniform over a finite set $A = \{a_1 < a_2 < \cdots < a_n\}$ and $\gamma n$ is an integer, we have $m_\gamma(\nu) = \frac{1}{(1-2\gamma)n} \sum_{i=\gamma n+1}^{(1-\gamma)n} a_i$.

Our robust mean estimate when $d = 1$ is $z_0 = m_\gamma(\tilde{\mu}_n)$ with $\gamma = 1/3$. The smaller choice of $\gamma = \varepsilon$ gives tighter guarantees at the cost of increased sample complexity; we keep the larger choice since we only require a coarse estimate.

**Lemma 8.** *Consider Setting B with $d = 1$, $\mathcal{G} = \mathcal{G}_{\mathrm{cov}}$, $\rho_0 = 0$, $\varepsilon \leq 1/3$. Fix sample size $n = \Omega(\log(1/\delta))$, for $0 < \delta < 1/2$. Then, $\|m_\gamma(\tilde{\mu}_n) - \mathbb{E}_\mu[Z]\| \lesssim 1$ with probability at least $1 - \delta$.*

*Proof.* This follows by Proposition 1.18 of [16] applied to the distribution $\mu$ with corruption fraction $\gamma$, $\varepsilon' = 4\gamma/3 < 1/2$, and resilience bound $\tau(\mathcal{G}_{\mathrm{cov}}, 2\varepsilon') \lesssim \sqrt{\gamma} \lesssim 1$. $\qquad \square$

Now, since we are free to permute the order of the TV and $\mathsf{W}_p$ corruptions (see Lemma 1), there exist $\{W_i\}_{i=1}^n \subseteq \mathbb{R}$ with empirical measure $\nu_n \in \mathcal{P}(\mathbb{R})$ such that $\|\nu_n - \hat{\mu}_n\|_{\mathsf{TV}} \leq \varepsilon$ and $\mathsf{W}_p(\nu_n, \tilde{\mu}_n) \leq \rho_0$. By Lemma 8, we have $|m_\gamma(\nu_n) - \mathbb{E}_\mu[Z]| \lesssim 1$. Of course, we do not observe $\nu_n$, so this result is not immediately useful. To apply this fact, we use that $\mathsf{T}_\gamma$ is an approximate Wasserstein contraction.

**Lemma 9.** *If $\alpha, \beta \in \mathcal{P}(\mathbb{R})$ and $0 \leq \gamma < 1/2$, then $\mathsf{W}_p(\mathsf{T}_\gamma(\alpha), \mathsf{T}_\gamma(\beta)) \leq (1 - 2\gamma)^{-1/p} \mathsf{W}_p(\alpha, \beta)$.*

*Proof.* Writing $F_\alpha$ and $F_\beta$ for the CDFs of $\alpha$ and $\beta$, respectively, we compute

$$\mathsf{W}_p(\mathsf{T}_\gamma(\alpha), \mathsf{T}_\gamma(\beta))^p = \frac{1}{1 - 2\gamma} \int_\gamma^{1-\gamma} |F_\alpha^{-1}(t) - F_\beta^{-1}(t)|^p \, dt$$

$$\leq \frac{1}{1 - 2\gamma} \int_0^1 |F_\alpha^{-1}(t) - F_\beta^{-1}(t)|^p \, \mathrm{d}t = \frac{1}{1 - 2\gamma} \mathsf{W}_p(\alpha, \beta)^p,$$

as desired. □

Applying Lemma 9, we bound

$$
\begin{aligned}
|m_\gamma(\tilde{\mu}_n) - \mathbb{E}_\mu[Z]| &\leq |m_\gamma(\nu_n) - \mathbb{E}_\mu[Z]| + |m_\gamma(\nu_n) - m_\gamma(\tilde{\mu}_n)| \\
&\leq |m_\gamma(\nu_n) - \mathbb{E}_\mu[Z]| + \mathsf{W}_1(\mathsf{T}_\gamma(\nu_n), \mathsf{T}_\gamma(\tilde{\mu}_n)) \\
&\lesssim 1 + \mathsf{W}_1(\nu_n, \tilde{\mu}_n) \\
&\lesssim 1 + \rho_0.
\end{aligned}
$$

This matches the proposition statement when $d = 1$.

For general $d > 1$, we propose the coordinate-wise trimmed estimate $z_0 \in \mathbb{R}^d$ given by $(z_0)_i = m_\gamma(\mathbf{e}_i^\top {}_\# \tilde{\mu}_n)$. Plugging in $\delta \leftarrow 1/(100d)$ into Lemma 8 and taking a union bound over coordinates, we condition on the 0.99 probability event that the one-dimensional bound holds for all coordinates. We can then bound

$$
\begin{aligned}
\|z_0 - \mathbb{E}_\mu[Z]\|^2 &= \sum_{i=1}^d \left( m_\gamma(\mathbf{e}_i^\top {}_\# \tilde{\mu}_n) - \mathbb{E}_\mu[\mathbf{e}_i^\top Z] \right)^2 \\
&\lesssim d + \sum_{i=1}^d \mathsf{W}_1(\mathbf{e}_i^\top {}_\# \nu_n, \mathbf{e}_i^\top {}_\# \tilde{\mu}_n)^2 \\
&\leq d + \mathsf{W}_1(\nu_n, \tilde{\mu}_n)^2 \\
&\leq d + \rho_0^2,
\end{aligned}
$$

as desired. The penultimate inequality is a consequence of the reverse Minkowski inequality.

**Lemma 10.** *Fix $\alpha, \beta \in \mathcal{P}(\mathbb{R}^d)$ Write $\alpha_i = \mathbf{e}_i^\top {}_\# \alpha$ and $\beta_i = \mathbf{e}_i^\top {}_\# \beta$, $i \in [d]$, for their coordinate-wise marginals. We then have*

$$\sum_{i=1}^d \mathsf{W}_1(\alpha_i, \beta_i)^2 \leq \mathsf{W}_1(\alpha, \beta)^2. \tag{8}$$

*Proof.* Take $(X, Y)$ to be an optimal coupling for the $\mathsf{W}_1(\alpha, \beta)$ problem. Writing $\Delta_i = \|Y_i - X_i\|^2$ for $i \in [d]$, the right hand side of (8) can be written as the $L^{1/2}$ norm $\| \sum_{i=1}^d \Delta_i \|_{1/2}$. We then bound

$$\sum_{i=1}^d \mathsf{W}_1(\alpha_i, \beta_i)^2 \leq \sum_{i=1}^d \|\Delta_i\|_{1/2} \leq \left\| \sum_{i=1}^d \Delta_i \right\|_{1/2},$$

where the final inequality follows by the reverse Minkowski inequality for $L^{1/2}$. □

### B.7 Proof of Proposition 6

We have

$$
\begin{aligned}
\sup_{\substack{\nu \in \mathcal{G}_2(\sigma, z_0): \\ \mathsf{W}_p^\varepsilon(\tilde{\mu}_n \| \nu) \leq \rho}} \mathbb{E}_\nu[\ell] &= \sup_{\substack{\mu', \nu \in \mathcal{P}(\mathcal{Z}) \\ \pi \in \Pi(\mu', \nu)}} \left\{ \mathbb{E}_\nu[\ell] : \begin{array}{l} \mathbb{E}_\nu[\|Z - z_0\|^2] \leq \sigma^2, \\ \mathbb{E}_\pi[\|Z' - Z\|^p] \leq \rho^p, \\ \mu' \leq \frac{1}{1-\varepsilon} \tilde{\mu}_n \end{array} \right\} \\
&= \sup_{\substack{m \in \mathbb{R}^n \\ \nu_1, \dots, \nu_n \in \mathcal{P}(\mathcal{Z})}} \left\{ \sum_{i \in [n]} m_i \, \mathbb{E}_{\nu_i}[\ell] : \begin{array}{l} \sum_{i \in [n]} m_i \, \mathbb{E}_{\nu_i}[\|Z_i - z_0\|^2] \leq \sigma^2, \\ \sum_{i \in [n]} m_i \, \mathbb{E}_{\nu_i}[\|\tilde{Z}_i - Z_i\|^p] \leq \rho^p, \\ 0 \leq m_i \leq \frac{1}{n(1-\varepsilon)}, \, \forall i \in [n] \\ \sum_{i \in [n]} m_i = 1 \end{array} \right\},
\end{aligned}
$$

where the first equality follows from the definitions of $\mathcal{G}_2(\sigma, z_0)$ and $\mathsf{W}_p^\varepsilon(\tilde{\mu}_n \| \nu)$. The second equality holds because $\tilde{\mu}_n = \frac{1}{n} \sum_{i \in [n]} \delta_{\tilde{Z}_i}$, which implies that the distributions $\mu', \nu$ and $\pi$ take the form

$\mu' = \sum_{i \in [n]} m_i \delta_{\tilde{Z}_i}$, $\nu = \sum_{i \in [n]} m_i \nu_i$, and $\pi = \sum_{i \in [n]} m_i \delta_{\tilde{Z}_i} \otimes \nu_i$, respectively. Note that the distribution $\nu_i$ models the probability distribution of the random variable $Z$ condition on the event that $Z' = \tilde{z}_i$. Using the definition of the expectation operator and introducing the positive measure $\nu_i' = m_i \nu_i$ for every $i \in [n]$, we arrive at

$$\sup_{\substack{\nu \in \mathcal{G}_2(\sigma, z_0): \\ \mathsf{W}_p^\varepsilon(\tilde{\mu}_n \| \nu) \leq \rho}} \mathbb{E}_\nu[\ell] = \sup_{\substack{m \in \mathbb{R}^n \\ \nu_1', \ldots, \nu_n' \geq 0}} \left\{ \sum_{i \in [n]} \mathbb{E}_{\nu_i'}[\ell] : \begin{array}{l} \sum_{i \in [n]} \int_{\mathcal{Z}} \|z_i - z_0\|^2 d\nu_i'(z_i) \leq \sigma^2, \\ \sum_{i \in [n]} \int_{\mathcal{Z}} \|z_i - \tilde{Z}_i\|^p d\nu_i'(z_i) \leq \rho^p, \\ 0 \leq m_i \leq \frac{1}{n(1-\varepsilon)}, \ \forall i \in [n], \\ \sum_{i \in [n]} m_i = 1 \\ \int_{\mathcal{Z}} d\nu_i'(z_i) = m_i, \ \forall i \in [n] \end{array} \right\}$$

$$= \inf_{\substack{\lambda_1, \lambda_2 \in \mathbb{R}_+ \\ r, s \in \mathbb{R}^n, \alpha \in \mathbb{R}}} \left\{ \lambda_1 \sigma^q + \lambda_2 \rho^p + \alpha + \frac{\sum_{i \in [n]} s_i}{n(1-\varepsilon)} : \begin{array}{l} s_i \geq \max\{0, r_i - \alpha\}, \ \forall i \in [n], \\ r_i \geq \ell(\xi) - \lambda_1 \|\xi - z_0\|^2 - \lambda_2 \|\xi - \tilde{Z}_i\|^p, \\ \forall \xi \in \mathcal{Z}, \forall i \in [n] \end{array} \right\},$$

where the second equality follows from strong duality, which holds because the Slater condition outlined in [47, Proposition 3.4] is satisfied thanks to Assumption 1. The proof concludes by removing the decision variables $r$ and $s$ and using the definition of $\tilde{\mu}_n$. $\qquad\square$

## B.8 Proof of Theorem 2

The proof requires the following preparatory lemma. We say that the function $f$ is proper if $f(x) > -\infty$ and $\mathrm{dom}(f) \neq \emptyset$.

**Lemma 11.** *The followings hold.*

    *(i) Let $f(x) = \lambda g(x - x_0)$, where $\lambda \geq 0$ and $g : \mathbb{R}^d \to \mathbb{R}$ is l.s.c. and convex. Then, $f^*(y) = x_0^\top y + \lambda g^*(y/\lambda)$.*

    *(ii) Let $f(x) = \|x\|^p$ for some $p \geq 1$. Then, $f^*(y) = h(y)$, where the function $h$ is defined as in (6).*

    *(iii) Let $f(x) = x^\top \Sigma x$ for some $\Sigma \succ 0$. Then, $f^*(y) = \frac{1}{4} y^\top \Sigma^{-1} y$.*

*Proof.* The claims follows from [25, §E, Proposition 1.3.1 ], [57, Lemma B.8 (ii)] and [25, §E, Example 1.1.3], respectively. $\qquad\square$

*Proof of Theorem 2.* By Proposition 6 and exploiting the definition of $\tilde{\mu}_n$, we have

$$\sup_{\substack{\nu \in \mathcal{G}_2(\sigma, z_0): \\ \mathsf{W}_p^\varepsilon(\tilde{\mu}_n \| \nu) \leq \rho}} \mathbb{E}_\nu[\ell] = \left\{ \begin{array}{ll} \inf & \lambda_1 \sigma^2 + \lambda_2 \rho^p + \alpha + \frac{1}{n(1-\varepsilon)} \sum_{i \in [n]} s_i \\ \text{s.t.} & \alpha \in \mathbb{R}, \lambda_1, \lambda_2 \in \mathbb{R}_+, s \in \mathbb{R}_+^n \\ & s_i \geq \sup_{\xi \in \mathcal{Z}} \ell(\xi) - \lambda_1 \|\xi - z_0\|^2 - \lambda_2 \|\xi - \tilde{Z}_i\|^p - \alpha \quad \forall i \in [n] \end{array} \right.$$

$$= \left\{ \begin{array}{ll} \inf & \lambda_1 \sigma^2 + \lambda_2 \rho^p + \alpha + \frac{1}{n(1-\varepsilon)} \sum_{i \in [n]} s_i \\ \text{s.t.} & \alpha \in \mathbb{R}, \lambda_1, \lambda_2 \in \mathbb{R}_+, s \in \mathbb{R}_+^n \\ & s_i \geq \sup_{\xi \in \mathcal{Z}} \ell_j(\xi) - \lambda_1 \|\xi - z_0\|^2 \\ & \hspace{3cm} - \lambda_2 \|\xi - \tilde{Z}_i\|^p - \alpha \quad \forall i \in [n], \forall j \in [J] \end{array} \right.$$

$$\tag{9}$$

where the second equality follows form Assumption 2. For any fixed $i \in [n]$ and $j \in [J]$, we have

$$\sup_{\xi \in \mathcal{Z}} \ell_j(\xi) - \lambda_1 \|\xi - z_0\|^2 - \lambda_2 \|\xi - \tilde{Z}_i\|^p - \alpha$$

$$= \left\{ \begin{array}{ll} \inf & (-\ell_j)^*(\zeta_{ij}^\ell) + z_0^\top \zeta_{ij}^{\mathcal{G}} + \tau_{ij} + \tilde{Z}_i^\top \zeta_{ij}^{\mathsf{W}} + P_h(\zeta_{ij}^{\mathsf{W}}, \lambda_2) + \chi_{\mathcal{Z}}^*(\zeta_{ij}^{\mathcal{Z}}) - \alpha \\ \text{s.t.} & \tau_{ij} \in \mathbb{R}_+^n, \ \zeta_{ij}^\ell, \zeta_{ij}^{\mathcal{G}}, \zeta_{ij}^{\mathsf{W}}, \zeta_{ij}^{\mathcal{Z}} \in \mathbb{R}^d, \ \zeta_{ij}^\ell + \zeta_{ij}^{\mathcal{G}} + \zeta_{ij}^{\mathsf{W}} + \zeta_{ij}^{\mathcal{Z}} = 0, \ \|\zeta_{ij}^{\mathcal{G}}\|^2 \leq \lambda_1 \tau_{ij} \end{array} \right.$$

where the equality is a result of strong duality due to [57, Theorem 2] and Lemma 11. The claim follows by substituting all resulting dual minimization problems into (9) and eliminating the corresponding minimization operators. $\qquad\square$

## B.9 Proof of Theorem 3

Thanks to Remark 3, we have

$$\sup_{\substack{\nu\in\mathcal{G}_2(\sigma,z_0):\\ \mathsf{W}_p^\varepsilon(\tilde{\mu}_n\|\nu)\leq\rho}} \mathbb{E}_\nu[\ell]$$

$$= \inf_{\lambda_1,\lambda_2\in\mathbb{R}_+} \lambda_1\sigma^2 + \lambda_2\rho^p + \mathrm{CVaR}_{1-\varepsilon,\tilde{\mu}_n}\left[\sup_{\xi\in\mathcal{Z}} \ell(\xi) - \lambda_1\|\xi-z_0\|^2 - \lambda_2\|\xi-Z\|^p\right]$$

$$= \inf_{\lambda_1,\lambda_2\in\mathbb{R}_+} \sup_{m\in\mathcal{M}_\varepsilon} \lambda_1\sigma^2 + \lambda_2\rho^p + \sum_{i\in[n]} m_i\left[\sup_{\xi\in\mathcal{Z}} \ell(\xi) - \lambda_1\|\xi-z_0\|^2 - \lambda_2\|\xi-\tilde{Z}_i\|^p\right],$$

where $\mathcal{M}_\varepsilon := \{m \in \mathbb{R}_+^n : m_i \leq 1/(n(1-\varepsilon)), \forall i \in [n], \sum_{i\in[n]} m_i = 1\}$, and the equality follows from the primal representation of the CVaR as a coherent risk measure [3, Example 4.1] and the fact that $\tilde{\mu}_n$ is discrete. By Assumption 2, we have

$$\sup_{m\in\mathcal{M}_\varepsilon} \sum_{i\in[n]} m_i\left[\sup_{\xi\in\mathcal{Z}} \ell(\xi) - \lambda_1\|\xi-z_0\|^2 - \lambda_2\|\xi-\tilde{Z}_i\|^p\right]$$

$$= \sup_{m\in\mathcal{M}_\varepsilon} \sup_{\xi_{ij}\in\mathcal{Z}} \sum_{i\in[n]} m_i\left[\max_{j\in[J]} \ell(\xi_{ij}) - \lambda_1\|\xi_{ij}-z_0\|^2 - \lambda_2\|\xi_{ij}-\tilde{Z}_i\|^p\right]$$

$$= \sup_{q\in\mathcal{Q}_\varepsilon} \sup_{\xi_{ij}\in\mathcal{Z}} \sum_{(i,j)\in[n]\times[J]} q_{ij}\left[\ell(\xi_{ij}) - \lambda_1\|\xi_{ij}-z_0\|^2 - \lambda_2\|\xi_{ij}-\tilde{Z}_i\|^p\right],$$

where $\mathcal{Q}_\varepsilon := \{q \in \mathbb{R}_+^{n\times J} : \sum_{j\in[J]} q_{ij} \leq \frac{1}{n(1-\varepsilon)}, \forall i \in [n], \sum_{(i,j)\in[n]\times[J]} q_{ij} = 1\}$, and the last equality easily follows by introducing the variables $q_{ij}$ as a means to merge the variables $m_i$ and the maximum operator. Note that the final supremum problem is nonconvex as we have bilinearity between $q_{ij}$ and $\xi_{ij}$. Using the definition of the perspective function and the simple variable substitution $\xi_{ij} \leftarrow \xi_{ij}/q_{ij}$, however, one can convexify this problem and arrive at

$$\sup_{\substack{\nu\in\mathcal{G}_2(\sigma,z_0):\\ \mathsf{W}_p^\varepsilon(\tilde{\mu}_n\|\nu)\leq\rho}} \mathbb{E}_\nu[\ell] = \inf_{\lambda_1,\lambda_2\in\mathbb{R}_+} \sup_{\substack{q\in\mathcal{Q}_\varepsilon\\ \xi_{ij}\in q_{ij}\cdot\mathcal{Z}}} \left\{\lambda_1\sigma^2 + \lambda_2\rho^p - \sum_{(i,j)\in[n]\times[J]} P_{-\ell_j}(\xi_{ij}, q_{ij})\right.$$

$$\left. - \lambda_1 P_{\|\cdot\|^2}(\xi_{ij} - q_{ij}z_0, q_{ij}) - \lambda_2 P_{\|\cdot\|^p}(\xi_{ij} - q_{ij}\tilde{Z}_i, q_{ij})\right\}.$$

Note that strong duality holds similar to the proof of [57, Section 6]. This allows us to interchange the infimum and supremum without changing the optimal value of the problem. Then, infimizing over $\lambda_1$ and $\lambda_2$, and noticing that the resulting supremum problem is solvable, since the feasible set is compact, conclude the first part of the proof. Following the discussion in [57, § 6], it is easy to show that the proposed discrete distribution $\nu^\star$ solves the worst-case expectation problem. The details are omitted for brevity. This concludes the proof.

# C  Proofs for Section 4

## C.1  Proof of Theorem 4

We start with the following lemma.

**Lemma 12.** *Under the setting of Theorem 4, we may decompose $\ell_\star = \tilde{\ell} \circ Q$ for $Q \in \mathbb{R}^{k\times d}$ with $QQ^\top = I_k$ and some $\tilde{\ell} : \mathbb{R}^k \to \mathbb{R}^d$. For any such decomposition, we have*

$$\sup_{\substack{\nu\in\mathcal{G}\\ \mathsf{W}_p^\varepsilon(\nu,\tilde{\mu})\leq\rho}} \mathbb{E}_\nu[\ell_\star] = \sup_{\substack{\nu\in\mathcal{G}_k\\ \mathsf{W}_p^\varepsilon(\nu,Q_\#\tilde{\mu})\leq\rho}} \mathbb{E}_\nu[\tilde{\ell}].$$

*Proof.* To start, we decompose $A(z) = RQz + z_0$, where $Q \in \mathbb{R}^{k \times d}$ with $QQ^\top = I_k$, $R \in \mathbb{R}^{k \times k}$, and $z_0 \in \mathbb{R}^k$. Note that the orthogonality condition ensures that $Q^\top$ isometrically embeds $\mathbb{R}^k$ into $\mathbb{R}^d$. We then choose $\tilde{\ell}(w) = \underline{\ell}(Rw + z_0)$.

Next, given $\nu \in \mathcal{G}$, we have $Q_{\#}\nu \in \mathcal{G}^{(k)}$ with $\mathsf{W}_p^\varepsilon(Q_{\#}\nu, Q_{\#}\tilde{\mu}) \leq \mathsf{W}_p^\varepsilon(\nu, \tilde{\mu})$, and $\mathbb{E}_\nu[\ell] = \mathbb{E}_{Q_{\#}\nu}[\tilde{\ell}]$. Thus, the RHS supremum is always at least as large as the LHS. It remains to show the reverse.

Fix $\nu \in \mathcal{G}_k$ with $\mathsf{W}_p^\varepsilon(\nu, Q_{\#}\tilde{\mu})$. Take any $\nu' \in \mathcal{P}(\mathbb{R}^k)$ with $\mathsf{W}_p(\nu, \nu') \leq \rho$ and $\|\nu' - Q_{\#}\tilde{\mu}\|_{\mathsf{TV}} \leq \varepsilon$. Write $\kappa = Q_{\#}^\top \nu \in \mathcal{G}$ and $\kappa' = Q_{\#}^\top \nu'$. Since $Q^\top$ is an isometric embedding, we have $\kappa \in \mathcal{G}$, $\mathsf{W}_p(\kappa, \kappa') = \mathsf{W}_p(\nu, \nu') \leq \rho$, and $\|\kappa' - \tilde{\mu}\|_{\mathsf{TV}} = \|\nu' - Q_{\#}\tilde{\mu}\|_{\mathsf{TV}} \leq \varepsilon$. Finally, we have $\mathbb{E}_\nu[\ell] = \mathbb{E}_\kappa[\tilde{\ell}]$. Thus, the RHS supremum is no greater than the LHS, and we have the desired equality. $\qquad\square$

Writing $\mu_k = Q_{\#}\mu$, we mirror the proof of Theorem 1 and bound

$$
\begin{aligned}
\mathbb{E}_\mu[\hat{\ell}] - \mathbb{E}_\mu[\ell_\star] &\leq \sup_{\substack{\nu \in \mathcal{G} \\ \mathsf{W}_p^\varepsilon(\tilde{\mu}, \nu) \leq \rho}} \mathbb{E}_\nu[\hat{\ell}] - \mathbb{E}_\mu[\ell_\star] && (\mu \in \mathcal{G}, \mathsf{W}_p^\varepsilon(\tilde{\mu}, \mu) \leq \rho) \\
&\leq \sup_{\substack{\nu \in \mathcal{G} \\ \mathsf{W}_p^\varepsilon(\tilde{\mu}, \nu) \leq \rho}} \mathbb{E}_\nu[\ell_\star] - \mathbb{E}_\mu[\ell_\star] && (\hat{\ell} \text{ optimal for (OR-WDRO)}) \\
&= \sup_{\substack{\nu \in \mathcal{G}^{(k)} \\ \mathsf{W}_p^\varepsilon(\nu, Q_{\#}\tilde{\mu}) \leq \rho}} \mathbb{E}_\nu[\tilde{\ell}] - \mathbb{E}_{\mu_k}[\tilde{\ell}] && (\text{Lemma 12}) \\
&\leq \sup_{\substack{\nu \in \mathcal{G}^{(k)} \\ \mathsf{W}_p^{2\varepsilon}(\nu, \mu_k) \leq 2\rho}} \mathbb{E}_\nu[\tilde{\ell}] - \mathbb{E}_{\mu_k}[\tilde{\ell}] && (\text{Lemma 2}) \\
&\leq \sup_{\substack{\nu \in \mathcal{G}^{(k)} \\ \mathsf{W}_p(\nu, \mu_k) \leq c\rho + 2\tau_p(\mathcal{G}^{(k)}, 2\varepsilon)}} \mathbb{E}_\nu[\tilde{\ell}] - \mathbb{E}_{\mu_k}[\tilde{\ell}] && (\text{Lemma 3}) \\
&\leq \sup_{\substack{\nu \in \mathcal{P}(\mathcal{Z}) \\ \mathsf{W}_p(\nu, \mu_k) \leq c\rho + 2\tau_p(\mathcal{G}^{(k)}, 2\varepsilon)}} \mathbb{E}_\nu[\tilde{\ell}] - \mathbb{E}_{\mu_k}[\tilde{\ell}] && (\mathcal{G} \subseteq \mathcal{P}(\mathcal{Z})) \\
&= \mathcal{R}_{\mu_k, p}\big(c\rho + 2\tau_p(\mathcal{G}^{(k)}, 2\varepsilon); \tilde{\ell}\big).
\end{aligned}
$$

To obtain the theorem, we apply Lemma 6 and observe that $\|\tilde{\ell}\|_{\mathrm{Lip}} = \|\ell_\star\|_{\mathrm{Lip}}$ and $\|\tilde{\ell}\|_{\dot{H}^{1,2}(\mu_k)} = \|\ell_\star\|_{\dot{H}^{1,2}(\mu)}$ (since $Q^\top$ is an isometric embedding from $\mathbb{R}^k$ into $\mathbb{R}^d$). $\qquad\square$

### C.2 Proof of Corollary 4

This follows as an immediate consequence of Theorem 4 and Corollary 1. $\qquad\square$

### C.3 Proof of Proposition 7

We simply instantiate the lower bound construction from Proposition 3 in $\mathbb{R}^k$, viewed as a subspace of $\mathbb{R}^d$. Extending each $\ell \in \mathcal{L}$ to $\mathbb{R}^d$ by $\ell(z) = \ell(z_{1:k})$, the same lower bound applies with $d \leftarrow k$. $\qquad\square$

### C.4 Proof of Proposition 8

For $Z \sim \mu \in \mathcal{G}_{\mathrm{cov}}$ and $z_0, w \in \mathbb{R}^d$, we bound

$$
\begin{aligned}
\mathbb{E}\big[w^\top(Z - z_0)(Z - z_0)^\top w\big]^{\frac{1}{2}} &= \mathbb{E}\big[w^\top(Z - z_0)^2\big]^{\frac{1}{2}} \\
&\leq \mathbb{E}\big[w^\top(Z - \mathbb{E}[Z])^2\big]^{\frac{1}{2}} + |w^\top(\mathbb{E}[Z] - z_0)| \\
&\leq \|w\|(1 + \|z_0 - \mathbb{E}[Z]\|).
\end{aligned}
$$

Consequently, we have $\mu \in \mathcal{G}_{\mathrm{cov}}(1 + \|z_0 - \mathbb{E}[Z]\|, z_0) \subseteq \mathcal{G}_{\mathrm{cov}}(\sigma, z_0)$. Moreover, we have $\mathsf{W}_p^\varepsilon(\tilde{\mu}_n, \mu) \leq \mathsf{W}_p(\mu'_m, \mu) \leq \rho_0 + \delta = \rho$. Decomposing $\ell_\star = \tilde{\ell} \circ Q$ as in Lemma 12 and writing $\mu_k = Q_{\#}\mu$, the same approach applied in the proof of Theorem 4 gives

$$
\mathbb{E}_\mu[\hat{\ell}] - \mathbb{E}_\mu[\ell_\star] \leq \mathcal{R}_{\mu_k, p}\big(c\rho + 2\tau_p(Q_{\#}\mathcal{G}_{\mathrm{cov}}(\sigma, z_0), 2\varepsilon); \tilde{\ell}\big)
$$

$$\leq \mathcal{R}_{\mu_k, p}\big(c\rho + O(\sigma\sqrt{k}\varepsilon^{\frac{1}{p}-\frac{1}{2}}); \tilde{\ell}\big).$$

When $p = 1$, we bound the regularizer radius by

$$c\rho + O(\sigma\sqrt{k}\varepsilon) \lesssim \rho_0 + \delta + (1 + \rho_0)\sqrt{k}\varepsilon \lesssim \sqrt{k}\rho_0 + \sqrt{k}\varepsilon + \delta,$$

using Lemma 5. Similarly, when $p = 2$, we bound the radius by

$$c\rho + O(\sigma\sqrt{k}) \lesssim \rho_0 + \delta + (1 + \rho_0)\sqrt{k} \lesssim \sqrt{k}\rho_0 + \sqrt{k} + \delta$$

We then conclude as in Theorem 4. $\qquad\square$

### C.5 Proof of Proposition 9

Since iterative filtering works by identifying a subset of samples with bounded covariance and $\mathsf{W}_p$ perturbations can arbitrarily increase second moments when $p < 2$, it is not immediately clear how to apply this method. Fortunately, $\mathsf{W}_2$ perturbations have a bounded affect on second moments, and, by trimming out a small fraction of samples, we can ensure that a $\mathsf{W}_1$ step is bounded under $\mathsf{W}_2$.

**Lemma 13.** *For any $\mu, \nu \in \mathcal{P}(\mathbb{R}^d)$ and $0 < \gamma \leq 1$, there exists $\nu' \in \mathcal{P}(\mathbb{R}^d)$ with $\|\nu' - \nu\|_{\mathsf{TV}} \leq \tau$ such that $\mathsf{W}_1(\mu, \nu') \leq \mathsf{W}_1(\mu, \nu)$ and $\mathsf{W}_\infty(\mu, \nu') \leq \mathsf{W}_1(\mu, \nu)/\gamma$.*

*Proof.* Let $(X, Y)$ be a coupling of $\mu$ and $\nu$ with $\mathbb{E}[\|X - Y\|] = \mathsf{W}_1(\mu, \nu)$. Writing $\Delta = \|X - Y\|$, the event $E$ that $\Delta \leq \mathsf{W}_1(\mu, \nu)/\gamma$ has probability at least $1 - \gamma$ by Markov's inequality. We shall take $\nu'$ be the law of $Y' = \mathbb{1}_E Y + (1 - \mathbb{1}_E)X$. By design, $\|\nu' - \nu\|_{\mathsf{TV}} \leq \gamma$, and

$$\mathsf{W}_1(\mu, \nu') \leq \mathbb{E}[\|X - Y'\|^p] = \mathbb{E}[\mathbb{1}_E \|X - Y\|^p] \leq \mathsf{W}_1(\mu, \nu).$$

Finally, we bound $\mathsf{W}_\infty(\mu, \nu') \leq \|\mathbb{1}_E \Delta\|_\infty \leq \mathsf{W}_1(\mu, \nu)/\gamma$. $\qquad\square$

For consistency between Settings B and B$'$, we let $m = n$ if we are the former. Thus, in both cases, we have $\mathsf{W}_p(\hat{\mu}_m, \mu'_m) \leq \rho_0$ and $\|\mu'_m - \tilde{\mu}_n\|_{\mathsf{TV}} \leq \varepsilon \leq \varepsilon_0 := 1/12$. It is well known that the empirical measure $\hat{\mu}_m$ will inherit the bounded covariance of $\mu$ for $m$ sufficiently large, so long as a small fraction of samples are trimmed out. In particular, by Lemma A.18 of [15] and our sample complexity requirement, there exists a uniform discrete measure $\alpha_k$ over a subset of $k = (1 - \varepsilon_0/120)m$ points, such that $\|\mathbb{E}_{\alpha_k}[Z] - \mathbb{E}_\mu[Z]\| \lesssim 1$ and $\Sigma_{\alpha_k} \preceq O(1)I_d$ with probability at least 0.99.

Moreover, applying Lemma 13 with $\gamma = \varepsilon_0/120$, there exists $\beta \in \mathcal{P}(\mathbb{R}^d)$ with $\|\beta - \mu'_m\|_{\mathsf{TV}} \leq \varepsilon_0/120$ and $\mathsf{W}_2(\beta, \hat{\mu}_n) \leq 240\rho_0/\varepsilon_0$. Combining, we have that $\mathsf{W}_2^{\varepsilon_0/120+\varepsilon_0/120+\varepsilon_0}(\alpha_m, \tilde{\mu}_n) = \mathsf{W}_2^{61\varepsilon_0/60}(\alpha_k, \tilde{\mu}_n) \leq 240\rho_0/\varepsilon_0$, and so there exists $\kappa \in \mathcal{P}(\mathbb{R}^d)$ such that $\mathsf{W}_\infty(\alpha_k, \kappa) \leq 240\rho_0/\varepsilon_0$ and $\|\kappa - \tilde{\mu}_n\|_{\mathsf{TV}} \leq 61\varepsilon_0/60$. The $\mathsf{W}_\infty$ bound implies that $\Sigma_\kappa \preceq O(1 + \rho_0^2\varepsilon_0^{-2})I_d$.

Thus, by the proof of Theorem 4.1 in [26] and our sample complexity requirement, the iterative filtering algorithm (Algorithm 1 therein, based on that of [15]) applied with an outlier fraction of $61/60\varepsilon_0 \leq 1/10$ returns a reweighting of $\tilde{\mu}_n$ whose mean $z_0 \in \mathbb{R}^d$ is within $O(\sqrt{\varepsilon_0} + \rho_0/\sqrt{\varepsilon_0}) = O(1 + \rho_0)$ of that of $\kappa$. Thus, we obtain

$$\begin{aligned}
\|z_0 - \mathbb{E}_\mu[Z]\| &\leq \|\mathbb{E}_\kappa[Z] - \mathbb{E}_\mu[Z]\| + O(1 + \rho_0) \\
&\leq \|\mathbb{E}_{\alpha_k}[Z] - \mathbb{E}_\mu[Z]\| + O(1 + \rho_0) \\
&\leq O(1 + \rho_0),
\end{aligned}$$

as desired. $\qquad\square$

### C.6 Proof of Proposition 10

We have

$$\sup_{\substack{\nu \in \mathcal{G}_{\mathrm{cov}}(\sigma, z_0): \\ \mathsf{W}_p^\varepsilon(\tilde{\mu}_n\|\nu) \leq \rho}} \mathbb{E}_\nu[\ell] = \sup_{\substack{\mu', \nu \in \mathcal{P}(\mathcal{Z}) \\ \pi \in \Pi(\mu', \nu)}} \left\{ \mathbb{E}_\nu[\ell] : \begin{aligned} &\mathbb{E}_\nu[(Z - z_0)(Z - z_0)^\top] \preceq \sigma I_d, \\ &\mathbb{E}_\pi[\|Z' - Z\|^p] \leq \rho^p, \\ &\mu' \leq \frac{1}{1-\varepsilon}\tilde{\mu}_n \end{aligned} \right\}$$

$$= \sup_{\substack{m\in\mathbb{R}^n \\ \nu_1,\dots,\nu_n\in\mathcal{P}(\mathcal{Z})}} \left\{ \sum_{i\in[n]} m_i\, \mathbb{E}_{\nu_i}[\ell] : \begin{array}{l} \sum_{i\in[n]} m_i\, \mathbb{E}_{\nu_i}[(Z_i-z_0)(Z_i-z_0)^\top] \preceq \sigma I_d, \\ \sum_{i\in[n]} m_i\, \mathbb{E}_{\nu_i}[\|\tilde{Z}_i-Z_i\|^p] \leq \rho^p, \\ 0\leq m_i \leq \frac{1}{n(1-\varepsilon)},\ \forall i\in[n], \\ \sum_{i\in[n]} m_i = 1 \end{array} \right\}$$

$$= \sup_{\substack{m\in\mathbb{R}^n \\ \nu_1',\dots,\nu_n'\geq 0}} \left\{ \sum_{i\in[n]} \mathbb{E}_{\nu_i'}[\ell] : \begin{array}{l} \sum_{i\in[n]} \int_{\mathcal{Z}} (z_i-z_0)(z_i-z_0)^\top d\nu_i'(z_i) \preceq \sigma I_d, \\ \sum_{i\in[n]} \int_{\mathcal{Z}} \|z_i-\tilde{Z}_i\|^p d\nu_i'(z_i) \leq \rho^p, \\ 0\leq m_i \leq \frac{1}{n(1-\varepsilon)},\ \forall i\in[n], \\ \sum_{i\in[n]} m_i = 1 \\ \int_{\mathcal{Z}} d\nu_i'(z_i) = m_i,\ \forall i\in[n] \end{array} \right\},$$

where the first equality follows from the definitions of $\mathcal{G}_{\mathrm{cov}}(\sigma,z_0)$ and $\mathsf{W}_p^\varepsilon(\tilde{\mu}_n\|\nu)$. The second and the third equalities follow from the same variable substitution as in the proof of Proposition 6. The last optimization problem admits the dual form

$$\inf_{\substack{\Lambda_1\in\mathbb{Q}_+^d,\lambda_2\in\mathbb{R}_+ \\ r,s\in\mathbb{R}^n,\alpha\in\mathbb{R}}} \left\{ \begin{array}{l} -z_0^\top \Lambda_1 z_0 + \sigma\,\mathrm{Tr}[\Lambda_1] + \lambda_2\rho^p + \alpha + \frac{\sum_{i\in[n]} s_i}{n(1-\varepsilon)} : \\ s_i \geq \max\{0, r_i-\alpha\},\quad \forall i\in[n], \\ r_i \geq \ell(\xi) - \xi^\top\Lambda_1\xi + 2\xi^\top\Lambda_1 z_0 - \lambda_2\|\xi-\tilde{Z}_i\|^p,\quad \forall\xi\in\mathcal{Z},\forall i\in[n] \end{array} \right\}.$$

Strong duality holds thanks to Assumption 3 and [47, Proposition 3.4]. The proof of the second claim concludes by removing the decision variables $r$ and $s$ and using the definition of $\tilde{\mu}_n$. $\qquad\square$

### C.7 Proof of Theorem 5

By Proposition 10 and exploiting the definition of $\tilde{\mu}_n$, we have

$$\sup_{\substack{\nu\in\mathcal{G}_{\mathrm{cov}}(\sigma,z_0): \\ \mathsf{W}_p^\varepsilon(\tilde{\mu}_n\|\nu)\leq\rho}} \mathbb{E}_\nu[\ell]$$

$$= \left\{ \begin{array}{ll} \inf & -z_0^\top\Lambda_1 z_0 + \sigma\,\mathrm{Tr}[\Lambda_1] + \lambda_2\rho^p + \alpha + \dfrac{1}{n(1-\varepsilon)}\sum_{i\in[n]} s_i \\[2mm] \mathrm{s.t.} & \Lambda_1\in\mathbb{Q}_+^d,\ \lambda_2\in\mathbb{R}_+,\ s\in\mathbb{R}_+^n \\[2mm] & s_i \geq \displaystyle\sup_{\xi\in\mathcal{Z}} \ell(\xi) - \xi^\top\Lambda_1\xi + 2\xi^\top\Lambda_1 z_0 - \lambda_2\|\xi-\tilde{Z}_i\|^p - \alpha \quad \forall i\in[n] \end{array} \right.$$

$$= \left\{ \begin{array}{ll} \inf & -z_0^\top\Lambda_1 z_0 + \sigma\,\mathrm{Tr}[\Lambda_1] + \lambda_2\rho^p + \dfrac{1}{n(1-\varepsilon)}\sum_{i\in[n]} s_i \\[2mm] \mathrm{s.t.} & \Lambda_1\in\mathbb{Q}_+^d,\ \lambda_2\in\mathbb{R}_+,\ s\in\mathbb{R}_+^n \\[2mm] & s_i \geq \displaystyle\sup_{\xi\in\mathcal{Z}} \ell_j(\xi) - \xi^\top\Lambda_1\xi + 2\xi^\top\Lambda_1 z_0 - \lambda_2\|\xi-\tilde{Z}_i\|^p - \alpha \quad \forall i\in[n],\forall j\in[J] \end{array} \right. \tag{10}$$

where the second equality follows form Assumption 2. For any fixed $i\in[n]$ and $j\in[J]$, we have

$$\sup_{\xi\in\mathcal{Z}} \ell_j(\xi) - \xi^\top\Lambda_1\xi + 2\xi^\top\Lambda_1 z_0 - \lambda_2\|\xi-\tilde{Z}_i\|^p - \alpha$$

$$= \left\{ \begin{array}{ll} \inf & (-\ell_j)^*(\zeta_{ij}^\ell) + \frac{1}{4}(\zeta_{ij}^\mathcal{G})^\top\Lambda_1^{-1}\zeta_{ij}^\mathcal{G} + \tilde{Z}_i^\top\zeta_{ij}^\mathsf{W} + \lambda_2 h(\zeta_{ij}^\mathsf{W}/\lambda_2,p) + \chi_\mathcal{Z}^*(\zeta_{ij}^\mathcal{Z}) - \alpha \\[1mm] \mathrm{s.t.} & \zeta_{ij}^\ell,\zeta_{ij}^\mathcal{G},\zeta_{ij}^\mathsf{W},\zeta_{ij}^\mathcal{Z}\in\mathbb{R}^d,\ \zeta_{ij}^\ell+\zeta_{ij}^\mathcal{G}+\zeta_{ij}^\mathsf{W}+\zeta_{ij}^\mathcal{Z} = 2\Lambda_1 z_0 \end{array} \right.$$

where the equality is a result of strong duality due to [57, Theorem 2] and Lemma 11. The claim follows by introducing the epigraph variable $\tau_{ij}$ for the term $\frac{1}{4}(\zeta_{ij}^\mathcal{G})^\top\Lambda_1^{-1}\zeta_{ij}^\mathcal{G}$, and substituting all resulting dual minimization problems into (10) and eliminating the corresponding minimization operators. Note that by the Schur complement argument, we have

$$\frac{1}{4}(\zeta_{ij}^\mathcal{G})^\top\Lambda_1^{-1}\zeta_{ij}^\mathcal{G} \leq \tau_{ij} \iff \begin{bmatrix} \Lambda_1 & \zeta_{ij}^\mathcal{G} \\ (\zeta_{ij}^\mathcal{G})^\top & 4\tau_{ij} \end{bmatrix} \succeq 0,$$

which implies that the resulting reformulation is indeed convex. $\qquad\square$

# D Comparison to WDRO with Expanded Radius around Minimum Distance Estimate (Remark 1)

First, we prove the claimed excess risk bound for WDRO with an expanded radius around the minimum distance estimate $\hat{\mu} = \hat{\mu}(\tilde{\mu}, \mathcal{G}, \varepsilon) := \operatorname{argmin}_{\nu \in \mathcal{G}} \mathsf{W}_p^\varepsilon(\nu, \tilde{\mu})$. We write $c = 2(1 - \varepsilon)^{-1/p}$ as in Theorem 1.

**Lemma 14.** *Under Setting A, let* $\hat{\ell} = \operatorname{argmin}_{\ell \in \mathcal{L}} \sup_{\nu \in \mathcal{P}(\mathcal{Z}): \mathsf{W}_p(\nu, \hat{\mu}) \leq \rho'} \mathbb{E}_\nu[\ell]$, *for the expanded radius* $\rho' := c\rho + 2\tau_p(\mathcal{G}, 2\varepsilon)$. *We then have* $\mathbb{E}_\mu[\hat{\ell}] - \mathbb{E}_\mu[\ell_\star] \leq \mathcal{R}_{\mu,p}(c\rho + 2\tau_p(\mathcal{G}, 2\varepsilon); \ell_\star)$.

*Proof.* Since $\mathsf{W}_p^\varepsilon(\mu, \tilde{\mu}) \leq \rho$ and $\mu \in \mathcal{G}$, we have $\mathsf{W}_p^\varepsilon(\hat{\mu}, \tilde{\mu}) \leq \rho$. Thus, Lemma 2 gives that $\mathsf{W}_p^{2\varepsilon}(\hat{\mu}, \mu) \leq 2\rho$. By Lemma 3, we then have $\mathsf{W}_p(\hat{\mu}, \mu) \leq c\rho + 2\tau_p(\mathcal{G}, 2\varepsilon)$, and so Lemma 7 gives the desired result. □

In practice, we are unaware of efficient finite-sample algorithms to compute $\hat{\mu}$. For the class $\mathcal{G}_{\text{cov}}$, we instead propose the spectral reweighing estimate $\check{\mu} = \check{\mu}(\tilde{\mu}, \varepsilon) := \operatorname{argmin}_{\nu \in \mathcal{P}_2(\mathcal{Z}), \nu \leq \frac{1}{1-2\varepsilon}\tilde{\mu}} \|\Sigma_\nu\|_{\text{op}}$, where $\| \cdot \|_{\text{op}}$ is the matrix operator norm (see [26] for varied applications of spectral reweighing). In practice, when $\tilde{\mu} = \tilde{\mu}_n$ is an $n$-sample empirical measure, one can efficiently obtain a feasible measure $\nu$ whose objective value is optimal up to constant factors for the problem with $\varepsilon \leftarrow 3\varepsilon$, using the iterative filtering algorithm [15]. We work with the exact minimizer $\check{\mu}$ for convenience, but our results are robust to such approximate solutions.

**Lemma 15.** *Under Setting A with* $\mathcal{G} = \mathcal{G}_{\text{cov}}$, $p = 1$, *and* $0 < \varepsilon \leq 0.2$, *we have* $\mathsf{W}_1(\check{\mu}, \mu) \lesssim \sqrt{d}\rho + \sqrt{d\varepsilon}$, *and this bound is tight; that is, there exists an instance* $(\mu, \tilde{\mu}) \in \mathcal{G}_{\text{cov}} \times \mathcal{P}(\mathbb{R}^d)$ *with* $\mathsf{W}_1^\varepsilon(\mu, \tilde{\mu}) \leq \rho$ *such that* $\mathsf{W}_1(\check{\mu}, \mu) \gtrsim \sqrt{d}\rho + \sqrt{d\varepsilon}$. *Consequently, the WDRO estimate* $\check{\ell} = \operatorname{argmin}_{\ell \in \mathcal{L}} \sup_{\nu \in \mathcal{P}(\mathcal{Z}): \mathsf{W}_1(\nu, \check{\mu}) \leq \check{\rho}} \mathbb{E}_\nu[\ell]$ *satisfies* $\mathbb{E}_\mu[\check{\ell}] - \mathbb{E}_\mu[\ell_\star] \leq \mathcal{R}_{\mu,1}(O(\sqrt{d}\rho + \sqrt{d\varepsilon}); \ell_\star)$.

*Proof.* **Upper bound:** For the upper bound on $\mathsf{W}_1$ estimation error, fix any $\tilde{\mu} \in \mathcal{P}(\mathbb{R}^d)$ with $\mathsf{W}_1^\varepsilon(\tilde{\mu}, \mu) \leq \rho$. Take any $\mu' \in \mathcal{P}(\mathbb{R}^d)$ such that $\mathsf{W}_1(\mu', \mu) \leq \rho$ and $\|\mu' - \tilde{\mu}\|_{\text{TV}} \leq \varepsilon$. By Lemma 13, there exists $\alpha \in \mathcal{P}(\mathbb{R}^d)$ with $\mathsf{W}_1(\alpha, \mu) \leq \rho$, $\mathsf{W}_2(\alpha, \mu) \leq \rho\sqrt{2/\varepsilon}$, and $\|\alpha - \tilde{\mu}\|_{\text{TV}} \leq 2\varepsilon$. Fixing an optimal coupling $\pi \in \Pi(\alpha, \mu)$ for the $\mathsf{W}_2(\alpha, \mu)$ problem and letting $(Z, W) \sim \pi$, we bound

$$\|\Sigma_\alpha\|_{\text{op}}^{\frac{1}{2}} = \sup_{\theta \in \mathbb{S}^{d-1}} \mathbb{E}[\theta^\top(Z - \mathbb{E}[Z])^2]^{\frac{1}{2}} \leq \sup_{\theta \in \mathbb{S}^{d-1}} \mathbb{E}[\theta^\top(Z - \mathbb{E}[W])^2]^{\frac{1}{2}} \leq \|\Sigma_\mu\|_{\text{op}}^{\frac{1}{2}} + \mathsf{W}_2(\alpha, \mu)$$

Thus, $\alpha \in \mathcal{G}_{\text{cov}}(1 + \rho\sqrt{2/\varepsilon})$. Write $\beta := \frac{1}{(\alpha \wedge \tilde{\mu})(\mathbb{R}^d)}\alpha \wedge \tilde{\mu}$, and note that this midpoint measure is feasible for the problem defining $\check{\mu}$. Hence, we have

$$\|\Sigma_{\check{\mu}}\|_{\text{op}} \leq \|\Sigma_\beta\|_{\text{op}} \leq \sup_{\theta \in \mathbb{S}^{d-1}} \mathbb{E}_\beta[(\theta^\top(Z - \mathbb{E}_\alpha[Z]))^2] \leq \frac{1}{1-2\varepsilon}\|\Sigma_\alpha\|_{\text{op}} \leq \frac{1}{1-2\varepsilon}(1 + \rho\sqrt{2/\varepsilon})^2,$$

and so $\check{\mu} \in \mathcal{G}_{\text{cov}}((1 - 2\varepsilon)^{-1/2}(1 + \rho\sqrt{2/\varepsilon}))$. Moreover, we have $\|\check{\mu} - \alpha\|_{\text{TV}} \leq 4\varepsilon$. Thus, using Lemma 5 and the fact that $4\varepsilon$ is bounded away from 1, we bound

$$\mathsf{W}_1(\check{\mu}, \alpha) \leq \mathsf{W}_1\left(\check{\mu}, \frac{1}{(\check{\mu} \wedge \alpha)(\mathbb{R}^d)}\check{\mu} \wedge \alpha\right) + \mathsf{W}_1\left(\frac{1}{(\check{\mu} \wedge \alpha)(\mathbb{R}^d)}\check{\mu} \wedge \alpha\right) \lesssim \sqrt{d}\rho + \sqrt{d\varepsilon}.$$

By the triangle inequality, we have $\mathsf{W}_1(\check{\mu}, \mu) \lesssim \sqrt{d}\rho + \sqrt{d\varepsilon}$. The risk bound follows by Lemma 7. Taking the final error measurement using distance between means instead of $\mathsf{W}_1$, we observe that $\|\mathbb{E}_{\check{\mu}}[Z] - \mathbb{E}_\mu[Z]\|_2 \lesssim \rho + \sqrt{\varepsilon}$.

**Lower bound:** To see that this guarantee cannot be improved, fix clean measure $\mu = \delta_0 \in \mathcal{G}_{\text{cov}}$, and consider the corrupted measure $\tilde{\mu} = (1 - 3\varepsilon)\delta_0 + 2\varepsilon\left(\frac{1}{2}\delta_{\frac{\rho}{2\varepsilon}\mathbf{e}_1} + \frac{1}{2}\delta_{-\frac{\rho}{2\varepsilon}\mathbf{e}_1}\right) + \varepsilon\mathcal{N}\left(0, \frac{\rho^2}{100\varepsilon^2}I_d\right)$, constructed so that $\mathsf{W}_1^\varepsilon(\tilde{\mu}, \mu) \leq \rho$. Intuitively, iterative filtering seeks to drive down the operator norm of covariance matrix and will thus focus on removing mass from the second mixture component.

To formalize this, we decompose the output of filtering as $(1 - 2\varepsilon)\check{\mu} = (1 - 3\varepsilon - \tau)\delta_0 + \alpha + \beta$, where $0 \leq \tau \leq 2\varepsilon$ and $\alpha, \beta \in \mathcal{M}_+(\mathbb{R}^d)$ such that $\alpha \leq 2\varepsilon\left(\frac{1}{2}\delta_{\frac{\rho}{2\varepsilon}\mathbf{e}_1} + \frac{1}{2}\delta_{-\frac{\rho}{2\varepsilon}\mathbf{e}_1}\right)$ and $\beta \leq \varepsilon\mathcal{N}\left(0, \frac{\rho^2}{100\varepsilon^2}I_d\right)$. We have $\tau + (2\varepsilon - \alpha(\mathbb{R}^d)) + (\varepsilon - \beta(\mathbb{R}^d)) = 2\varepsilon$ by the definition of $\check{\mu}$. Further note that $\|\mathbb{E}_{\check{\mu}}[Z]\| \lesssim$

$\sqrt{\varepsilon} + \rho \lesssim \rho$, by the bounds above. Now suppose for sake of contradiction that $\beta(\mathbb{R}^d) \le \varepsilon/2$. Then $\alpha(\mathbb{R}^d) \ge \varepsilon/2$, and so we bound

$$
\begin{aligned}
\|\Sigma_{\check{\mu}}\|_{\mathrm{op}}^{\frac{1}{2}} &\ge \sqrt{\mathbb{E}_{\check{\mu}}[Z_1^2]} - \|\mathbb{E}_{\check{\mu}}[Z]\| \\
&\ge \sqrt{\frac{\varepsilon}{2(1-2\varepsilon)}\frac{\rho^2}{4\varepsilon^2}} - \|\mathbb{E}_{\check{\mu}}[Z]\| \\
&\ge \frac{\rho}{\sqrt{8\varepsilon}} - O(\rho).
\end{aligned}
$$

On the other hand, another feasible outcome for spectral reweighing is $\mu' = \frac{1}{1-\varepsilon}(1-3\varepsilon)\delta_0 + \varepsilon\mathcal{N}\big(0,\frac{\rho^2}{100\varepsilon^2}I_d\big)$, for which we have $\|\Sigma_{\mu'}\|_{\mathrm{op}}^{1/2} = \frac{\rho}{10\sqrt{\varepsilon}}$. Since $10 > \sqrt{8}$, this contradicts optimality of $\check{\mu}$ if $\varepsilon \le c\rho^2$ for a sufficiently small constant $c$. However, if $\varepsilon > c\rho^2$, then a lower bound of $\Omega(\sqrt{d\varepsilon})$ suffices. This bound holds even without Wasserstein perturbations; see the $\mathcal{G}_{\mathrm{cov}}$ lower risk bound of $\Omega(\sqrt{d\varepsilon})$ in Theorem 2 of [38].

We now suppose that $\beta(\mathbb{R}^d) \ge \varepsilon/2$. Let $Z \sim \mathcal{N}\big(0, \frac{\rho^2}{100\varepsilon^2}\big)$ and write $F$ for the CDF of $\|Z\|^2$ (which has a scaled $\chi_d^2$ distribution). We then have $\int \|z\|\,\mathrm{d}\beta(z) \ge \varepsilon\,\mathbb{E}\big[\|Z\| \mid \|Z\|^2 \le F^{-1}(1/2)\big] \gtrsim \sqrt{d}\rho$, using concentration of $\chi_d^2$ about its mean. Thus, $W_1(\check{\mu}, \mu) \ge \mathbb{E}_{\check{\mu}}[\|Z\|] - \mathbb{E}_{\mu}[\|Z\|] \gtrsim \sqrt{d}\rho$. $\qquad\square$

# E  Smaller Robustness Radius for Outlier-Robust WDRO (Remark 2)

In the classical WDRO setting with $\rho_0 = \varepsilon = 0$, the radius $\rho$ can often be taken significantly smaller than $n^{-1/d}$ if $\mathcal{L}$ and $\mu$ are sufficiently well-behaved. In particular, when $\mu$ satisfies a $T_2$ transportation inequality, [18] proves that $\rho = \widetilde{O}(n^{-1/2})$ gives meaningful risk bounds. Recall that $\mu \in T_2(\tau)$ if

$$
W_2(\nu, \mu) \le \sqrt{\tau\mathsf{H}(\nu\|\mu)}, \quad \forall\nu \in \mathcal{P}_2(\mathcal{Z}),
$$

where $\mathsf{H}(\nu\|\mu) \coloneqq \int_{\mathcal{Z}}\log(d\nu/d\mu)d\nu$ denotes relative entropy when $\nu \ll \mu$ (and is $+\infty$ otherwise). We note that $T_2$ is implied by the log-Sobolev inequality, which holds for example when $\mu$ has strongly log-concave density. Under $T_2$, [18] shows the following.

**Proposition 11** (Example 3 in [18]). *Fix $\mathcal{Z} = \mathbb{R}^d \times \mathbb{R}$, $\tau, B > 0$, and an $\alpha$-smooth and $L$-Lipschitz function $f : \mathbb{R} \to \mathbb{R}$. Consider the parameterized family of loss functions $\mathcal{L} = \{(x,y) \mapsto \ell_\theta(x,y) = f(\theta^\top x - y) : \theta \in \Theta\}$, where $\Theta \subset \{\theta \in \mathbb{R}^d : \|\theta\| \le B\}$. Fix $\mu \in \mathcal{P}(\mathcal{Z})$ whose first marginal $\mu_X = \mu(\cdot \times \mathbb{R})$ satisfies $\mu_X \in T_2(\tau)$ and such that $\inf_{\theta\in\Theta}\mathbb{E}_\mu[f'(\theta^\top X - Y)^2] > 0$. Write*

$$
\sigma = \sup_{\theta\in\Theta}\frac{\mathbb{E}_\mu[f'(\theta^\top X, Y)^4]^{\frac{1}{2}}}{\mathbb{E}_\mu[f'(\theta^\top X, Y)^2]} \le \frac{L^2}{\inf_{\theta\in\Theta}\mathbb{E}_\mu[f'(\theta^\top X, Y)^2]} < \infty.
$$

*For $t > 0$, define*

$$
\rho_n = \sqrt{\frac{\tau t\big(1 + d\log(2 + 2Bn)\big)}{n}}\left(1 + \sigma\sqrt{\frac{2t\big(1 + d\log(2 + 2Bn)\big)}{n}}\right),
$$

$$
\delta_n = \frac{2L + 2B\alpha\,\mathbb{E}_\mu[\|X\|] + B^2\alpha^2\,\mathrm{Var}_\mu(\|X\|) + \rho_n\sqrt{\mathbb{E}_\mu\big[(L + B\alpha\|X\|)^2\big] + \mathrm{Var}_\mu\big((L + B\alpha\|X\|)^2\big)}}{n},
$$

$$
\eta_n = \frac{2\alpha B^2\tau t\big(1 + d\log(2 + 2Bn)\big)}{n}.
$$

*Then, with probability at least $1 - 2/n - 2e^{-t}$, we have*

$$
|\mathbb{E}_\mu[\ell_\theta] - \mathbb{E}_{\hat{\mu}_n}[\ell_\theta]| \le \mathcal{R}_{\hat{\mu}_n, 2}(\rho_n; \ell_\theta) + \delta_n + \eta_n \quad \forall\theta \in \Theta. \tag{11}
$$

We note that (11) is stated without the absolute value on the right hand side, but that this strengthened result holds due to the discussion after [18, Theorem 1]. This generalization bound immediately gives the following excess risk bound.

**Corollary 5.** *Assume $n \geq 800$. Fix $\rho_n$, $\delta_n$, $\eta_n$, and $\mathcal{L}$ as in Proposition 11 with $t = 7$, and take $\ell_{\hat{\theta}} \in \mathcal{L}$ minimizing (1) with $p = 2$ and radius $\rho = \rho_n$. Then, with probability at least 0.99, we have*

$$\mathbb{E}_\mu[\ell_{\hat{\theta}}] - \mathbb{E}_\mu[\ell_\theta] \lesssim \rho_n \|\ell_\theta\|_{\dot{H}^{1,2}(\mu)} + \alpha \rho_n^2 + \delta_n + \eta_n \quad \forall \theta \in \Theta.$$

*Proof.* By Proposition 11 and [18, Remark 1], we have

$$\mathbb{E}_\mu[\ell_{\hat{\theta}}] - \mathbb{E}_\mu[\ell_\theta] \leq 2\mathcal{R}_{\hat{\mu}_n,2}(\rho_n; \ell_\theta) + 2\delta_n + 2\eta_n \quad \forall \theta \in \Theta,$$

with probability at least $1 - 2/n - 2e^{-t} \geq 0.995$. Since $\ell_\theta$ is $\alpha$-smooth, Lemma 6 gives that

$$\mathbb{E}_\mu[\ell_{\hat{\theta}}] - \mathbb{E}_\mu[\ell_\theta] \leq 2\rho_n \|\ell_\theta\|_{\dot{H}^{1,2}(\hat{\mu}_n)} + 2\alpha \rho_n^2 + 2\delta_n + 2\eta_n \quad \forall \theta \in \Theta,$$

with probability at least 0.995. By Markov's inequality, we can substitute $\hat{\mu}_n$ with $\mu$ at the cost of a constant factor blow-up in excess risk and a decrease in the confidence probability to, say, 0.99. □

In the example above, excess risk is controlled by the 2-Wasserstein regularizer with radius $\rho_n = O(n^{-1/2})$, up to $O(n^{-1})$ correction terms, which is significantly smaller that the typical radius size of $O(n^{-1/d})$. We shall now lift this improvement to the outlier-robust setting. Similar to Proposition 4, we perform outlier-robust DRO with a modified choice of $\mathcal{A}$. This time, writing $\mathcal{G}_2(\sigma) = \cup_{z_0 \in \mathcal{Z}} \mathcal{G}_2(\sigma, z_0)$, we have the following.

**Proposition 12** (Outlier-robust WDRO under $T_2$). *Assume $n \geq 800$ and $\mu \in \mathcal{G}_{\mathrm{cov}}$. Fix $\rho_n$, $\delta_n$, $\eta_n$, and $\mathcal{L}$ as in Proposition 11 with $t = 8$, and take $\ell_{\hat{\theta}}$ minimizing (OR-WDRO) with center $\tilde{\mu}_n$, radius $\rho = \rho_0 + 15\rho_n + 200\sqrt{d}$, and $\mathcal{A} = \mathcal{G}_2(15\sqrt{d} + \rho_n)$. Then, with probability at least 0.99, we have*

$$\mathbb{E}_\mu[\ell_{\hat{\theta}}] - \mathbb{E}_\mu[\ell_\theta] \lesssim \|\ell_\theta\|_{\dot{H}^{1,2}(\mu)}(\rho_0 + \rho_n + \sqrt{d}) + \alpha(\rho_0 + \rho_n + \sqrt{d})^2 + \delta_n + \eta_n \quad \forall \theta \in \Theta.$$

*Proof.* Noting that $\mathcal{G}_{\mathrm{cov}} \subseteq \mathcal{G}_2(\sqrt{d})$, we have by Markov's inequality that $\hat{\mu}_n \in \mathcal{G}_2(15\sqrt{d})$ with probability at least 0.995. In other words, there exists $z_0 \in \mathcal{Z}$ such that $\mathsf{W}_2(\hat{\mu}_n, \delta_{z_0}) \leq 15\sqrt{d}$. Thus, for any $\nu \in \mathcal{P}(\mathcal{Z})$ with $\mathsf{W}_2(\hat{\mu}_n, \nu) \leq \rho_n$, we have $\mathsf{W}_2(\nu, \delta_{z_0}) \leq 15\sqrt{d} + \rho_n$, and so $\nu \in \mathcal{G}_2(15\sqrt{d} + \rho_n)$. By Lemmas 4 and 5, this implies that

$$\begin{aligned}
\mathsf{W}_2^\varepsilon(\tilde{\mu}_n \| \nu) &\leq \mathsf{W}_2^\varepsilon(\tilde{\mu}_n, \nu) + \tau_2(\nu, \varepsilon) \\
&\leq \rho_0 + \rho_n + 8(15\sqrt{d} + \rho_n)(1 - \varepsilon)^{-1/2} \\
&< \rho.
\end{aligned}$$

Next, by Proposition 11, with probability at least $1 - 1/400 + 2e^{-8} \geq 0.9985$, we have for each $\theta \in \Theta$ that

$$\mathbb{E}_\mu[\ell_{\hat{\theta}}] - E_\mu[\ell_\theta] \leq \mathbb{E}_{\hat{\mu}_n}[\ell_{\hat{\theta}}] + \mathcal{R}_{\hat{\mu}_n,2}(\rho_n; \ell_{\hat{\theta}}) - \mathbb{E}_\mu[\ell_\theta] + \delta_n + \eta_n$$

$$= \mathbb{E}_{\hat{\mu}_n}[\ell_{\hat{\theta}}] + \left( \sup_{\substack{\nu \in \mathcal{P}(\mathcal{Z}) \\ \mathsf{W}_2(\hat{\mu}_n, \nu) \leq \rho_n}} \mathbb{E}_\nu[\ell_{\hat{\theta}}] - \mathbb{E}_{\hat{\mu}_n}[\ell_{\hat{\theta}}] \right) - \mathbb{E}_\mu[\ell_\theta] + \delta_n + \eta_n$$

$$= \mathbb{E}_{\hat{\mu}_n}[\ell_{\hat{\theta}}] + \left( \sup_{\substack{\nu \in \mathcal{G}_2(15\sqrt{d}+\rho_n) \\ \mathsf{W}_2(\hat{\mu}_n, \nu) \leq \rho_n}} \mathbb{E}_\nu[\ell_{\hat{\theta}}] - \mathbb{E}_{\hat{\mu}_n}[\ell_{\hat{\theta}}] \right) - \mathbb{E}_\mu[\ell_\theta] + \delta_n + \eta_n$$

$$\leq \mathbb{E}_{\hat{\mu}_n}[\ell_{\hat{\theta}}] + \left( \sup_{\substack{\nu \in \mathcal{G}_2(15\sqrt{d}+\rho_n) \\ \mathsf{W}_2^\varepsilon(\tilde{\mu}_n \| \nu) \leq \rho}} \mathbb{E}_\nu[\ell_{\hat{\theta}}] - \mathbb{E}_{\hat{\mu}_n}[\ell_{\hat{\theta}}] \right) - \mathbb{E}_\mu[\ell_\theta] + \delta_n + \eta_n.$$

Let $c = 2\left(\frac{1-\varepsilon}{1-2\varepsilon}\right)^{1/p}$. Using optimality of $\hat{\ell}$ and Lemma 2, we further bound

$$\mathbb{E}_\mu[\ell_{\hat{\theta}}] - E_\mu[\ell_\theta] \leq \left( \sup_{\substack{\nu \in \mathcal{G}_2(15\sqrt{d}+\rho_n) \\ \mathsf{W}_2^\varepsilon(\tilde{\mu}_n \| \nu) \leq \rho}} \mathbb{E}_\nu[\ell_\theta] - \mathbb{E}_{\hat{\mu}_n}[\ell_\theta] \right) + \mathbb{E}_{\hat{\mu}_n}[\ell_\theta] - \mathbb{E}_\mu[\ell_\theta] + \delta_n + \eta_n$$

$$\leq \left( \sup_{\substack{\nu \in \mathcal{G}_2(15\sqrt{d}+\rho_n) \\ \mathsf{W}_2^\varepsilon(\tilde{\mu}_n \| \nu) \leq \rho}} \mathbb{E}_\nu[\ell_\theta] - \mathbb{E}_{\hat{\mu}_n}[\ell_\theta] \right) + \mathcal{R}_{\hat{\mu}_n, 2}(\rho_n; \ell_\theta) + 2\delta_n + 2\eta_n$$

$$\leq \left( \sup_{\substack{\nu \in \mathcal{G}_2(15\sqrt{d}+\rho_n) \\ \mathsf{W}_2^{2\varepsilon}(\hat{\mu}_n, \nu) \leq c\rho}} \mathbb{E}_\nu[\ell_\theta] - \mathbb{E}_{\hat{\mu}_n}[\ell_\theta] \right) + \mathcal{R}_{\hat{\mu}_n, 2}(\rho_n; \ell_\theta) + 2\delta_n + 2\eta_n$$

$$\leq \left( \sup_{\substack{\nu \in \mathcal{G}_2(15\sqrt{d}+\rho_n) \\ \mathsf{W}_2(\nu, \hat{\mu}_n) \leq c\rho + \tau_2(\hat{\mu}_n, 2\varepsilon) + \tau_2(\mathcal{A}, 2\varepsilon)}} \mathbb{E}_\nu[\ell_\theta] - \mathbb{E}_{\hat{\mu}_n}[\ell_\theta] \right) + \mathcal{R}_{\hat{\mu}_n, 2}(\rho_n; \ell_\theta) + 2\delta_n + 2\eta_n$$

$$\leq \mathcal{R}_{\hat{\mu}_n 2}\big(c\rho + \tau_2(\hat{\mu}_n, 2\varepsilon) + \tau_2(\mathcal{A}, 2\varepsilon); \ell_\theta\big) + \mathcal{R}_{\hat{\mu}_n, 2}(\rho_n; \ell_\theta) + 2\delta_n + 2\eta_n$$

$$\lesssim \|\ell_\theta\|_{\dot{H}^{1,2}(\hat{\mu}_n)}\big(\rho_0 + \rho_n + \sqrt{d}\big) + \alpha\big(\rho_0 + \rho_n + \sqrt{d}\big)^2 + \delta_n + \gamma_n.$$

As in Corollary 5, we can substitute $\hat{\mu}_n$ with $\mu$ at the cost of a constant factor increase in excess risk along with a small decrease in the confidence probability (in this case, sufficiently small such that the total failure probability is at most 0.01). $\qquad\square$

The main goal of Proposition 12 was to demonstrate that one can expect improved excess risk bounds for outlier-robust WDRO in situations where such improvements hold for standard WDRO. We conjecture that similar guarantees hold for additional settings and under milder assumptions like the $T_1$ inequality, but leave such refinements for future work. In particular, for the class $\mathcal{G}_{\mathrm{cov}}$, it would be desirable to prove such bounds when $p = 1$, so that the TV contribution to the risk vanishes as $\varepsilon \to 0$.

## F Parameter Tuning (Remark 5)

To clarify the parameter selection process, we consider Setting B with the class $\mathcal{G} = \mathcal{G}_{\mathrm{cov}}(\sigma)$, $p = 1$, and $\varepsilon \leq 1/3$. We aim to efficiently achieve excess risk

$$\mathbb{E}_\mu[\hat{\ell}] - \mathbb{E}_\mu[\ell_\star] \lesssim \|\ell_\star\|_{\mathrm{Lip}}\big(\rho_0 + \sigma\sqrt{d}\varepsilon + \sigma\sqrt{d}n^{-1/d}\big), \tag{12}$$

matching Proposition 4, when

$$\hat{\ell} = \operatorname{argmin}_{\ell \in \mathcal{L}} \sup_{\nu \in \mathcal{G}_2(\hat{\sigma}, z_0) : \mathsf{W}_1^{\hat{\varepsilon}}(\tilde{\mu}_n, \nu) \leq \hat{\rho}} \mathbb{E}_\nu[\ell] \tag{13}$$

for some parameter guesses $\hat{\sigma}$, $\hat{\varepsilon}$, and $\hat{\rho}$, and a robust mean estimate $z_0$. First, we observe that the coordinate-wise trimmed mean estimate from Proposition 5 is computed without knowledge of the parameters, so it is safe to assume that $\|z_0 - \mathbb{E}_\mu[Z]\| \lesssim \sqrt{d} + \rho_0$. If the parameter guesses are conservative, i.e., $\hat{\sigma} \geq \sigma$, $\hat{\varepsilon} \geq \varepsilon$, and $\hat{\rho} \geq \rho_0 + \mathsf{W}_1(\mu, \hat{\mu}_n)$, then we may still employ Proposition 4. If they are not too large, i.e., $\hat{\sigma} \lesssim \sigma$, $\hat{\varepsilon} \lesssim \varepsilon$, and $\hat{\rho} \leq \rho_0 + \sigma\sqrt{d}n^{-1/d}$, this gives the desired excess risk.

We now explore what prior knowledge of $\sigma$, $\varepsilon$, and $\rho_0$ is needed to obtain such guesses. First, we show that effective learning is impossible without knowledge of $\rho_0$, even for standard WDRO (i.e., $\varepsilon = 0$, $\sigma = \infty$). For ease of presentation, we present the following lower bound without sampling.

**Lemma 16.** *There exists a family of loss functions $\mathcal{L}$ over $\mathbb{R}$ such that, for any $C > 0$ and decision rule $\mathsf{D} : \mathcal{P}(\mathbb{R}) \to \mathcal{L}$, there are $\mu, \tilde{\mu} \in \mathcal{P}(\mathbb{R})$ such that $\mathbb{E}_\mu[\mathsf{D}(\tilde{\mu})] > C(\inf_{\ell \in \mathcal{L}} \mathbb{E}_\mu[\ell] + \mathsf{W}_1(\mu, \tilde{\mu})\|\ell\|_{\mathrm{Lip}})$.*

*Proof.* Let $\mathcal{L} = \{\ell_\theta : \theta > 0\}$, where $\ell_\theta(z) := z/\theta + \theta$. By design, we have $\|\ell_\theta\|_{\mathrm{Lip}} = 1/\theta$. Let $\tilde{\mu} = \delta_0$ and write $\mathsf{D}(\tilde{\mu}) = \ell_{\hat{\theta}}$ for some $\hat{\theta} > 0$. We then set $\mu = \delta_\rho$ for $\rho = 10\hat{\theta}^2 C^2$. This gives

$$\mathbb{E}_\mu[\mathsf{D}(\tilde{\mu})] = \ell_{\hat{\theta}}(\rho) = \frac{\rho}{\hat{\theta}} + \hat{\theta} > \frac{\rho}{\hat{\theta}} = 10C^2\hat{\theta},$$

and

$$\inf_{\ell \in \mathcal{L}} \mathbb{E}_\mu[\ell] + \mathsf{W}_1(\mu, \tilde{\mu})\|\ell\|_{\mathrm{Lip}} = \inf_{\theta > 0} \ell_\theta(\rho) + \frac{\rho}{\theta} = \inf_{\theta > 0} \frac{2\rho}{\theta} + \theta = 2\sqrt{2\rho} = \sqrt{80}C\hat{\theta}.$$

Thus, we have $\mathbb{E}_\mu[\mathsf{D}(\tilde{\mu})] > \inf_{\ell \in \mathcal{L}} \mathbb{E}_\mu[\ell] + \mathsf{W}_1(\mu, \tilde{\mu})\|\ell\|_{\mathrm{Lip}}$, as desired. □

Thus, we assume in what follows that $\hat{\rho} = \rho_0$ is known. Moreover, we require knowledge of at least one of $\varepsilon$ and $\sigma$. If both are unknown, then is information theoretically impossible to meaningfully distinguish inliers from outliers (see Exercise 1.7b of [16] for a discussion of this issue in the setting of robust mean estimation). If $\varepsilon$ is known, then we can choose $\hat{\sigma}$ as $2^i$ for the smallest $i$ such that the supremum of Eq. (13) is feasible (or, equivalently, such that the associated dual is bounded for some fixed $\ell \in \mathcal{L}$). We can overshoot by at most a factor of two and thus achieve the desired risk bound. Using binary search, this adds a multiplicative overhead logarithmic in the ratio of the initial guess for $\sigma$ and its true value. The same approach can be employed if $\sigma$ is known but not $\varepsilon$.

## G  Additional Experiments

We now provide several experiments in addition to those in the main body. Code is again provided at `https://github.com/sbnietert/outlier-robust-WDRO`.

First, in Fig. 3, we extend the experiment summarized Fig. 2 (top) to include runs with $\hat{\varepsilon} = \varepsilon$ and varied Wasserstein radius $\hat{\rho} \in \{\rho/2, \rho, 2\rho\}$. For this simple learning problem and perturbation model, we find that the choice of $\hat{\rho}$ plays a minor role in the resulting excess risk. We emphasize that, in the worst case, selection of $\hat{\rho}$ can be critical, as demonstrated by Lemma 16.

Next, we consider linear classification with the hinge loss, i.e. $\mathcal{L} = \{\ell_\theta(x, y) = \max\{0, 1 - y(\theta^\top x)\} : \theta \in \mathbb{R}^{d-1}\}$. This time (to ensure that the resulting optimization problem is convex), our approach supports Euclidean Wasserstein perturbations in the feature space, but no Wasserstein perturbations in the label space (this corresponds to using $\mathcal{Z} = \mathbb{R}^{d-1} \times \mathbb{R}$ equipped with the (extended) norm $\|(x, y)\| = \|x\|_2 + \infty \cdot \mathbb{1}\{y \neq 0\}$. We consider clean data $(X, Y = \mathrm{sign}(\theta_\star^\top X)) \sim \mu$, where $X \sim \mathcal{N}(0, I_{d-1})$. The corrupted data $(\tilde{X}, \tilde{Y}) \sim \tilde{\mu}$ satisfies $(\tilde{X}, \tilde{Y}) = (X + \rho e_1, Y)$ with probability $1 - \varepsilon$ and $(\tilde{X}, \tilde{Y}) = (100X, -Y)$ with probability $\varepsilon$, so that $\mathsf{W}_p^\varepsilon(\tilde{\mu}\|\mu) \leq \rho$. In Figure 4 (left), we fix $d = 10$ and compare the excess risk $\mathbb{E}_\mu[\ell_{\hat{\theta}}] - \mathbb{E}_\mu[\ell_{\theta_\star}]$ of standard

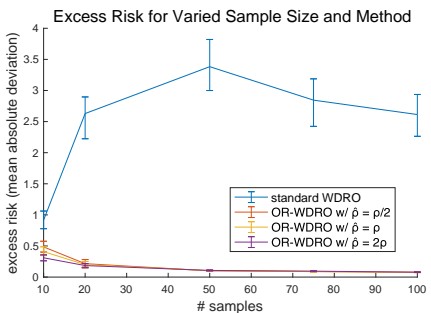

**Figure 3:** Excess risk of standard WDRO and outlier-robust WDRO for linear regression under $\mathsf{W}_p$ and TV corruptions, with varied sample size and dimension.

WDRO and outlier-robust WDRO with $\mathcal{G} = \mathcal{G}_2$, as described by Proposition 4 and implemented via Theorem 2. The results are averaged over $T = 20$ runs for sample size $n \in \{10, 20, 50, 75, 100\}$. We note that this example cannot drive the excess risk of standard WDRO to infinity, so the separation between standard and outlier-robust WDRO is less striking than regression, though still present.

We further present results for multivariate regression. This time, we consider $\mathcal{Z} = \mathbb{R}^{d \times k}$ equipped with the $\ell_2$ norm, and use the loss family $\mathcal{L} = \{\ell_M(x, y) = \|Mx - y\|_1 : M \in \mathbb{R}^{k \times d}\}$. We consider clean data $(X, Y = M_\star^\top X) \sim \mu$, where $M_\star \in \mathbb{R}^{k \times d}$ and $X$ have standard normal entries. The corrupted data $(\tilde{X}, \tilde{Y}) \sim \tilde{\mu}$ satisfies $(\tilde{X}, \tilde{Y}) = (X + \rho e_1, Y)$ with probability $1 - \varepsilon$ and $(\tilde{X}, \tilde{Y}) = (10X, -100M_\star X)$ with probability $\varepsilon$, so that $\mathsf{W}_p^\varepsilon(\tilde{\mu}\|\mu) \leq \rho$. In Figure 4 (right), we fix $d = 10$ and $k = 3$, and compare the excess risk $\mathbb{E}_\mu[\ell_{\hat{M}}] - \mathbb{E}_\mu[\ell_{M_\star}]$ of standard WDRO and outlier-robust WDRO with $\mathcal{G} = \mathcal{G}_2$, as described by Proposition 4 and implemented via Theorem 2. The results are averaged over $T = 10$ runs for sample size $n \in \{10, 20, 50, 75, 100\}$. We are restricted to low $k$ since the $\ell_1$ norm in the losses is expressed as the maximum of $2^k$ concave functions (specifically, we use that $\ell_M(x, y) = \max_{\alpha \in \{-1, 1\}^k} \alpha^\top(Mx - y)$).

Finally, we turn to a classification task with image data. We train a robust linear classifier to distinguish

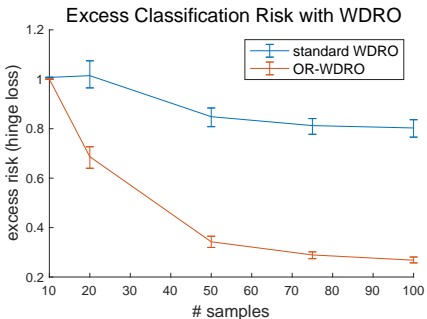
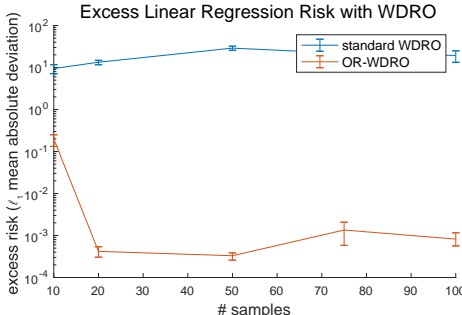

**Figure 4:** Excess risk of standard WDRO and outlier-robust WDRO for classification and multivariate linear regression under $\mathsf{W}_p$ and TV corruptions, with varied sample size.

between the MNIST [14] digits "3" and "8" when 10% of training labels are flipped uniformly at random, using outlier-robust WDRO with $\mathcal{G} = \mathcal{G}_2$ and the hinge loss, as above, applied with $\varepsilon = 0.1$ and $\rho = 0.01$. To ensure tractability, we first pass the images through principal component analysis to reduce the input to 50 dimensions, and we again use Theorem 2 for implementation. In Fig. 5, we plot the classification accuracy of our robust classifier compared to one learned via standard WDRO for training set size $n \in \{10, 20, 50, 100, 150\}$, averaged over $T = 10$ runs. In this case, we do not witness a noticeable improvement over standard WDRO. We suspect that the relevant $\mathcal{G}_2$ class is too large to be of significant use in this high-dimensional setting.

For all experiments, confidence bands are plotted representing the top and bottom 10% quantiles

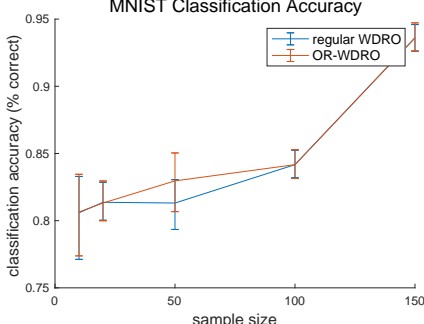

**Figure 5:** Classification accuracy of standard WDRO and outlier-robust WDRO predictors for MNIST digit classification under random label flips, with varied number of training samples.

among 100 bootstrapped means from the $T$ runs. The additional experiments were performed on an M1 Macbook Air with 16GB RAM in roughly 30 minutes each.

