# OpenReview forum: "Outlier-Robust Wasserstein DRO"
_NeurIPS.cc/2023/Conference — NeurIPS 2023 poster_

### Official Review · Reviewer_L4Xq · 2023-07-05

**Soundness:** 3 good
**Presentation:** 4 excellent
**Contribution:** 3 good
**Rating:** 6
**Confidence:** 4

**Summary:**

This paper empowers WDRO with the ability to resist outliers, building upon the outlier-robust Wasserstein distance $W_p^\epsilon$. The excess risk of the solution to both the outlier-robust WDRO and its empirical version is given. An improved bound of the excess risk is derived for the setting of low-dimensional features. The optimization algorithm for the outlier-robust WDRO is developed with the dual form of the minimax problem. Empirical study validates the effectiveness of the proposed method with a simple regression setting involving both Wasserstein and Total Variance contaminiation.

**Strengths:**

1. DRO is known to be over-pessimistic in the existence of outliers. Empowering Distributionally Robust Optimziation with robustness against noisy labels is important. The outlier-robust Wasserstein distance is a well-developed revision of the original Wasserstein distance to account for certain degree of contamination. Thus, constructing the uncertainty set of DRO with $W_p^\epsilon$ is both significant and reasonable.
2. The theoretical analysis of the excess risk of the proposed method is sound and comprehensive. The results quantify how the solution adapts to given geometric and TV contamination. The effectiveness of the empircial algorithm is also guaranteed by the empirical bound of excess risk.

**Weaknesses:**

1. The effectiveness of outlier-robust WDRO is theoretically guaranteed by the excess risk bound. My main concern is over the **advantage** of outlier-robust WDRO v.s. WDRO. As is stated by the authors, the excess risk of outlier-robust WDRO is upper bounded by the $W_p$ regularizer with a larger Wasserstein radius. The authors also recognize that the same upper bound could be achieved by a radius-expanded WDRO. Two advantages of WDRO given in Remark 1 are not convincing.

- The authors claim that a heavy preprocessing step is required by WDRO to estimate the exact radius. However, in practical settings beyond simulated experiments, the exact parameters of contamination levels $\rho, \epsilon$  are also latent. Either an estimate or tuning of the radius is necessary for both WDRO and the outlier-robust version.
- The authors prove an improved bound of the excess risk for outlier-robust WDRO in the setting of low-dimensional features and claim that the bound of WDRO could not be improved, which is unsupported. A tight bound of the excess risk of WDRO with low-dimensional features might be given to consolidate the author's claim. Otherwise, an inequality between the excess risk of WDRO and its outlier-robust version might be given similarly to Eq.4. Furthermore, the proposed low-dimensional feature setting is somewhat impractical because the exact dimension $k$ is typically unknown in real datasets.

In the experiment section, I suppose the radius for WDRO is selected to be the true contaminaton radius. The curve of performances of WDRO and outlier-robust version with increasing radius might demonstrate their gap more convincingly.

2. I am concerned if the formulation of outlier-robust WDRO (eq.3) is well defined. Consider a simple case where all the samples of $\tilde \mu$ are discretely distributed on $2/\epsilon$ points, one of which is denoted by $(x_0,y_0)$. By arbitarily modifying $y_0$ to $y'$ we get a new distribution $\nu(y')$. According to Line 131 we have $W_p^\epsilon(\tilde \mu, \nu) =0$. Thus, all the $\nu(y')$ is included in the uncertainty set of the correspondong $W_p^\epsilon$ DRO. However, the risk of a given predictor on the group of $\nu(y')$ could be unbounded since $y'$ is unconstrained. As a consequence, there might be no solution to eq.3. Intuitively, though eq.3 incorporates the clean distribution into the uncertainty set, it might also include more dirty distributions since the outlier-robust Wasserstein distance tolerates a small fraction of outliers, but the risk on these outliers could be unbounded. Therefore, I'm concerned if the porposed outlier-robust DRO would be biased towards more dirty distributions instead of recovering the clean one.

3. I am also concerned with the selection of the radius $\rho$ of outlier-robust WDRO. As is indicated by Theorem 1, larger $\rho$ leads to higher excess risk while Line 203 states that the excess risk bound only stands for $\rho \geq \rho_0 + W_p(\mu, \hat \mu_n)$, implying that $\rho$ shall not be too small. Since both $\rho_0$ and $W_p(\mu, \hat \mu_n)$ are unavailable in practical settings, the selection of the parameter seems tricky.

**Questions:**

1. Why doesn't the dual form of outlier-robust WDRO in Proposition 7 reduce to that of WDRO when $\epsilon \rightarrow 0$, even if neglecting $\lambda_1$? Specifically, the operator $[\cdot]_+$ claimed to be vital for outlier-robust WDRO does not disappear when outlier-robust WDRO reduce to WDRO.

Some typos:

- Line 158: $\nu$ v.s. $\mu$
- Supplementary Line 565: In second equality, it might be $s_i\geq r_i$ and $s_i \geq 0$.

---

> ### Author Rebuttal · Authors · 2023-08-10
>
> We thank the reviewer for their thoughtful feedback. We address their concerns below:
>
> **Comparison to WDRO with expanded radius:**  We agree that this warrants further discussion. Please see **Common Response 2**.
>
> **Tight performance bound for WDRO with expanded radius with low-dimensional features:** As discussed in **Common Response 2**, the bounds $\mathsf{W}\_1(\hat{\mu},\mu) \lesssim \rho + \sqrt{d\ep}$ and $\mathsf{W}\_1(\check{\mu},\mu) \lesssim \sqrt{d}\rho + \sqrt{d\ep}$, which guarantee that the Wasserstein ball contains the true distribution, are tight in general. Moreover, we can show that the performance of standard WDRO with any arbitrary radius $\rho'$ is tightly characterized by the Wasserstein regularizer from [11] with that radius, even if the reference loss function depends only on $k$-dimensional features. Combining the above claims, if we perform standard WDRO centered around either $\hat{\mu}$ or $\check{\mu}$ and choose the radius $\rho'\in\\{\rho + \sqrt{d\ep},\sqrt{d}\rho + \sqrt{d\ep}\\}$ as above, we cannot recover the excess risk bound of Theorem 3 unless $k = \Omega(d)$.
>
> Further, we are currently working to prove a stronger lower bound that will establish a clear separation between our approach and standard WDRO with expanded radius. Namely, we seek to show that any choice of $\rho'$ around these centers (even if the corresponding Wasserstein ball does not contain $\mu$) will incur suboptimal excess risk. We will report any findings in that direction in the final version.
>
> **Parameter selection:** We agree that this is an important practical concern. Please see **Common Response 1**.
>
> **Impracticality of low-dimensional features:** Section 4 applies to the setting where the loss functions depend only on $k$-dimensional features. Importantly, this is a property of the loss family $\mathcal{L}$, not the data. Although one may seek to match this dimension with an unknown latent dimension of the data, in practice $k$ is often fixed due to tractability or interpretability concerns (e.g., with linear regression, we have $k=1$). In that sense, the parameter $k$ can be viewed as known to the learner.
>
> **Experiments with varied radius:** Thank you for this suggestion. We will add experiments to the final version which vary $\rho$ as well as $\ep$. In the worst case, selecting $\rho$ too small can lead to unbounded risk (see **Common Response 1**), though this may not occur in our simple regression environment.
>
> **Well-definedness of outlier-robust WDRO:** In the proposed example, so long as $\mathcal{A}$ is encoding meaningful moment bounds, $\nu(y')$ does not belong in the uncertainty set for arbitrary $y'$. For example, if $\mathcal{A} = \mathcal{G}\_\mathrm{cov}$, observe that the variance level $\\|\Sigma\_{\nu(y')}\\|\_{\mathrm{op}} \to \infty$ as $\\|y' - y\_0\\| \to \infty$. That is, we only have $\nu(y') \in \mathcal{G}_\mathrm{cov}$ for appropriately small perturbations. Incorporating distributional knowledge of the clean data is essential to obtain meaningful risk bounds (not only in our setting, but throughout robust statistics).
>
> **Radius selection:** Our analysis indeed relies upon $\rho$ being taken sufficiently large. By the argument described in **Parameter selection** above, knowledge of $\rho\_0$ is necessary to obtain meaningful risk bounds. As discussed in Proposition 4, $\mathsf{W}\_p(\mu,\hat{\mu}\_n)$ is bounded by $O(\sqrt{d}n^{-1/d})$ with high probability if $\mu \in \mathcal{G}\_\mathrm{cov}$, so no additional knowledge is necessary.
>
> **Recovering classic WDRO when $\ep = 0$:** As noted by Reviewer zQim, our stated dual form was actually missing an extra Lagrange multiplier (corresponding to the unit mass constraint for probability distributions). The corrected form is provided under **Unit mass constraint** in our response for that reviewer, and it indeed recovers the classic WDRO dual when $\ep = 0$ and $\sigma \to \infty$.
>
> **Typos:** Thanks for noticing these; we will fix them in the final version.

---

> > ### Comment · Reviewer_L4Xq · 2023-08-11
> >
> > Thanks for the authors for their response. My concern over Weakness 2,3 and Question 1 have been addressed. Notably, the authors made a major mistake in the dual reformulation of DRO's objective and the empirical implementation. Fortunately, the authors have corrected both the theoretical and empirical results. My major concern over Weakness 1 around the advantage of outlier-robust WDRO over WDRO remains. The authors provide the excess risk of vanilla WDRO formulated as $\||\ell\|| (\sqrt{d}\rho + \sqrt{d\epsilon})$, in contrast to that of outlier-robust WDRO formulated as  $\||\ell\|| (\rho + \sqrt{d\epsilon})$ in Corollary 1. However, the gap between two DROs is $(\sqrt{d}-1)\rho$ which does not depend on the TV contamination level $\epsilon$. Why does outlier-robust WDRO outperform WDRO when $\epsilon=0$ and only geometric contamination exists? The result does not validate WDRO's superiority in the case of TV contamination. I believe the stronger bound of the excess risk for vanilla WDRO with any radius, which is ongoing work, might elucidate my concern. Since the paper is motivated from WDRO under TV contamination, I insist that a thorough comparison is important. I would like to keep my score for now until further results or clarification is made by the authors.

---

> > > ### Author Response · Authors · 2023-08-11
> > >
> > > When $\varepsilon = 0$, one may center standard WDRO around the observed distribution $\tilde{\mu}$ with radius $\rho$, since a Wasserstein ball of radius $\rho$ about $\tilde{\mu}$ will contain the true distribution $\mu$. In this case, the performance of outlier-robust WDRO and standard WDRO are the same.
> > >
> > > However, as soon as $\varepsilon > 0$ (no matter how small), standard WDRO on its own is no longer sufficient, because we have no bound on $\mathsf{W}_p(\tilde{\mu},\mu)$ (indeed, $\mathsf{W}_p(\mu,(1-\varepsilon)\mu + \varepsilon \delta_z) \to \infty$ as $||z|| \to \infty$). To remedy this efficiently, we proposed above to recenter standard WDRO around the efficiently computable estimate $\check{\mu}$ produced via iterative filtering. However, this estimate can only be guaranteed to satisfy $\mathsf{W}_p(\check{\mu},\mu) \leq \sqrt{d}\rho + \sqrt{d\varepsilon}$, hence the degraded risk bound. We note that this guarantee for $\check{\mu}$ is tight and novel, and in general that no approach using standard WDRO existed for our problem before this work.
> > >
> > > We will update the reviewer if we can prove the mentioned stronger lower bound w/in the discussion period. In any case, we will add this filter + standard WDRO approach as a baseline for comparison to our experiments in the final version.

---

> > > > ### Comment · Reviewer_L4Xq · 2023-08-11
> > > >
> > > > Thanks for the authors’ response. My concern over parameter selection has been addressed and the advantage over WDRO is proven for a remedy of WDRO utilizing iterative filtering. Thus, I would like to raise the score. My remaining concern is over a direct comparison of the excess risk in the general condition where the clean distribution might not be included in the uncertainty set.

---

> > > > > ### Author Response · Authors · 2023-08-15
> > > > >
> > > > > Below we provide a separation between our approach and standard WDRO centered around the iterative filtering estimate (with any radius).
> > > > >
> > > > > Consider the setting where clean data is distributed according to $\mu = \delta\_0$ and the learner observes $\tilde{\mu} = (1-2\varepsilon)\delta\_0 + \varepsilon \mathcal{N}(0,\frac{\rho^2}{\varepsilon^2} e\_1 e\_1^\top) + \varepsilon \mathcal{N}(0,\frac{\rho^2}{2\varepsilon^2}I\_d)$. Here, the second component involves a Wasserstein step of size $\leq \rho$ and the third component is the TV corruption. Iterative filtering seeks to remove an $\varepsilon$ fraction of mass to drive down the operator norm of the covariance matrix, and as a result, the filtered estimate $\check{\mu}$ based on $\tilde{\mu}$ will mostly remove mass from the second component. Based on this guarantee of filtering, we can prove that
> > > > >
> > > > > $$\mathsf{W}\_1(\check{\mu},\mu) \geq \mathbb{E}\_{\check{\mu}}[\\|Z\\|] \gtrsim \varepsilon\, \mathbb{E}\_{\mathcal{N}\big(0,\tfrac{\rho^2}{2\varepsilon^2}I\_d\big)} [\\|Z\\|]\gtrsim \sqrt{d}\rho =: A.$$
> > > > >
> > > > > Now suppose the learner is faced with deciding between $\ell_0(z) =\\|z\\|$ and $\ell_1(z) = A - \\|z\\|$. If they perform standard WDRO with any radius centered around $\check{\mu}$, they will choose $\ell_1$ (since the Lipschitz regularizer resulting from WDRO is identical for these two functions). However, $\mathbb{E}\_\mu[\ell_1] = A \gg \mathbb{E}\_\mu[\ell_0] = 0$, i.e., filtering + WDRO yields an excess risk $\Omega(\sqrt{d}\rho)$.
> > > > >
> > > > > We note that a choice of $\mu$ with unit variance could drive this risk up to $\sqrt{d}\rho + \sqrt{d\varepsilon}$. In any case, this $\sqrt{d}\rho$ component of the risk is highly suboptimal when $d$ is large. This establishes the desired separation between the proposed outlier-robust WDRO approach and standard WDRO with expanded radius. A formal statement of this separation, along with the proof and corresponding discussion, will be added to the revised paper to supplement the current Remark 1.

---

### Official Review · Reviewer_ZiUC · 2023-07-05

**Soundness:** 3 good
**Presentation:** 3 good
**Contribution:** 3 good
**Rating:** 6
**Confidence:** 3

**Summary:**

It is well known that Wasserstein distances do not commute well with total variation distance - a slight perturbation in TV can change the Wasserstein distance by a lot. This means that models that are robust to corruptions in the data distribution in Wasserstein distance can still be vulnerable to outliers or corruptions in TV sense. This paper addresses this problem by considering robustness with respect to both TV and Wasserstein. The paper studies worst-case excess risk w.r.t. corruptions in a new distance termed the 'outlier robust Wasserstein distance'. The authors derive upper and lower bounds on the worst-case excess risk for families of distributions that are either sub-Gaussian or have bounded covariance. The upper bounds depend on a recently introduced term called the "resilience" of a probability measure. Intuitively, resilience of a measure quantifies the deviation in expectation of a function (in this case the loss function) when taken w.r.t. a measure in a probability ball around the original measure. The authors also propose a tractable reformulation of the min-max problem via strong duality. Finally, the authors tighten their results for the case when the true distribution lies on a low dimensional linear subspace, by extending their results to corruptions in outlier robust max-sliced Wasserstein distance. The authors also verify their excess risk bounds on a toy dataset for linear regression problem with mean absolute deviation loss.



**Strengths:**

- significance: the problem of incorporating outlier robustness into the framework of wasserstein distributionally robust optimization (WDRO) is of significance to both ML and optimization communities, and has already received some recent interest. This paper makes progress on this significant problem.
- novelty: I think the upper and lower bounds on the min-max excess risk are novel. The strong duality result appears to be a generalization of Gao and Kleywegt's result on strong duality for WDRO, but I think it is is non-trivial.

**Weaknesses:**

- **Contextualizing the work within related literature**: Although the paper is overall well written, I wish it did a better job at contextualizing their results properly by comparing it with the two special cases of WDRO without TV corruption and of TV robustness without WDRO. For WDRO without TV corruption for example, a natural point of comparison is [this](https://optimization-online.org/wp-content/uploads/2016/04/5396.pdf) paper by Gao and Kleywegt. One point of comparison could be the existence and the form of the worst-case distribution in the min-max risk. This question is answered for the WDRO without TV corruption in Gao and Kleywegt's paper but this paper does not address it at all, which seems odd to me. For TV corruption without WDRO (aka Huber contamination), there are several works from the robust statistics community.
- **Usefulness on top of stricter WDRO**: I think Remark 1 deserves much more discussion. From proposition 1 it is clear that the new compound model of outlier robust WDRO can be subsumed under plain vanilla DRO by increasing the budget of the Wasserstein contamination in proportion to the resilience of the true distribution. Then, how much additional utility do we gain by analyzing the min-max excess risk under the compound model in detail? I must admit I don't understand what the authors mean by the expensive pre-processing step for WDRO. Please explain this to me satisfactorily, and I am willing to change my opinion on this particular weakness.


**Questions:**

- Proposition 3: Is this lower bound valid for p = 2, or only for p = 1? The loss function family is only assumed to be Lipsclitz and not $\alpha$-smooth, whereas for the upper bound, the loss is assumed to be $\alpha$-smooth for the case of p = 2.
- Proposition 4 could just be a remark.
- Something is broken in reference [38]. Are you sure this is the correct reference? I did not find a Proposition 3.4 in it. Also, as I have stated previously, I would very much like a closer comparison of the strong duality result with that of Gao and Kleywegt. Do both the results become identical if one of the lambda's is zero?
- Do the results of section 4 still go through if the max-sliced wasserstein distance is replaced with the sliced wasserstein distance that takes expectation over all slices instead of max?
- Lemma 5, which seems crucial to the proof of Theorem 1, is taken from [12], supposedly a forthcoming paper, which I am unable to access anywhere. So, I am unable to check Theorem 1 fully.

**Limitations:**

Yes.

---

> ### Author Rebuttal · Authors · 2023-08-10
>
> We thank the reviewer for their thoughtful feedback. We address their concerns below:
>
> **Contextualization w/in related work:**
> Thanks for raising this fair point. We will add further discussion that compares our results to those in the literature when either $\varepsilon = 0$ or $\rho = 0$.
>
> For WDRO without TV corruptions (i.e., when $\varepsilon=0$), we will add worst-case distribution results and show how both our strong dual and the said distribution reduce to those of standard WDRO as $\varepsilon\to 0$ and $\sigma\to\infty$. For the dual, after introducing a slight correction to our formulation (see **Unit mass constraint** in response to Reviewer zQim), we indeed recover the classic WDRO dual as a special case in the above limit. For the worst-case distribution, we will derive its existence and structure á la Gao and Kleywegt. Specifically, we can show under the setting of Theorem 2 that $$
>     \sup\_{\substack{\nu \in \cG\_2(\sigma,z\_0):\\\\ \RWp(\tilde{\mu}\_n\\|\nu) \leq \rho}} \E\_\nu[\ell]
>     = \left\\{
>     \begin{array}{cll}
>         \max & \sum\_{(i,j) \in [n]\times [J]} P\_{\ell\_j}(\xi\_{ij}, q\_{ij})  \\\\
>         \mathrm{s.t.} & q\_{ij} \in \R\_+, \xi\_{ij} \in \R^d & \forall i \in [n], \forall j \in [J] \\\\
>         & \xi\_{ij} \in q\_{ij} \cdot \mathcal Z & \forall i \in [n], \forall j \in [J] \\\\
>         & \sum\_{j \in [J]} q\_{ij} \leq \frac{1}{n(1-\varepsilon)} & \forall i \in [n] \\\\
>         & \sum\_{(i,j) \in [n]\times  [J]} q\_{ij} = 1 \\
>         & \sum\_{(i,j) \in [n]\times  [J]} P\_{\| \cdot\|^p} (\xi\_{ij} - q\_{ij} \tilde Z\_i , q\_{ij}) \leq \rho \\\\
>         & \sum\_{(i,j) \in [n]\times  [J]} P\_{\| \cdot\|^2} (\xi\_{ij} - q\_{ij} z\_0 , q\_{ij}) \leq \sigma^2
>     \end{array}
>     \right.
>     $$
> The discrete distribution $\nu^\star = \sum\_{(i,j) \in \mathcal Q} q\_{ij}^\star \delta\_{ \xi\_{ij}^\star / q\_{ij}^\star}$ achieves the worst-case expectation on the left-hand side, where $(q\_{ij}^\star, \xi\_{ij}^\star)_{(i,j) \in [n]\times [J]} $ are optimizers of the maximization problem on the right and $\mathcal Q := \\{(i,j) \in [n] \times [J] : q\_{ij}^\star > 0 \\}$. Note that we recover the classic worst-case distribution when $\ep = 0$ and $\sigma \to \infty$. Moreover, a non-constructive argument based on counting active constraints guarantees that some worst-case distribution exists with support size $n+2$.
>
> For the robust statistics setting without Wasserstein perturbations ($\rho = 0$), we will add comparison to existing work on robust supervised learning (e.g., results of [47] for robust linear regression or those of [45] with a Wasserstein radius of 0). In general, our results improve upon existing risk bounds by scaling with the complexity of the optimal hypothesis for the clean data, rather than requiring a uniform complexity bound for the hypothesis class.
>
> **Usefulness on top of stricter WDRO:** We agree that this warrants further discussion. Please see **Common Response 2**.
>
> **Proposition 3:** The lower bound is valid as stated for any $p \geq 1$ (since the worst-case Wasserstein perturbation we construct is a translation). However, as the bound is in terms of the Lipschitz constant $L$, rather than the Sobolev norm and smoothness constant, it is best viewed as proving the tightness for the $p=1$ component of Theorem 1.
>
> **Proposition 4:** We will convert this statement into a remark.
>
> **Reference 38:** Thanks for pointing this out, some reference numbers were broken in the supplement. The corrected reference should be [36] ("On duality theory of conic linear programs," A. Shapiro).
>
> **Average vs. max-sliced $\mathsf{W}\_p$ in Section 4:** An average-sliced version of $\mathsf{W}\_p$ will not suffice for the Section 4 results; our proof of Lemma 8 in Appendix D requires the Wasserstein bound to hold uniformly over all $k$-dimensional projections since we do not know the relevant map $M \in \R^{k \times d}$ in advance.
>
> **Lemma 5:** This result is from [12], which while not yet officially published in *Operations Research*, is available on the INFORMS website (with DOI 10.1287/opre.2022.2383), and on arXiv with identifier 2009.04382.

---

> > ### Comment · Reviewer_ZiUC · 2023-08-15
> >
> > Thank you for your response. I did not notice the missing constraint in your original Lagrangian formulation, but it is good that this error is fixed and we indeed recover the pure WDRO result as $\epsilon \to 0$. I also appreciate your response to my second weakness.
> >
> > Regarding my comment on your response to reviewer vYVj, could you also perhaps comment on what effect does taking p in Wp to infinity have on your results? I am especially curious to see what happens to the strong duality result as $p\to\infty$. I understand this question may be non-trivial to answer, but my interest in this is stemming from the work of Pydi & Jog Neurips 2021 who showed a connection between $W_\infty$ perturbations and adversarial attacks.

---

### Official Review · Reviewer_zQim · 2023-07-06

**Soundness:** 3 good
**Presentation:** 3 good
**Contribution:** 2 fair
**Rating:** 6
**Confidence:** 3

**Summary:**

This paper introduces an outlier-robust Wasserstein Distributionally Robust Optimization (DRO) framework that aims to capture both geometric uncertainties and non-geometric perturbations, such as adversarial outliers. By utilizing the outlier-robust Wasserstein distance, the proposed framework allows for the arbitrary corruption of a fraction of the data. The authors design an uncertainty set using a robust Wasserstein ball and derive minimax optimal excess risk bounds. They also establish a strong duality for efficient computation. The resulting problem involves tuning three parameters:

- the bounded covariance parameter $\sigma$,

- the radius of the ambiguity set $\rho$,

- and the contamination parameter $\varepsilon$.


Moreover, the authors address dimension dependencies in risk bounds for low-dimensional features by introducing the projection robust optimal transport. The paper concludes with experimental validation of the theory on regression and classification tasks.

**Strengths:**

- The paper replaces the Wasserstein distance with the outlier-robust Wasserstein distance to tackle the case that the observed distribution is contaminated with outliers.
- The authors establish the excess risk bounds of decisions for the cases $p=1,2$.
- The authors derive tractable reformulation by replacing $\mathcal{G}_{\text{cov}}$ and leveraging dual of the problem.
- The low-dimensional features are considered to address the problem of dimension dependency.

**Weaknesses:**

- In Remark 3 and Appendix E, the paper briefly touches on the selection of the parameter $\varepsilon$. However, the overall discussion on parameter selection is limited. The model introduced in the paper involves several parameters, such as $\rho$, $\varepsilon$, and $\sigma$, which may not be fully independent. Therefore, a more comprehensive discussion about the tuning of these parameters is warranted.

- In the experiments, the paper exclusively uses the standard Wasserstein DRO (WDRO) as the baseline for comparison. However, considering that both models aim to address the outlier challenge, it would be valuable to include DFO [A], which is another model specifically designed for handling outliers, as a reasonable baseline for comparison.

- The presence of a few missing references should be addressed in order to enhance the completeness and accuracy of the paper. In Section 4, it would be beneficial to include references [B, C, D], which introduce the concepts of Wasserstein projection pursuit and projection robust Wasserstein, respectively.

[A] Jiang, Nan, and Weijun Xie. "DFO: A Framework for Data-driven Decision-making with Endogenous Outliers." (2022).

[B] Huang, Minhui, Shiqian Ma, and Lifeng Lai. "A riemannian block coordinate descent method for computing the projection robust wasserstein distance." *International Conference on Machine Learning*. PMLR, 2021.

[C] Paty, François-Pierre, and Marco Cuturi. "Subspace robust Wasserstein distances." *International conference on machine learning*. PMLR, 2019.

[D] Niles-Weed, Jonathan, and Philippe Rigollet. "Estimation of wasserstein distances in the spiked transport model." *Bernoulli* 28.4 (2022): 2663-2688.

**Questions:**

- Would it be possible to provide further elaboration on the process of determining suitable parameter values for $\rho$, $\varepsilon$, and $\sigma$?
- In the proof of Proposition 7 (Appendix B.8), is the constraint $\sum_{i\in [n]}m_i = 1$ included to ensure that $\mu^\prime$ is a valid probability distribution? It appears that in the problem mentioned above l566, the constraint should be $0 \leq r_i \leq s_i$ rather than the constraint $0 \leq s_i \leq r_i$.
- In lines 48-49, it appears that the term $W_p^\varepsilon$ is defined in reference to outlier-robust optimal transport, rather than the partial optimal transport as described in [D], which focuses on transporting partial mass from the source distribution to the target distribution.

**Limitations:**

Unless I missed it, I believe the authors do not expand on the limitations of their approach. It would help to add a short section discussing that. This work does not have a negative societal impact.

---

> ### Author Rebuttal · Authors · 2023-08-10
>
> We thank the reviewer for their thoughtful feedback. We address their concerns below:
>
> **Parameter selection:**
> Thank you for this important question. Please see **Common Response 1**. We further provide a proof of our claim therein that knowledge of $\rho$ is necessary for meaningful risk bounds under adversarial Wasserstein perturbations.
>
> Lower bound (Necessity of knowing $\rho$): We construct a family of one-dimensional loss functions $\mathcal{L}$ and an observed distribution $\tilde{\mu} \in \mathcal{P}(\R)$ against which any decision $\hat{\ell} \in \mathcal{L}$ chosen as a function only of $\tilde{\mu}$ must suffer risk
> $$
>  \E_\mu[\hat{\ell}] \gg \inf\_{\ell \in \mathcal{L}} \E\_\mu[\ell] + \mathsf{W}\_1(\tilde{\mu},\mu) \\|\ell\\|\_{\mathrm{Lip}}
> $$
> for some clean distribution $\mu \in \mathcal{P}(\R)$.
>
> Specifically, we consider the family $\mathcal{L} = \{ \ell_\theta: \theta > 0 \}$, where
> $$\ell_\theta(z) := \frac{z}{\theta} + \theta.$$
> Assume that the learner observes $\tilde{\mu} = \delta_0$ and selects decision $\hat{\ell} = \ell_{\hat{\theta}}$ for $\hat{\theta} > 0$. If the true distribution was $\mu = \delta_\rho$, then the optimal decision would have been $\theta_\star = \sqrt{\rho}$. In this case, the learner suffers excess risk
> $$
>     \ell_{\hat{\theta}}(\rho) - \ell_{\theta_\star}(\rho) = \frac{\rho}{\hat{\theta}} + \hat{\theta} - 2\sqrt{\rho} = \left(\sqrt{\rho/\hat{\theta}} - \sqrt{\hat{\theta}}\right)^2.
> $$
> As $\rho \to \infty$, this far exceeds the desired lower bound of $ \mathsf{W}\_1(\tilde{\mu},\mu) \cdot\\|\ell\_{\theta\_\star(\rho)}\\|\_{\mathrm{Lip}} = \rho \cdot \rho^{-1/2} = \sqrt{\rho}$. This construction fails when $\rho$ is known, since the learner may simply select $\hat{\theta} = \sqrt{\rho}$.
>
> **Comparison to [A]:** Thank you for this reference. We will include [A] in our related work and add it as a baseline for comparison. We would like to emphasize, however, that the DFO approach requires solving a non-convex optimization problem, significantly impacting its scalability. Further, this method is not accompanied by any proof of minimax optimality.
>
> **Missing references:** We will update our related work and discussion to include these citations. We note that the notion of robustness considered in these papers is distinct from ours  (although we do employ such sliced distances in our analysis for Section 4).
>
> **Unit mass constraint:** We thank the reviewer for raising this important point. Our tractable dual reformulation is indeed missing a Lagrange multiplier corresponding to the constraint that probability distributions have unit mass. The corrected dual is
> $$
> \sup\_{\nu \in \cG\_2(\sigma,z\_0):\\, \mathsf{W}\_p^\ep(\tilde{\mu}\_n\\|\nu) \leq \rho} \E\_\nu[\ell] = \inf\_{\substack{\lambda\_1, \lambda\_2 \in \R\_+ \\\\ \alpha \in \R}} \lambda\_1 \sigma^2 + \lambda_2 \rho^p + \alpha + \frac{1}{1-\ep} \E\_{\tilde{\mu}\_n} \big[\\,\overline{\ell}(\cdot\,;\lambda\_1,\lambda\_2, \alpha)  \big],
> $$
> where $\overline{\ell}(z;\lambda\_1,\lambda\_2, \alpha) := \sup\_{\xi \in \mathcal{Z}} \\,\big[\\,\ell(\xi) - \lambda\_1 \| \xi - z\_0 \|^2 - \lambda\_2 \| \xi - z \|^p - \alpha \big]\_+$.
> Recalling the conditional variance at risk defined by $\mathrm{CVaR}\_{1-\ep, \mu} [\ell(Z)] = \inf\_{\alpha \in \R} \alpha + \frac{1}{1 - \ep} \E\_{Z \sim \mu} \big[[\ell(Z) - \alpha]\_+\big]$, this can be restated as
> $$
> \inf\_{\lambda\_1, \lambda\_2 \in \R\_+ } \lambda\_1 \sigma^2 + \lambda\_2 \rho^p +
> \mathrm{CVaR}\_{1-\ep, \tilde{\mu}\_n} \left[ \sup\_{\xi \in \mathcal{Z}} \\, \ell(\xi) - \lambda\_1
> \| \xi - z\_0 \|^2 - \lambda\_2 \| \xi - Z \|^p \right].
> $$
> When $\ep \to 0$ and $\sigma \to \infty$, CVaR reduces to the standard expected value and the minimizing value for $\lambda_1$ is 0; we thus recover the classic WDRO dual as a special case.
>
> In the new Figure 2, we display experiments updated with this correction. Results are essentially unchanged, with the exception of reduced performance when $\ep$ is too small. Indeed, the corrected dual form invalidates Remark 3 (it relies on an insensitivity of the incorrect dual to $\ep$ that is no longer the case). We will replace this with a remark on parameter selection as described in **Common Response 1**. We are currently investigating theoretical justification for the strong performance without the constraint and will include our findings in the final version.
>
> \textbf{Outlier-robust vs. partial OT:} We note that our definition of outlier-robust OT $\RWp$ corresponds to a partial OT distance up to a constant prefactor:
> $$
> \RWp(\mu,\nu)^p = (1-\ep)^{-1}\inf\_{\substack{\pi \in \Pi(\mu,\nu)\\\\\pi\_1 \leq \mu, \pi_2 \leq \nu\\\\\pi(\mathcal{Z} \times \mathcal{Z}) = 1-\ep}} \int \|x - y\|^p d \pi(x,y).
> $$
>
> **Limitations:** Thank you for this suggestion. We will add a discussion of limitations before the conclusion. In particular, we will reiterate the required knowledge of problem parameters and discuss the practical implications of Assumption 2 (see response to Reviewer vYVj).

---

> > ### Comment · Reviewer_zQim · 2023-08-16
> > **Response to authors**
> >
> > Thank you for providing clarifications that address my concerns. I'm pleased to see the correct dual problem formulation. After carefully reading all the reviews and rebuttals, I have decided to revise my score from 4 to 6.

---

### Official Review · Reviewer_vYVj · 2023-07-07

**Soundness:** 3 good
**Presentation:** 3 good
**Contribution:** 3 good
**Rating:** 6
**Confidence:** 3

**Summary:**

This paper introduces a novel approach to make Wasserstein distributionally robust optimization problems robust to adversarial outliers including geometric perturbations and non-geometric contamination of the data. This goal is achieved by considering relevant Wasserstein ball which includes both type of adversarial attacks. Later, the authors provides strong duality results which allows them to improve the computation complexity of the proposed outlier-robust WDRO problem.

**Strengths:**

This is a well written paper with novel contributions. The robustness of WDRO to the outliers is a very natural problem and the authors proposed a Wasserstein distance based constraints to provide robustness. This is particularly hard if we do not make any presumptions on the distribution of data; therefore, I find the contributions of this paper to be significant to the literature. There might be some concerns that I have mentioned in the weaknesses part, but, overall I am impressed by the work done here.


**Weaknesses:**

As I have mentioned at the strengths part, I believe this paper has a novel contribution to the robust optimization literature. Some few weaknesses that I observed was that authors assumed the readers have prior knowledge on relation between adverserial attacks and Wasserstein and TV perturbations. These are not so obvious for me and might be useful to include some discussion/motivation about how these attacks can be associated with claimed perturbations in metric space.

I believe it would strengthen the paper if authors can obtain some results on larger datasets. Considering the time constraint of the conference, I believe it would be sufficient to add one large dataset to experiments.


**Questions:**

- What is the definition for $\theta$ performing uniformly well over Wasserstein ball (stated in Line 29)?
- I could not see the definition of $\Sigma_\mu$ in the paper.

**Limitations:**

The Assumption 2 can be challenging to maintain/check in practice. Moreover, this application might also be hard to implement on large datasets such as classification problems on MNIST or FMNIST. That being said, I don't see these possible limitations as a major problem considering the theoretical novelty of this work.

---

> ### Author Rebuttal · Authors · 2023-08-10
>
> We thank the reviewer for their thoughtful feedback. We address their concerns below:
>
> **Relation between adversarial attacks and Wasserstein/TV perturbations:** Thank you for bringing this up. We will add the new Figure 1, along with accompanying discussion, to the introduction to clarify the nature of our allowed perturbations. Intuitively, the perturbation set $\mathcal{B}_{\varepsilon,\rho}(\mu) := \\{\tilde{\mu} : \mathsf{W}_p^\varepsilon(\tilde{\mu},\mu) \leq \rho\\}$ describes the following adversary. First, the $\mathsf{W}_p$ perturbation of radius $\rho$ enables the adversary to geometrically move samples of $\mu$ around for a total $L^p$ displacement cost of $\rho$, to arrive at a new distribution $\mu'$. The $\varepsilon$-TV perturbation further enables mixing $\mu'$ with an arbitrary outlier mass $\alpha$ to generate the distribution $\tilde\mu=(1-\varepsilon)\mu'+\varepsilon\alpha$. More formally, the adversary replaces each sample $X \sim \mu$ with $X + \Delta(X)$, for some (potentially stochastic) displacement map $\Delta : \mathbb{R}^d \to \mathbb{R}^d$ such that $\mathbb{E}[\|\Delta(X)\|^p | \mathcal{E}] \leq \rho^p$, where $\mathcal{E}$ is some event with probability $1-\varepsilon$.
>
> **Results on larger datasets:** Our approach can indeed scale to more complex data sets. As a proof of concept, we compare standard and outlier-robust WDRO (implemented via Theorem 2) for binary classification between two MNIST digit categories with 10\% of training labels flipped, see the new Figure 3.
> Training linear classifiers with hinge loss on 10, 20, 50, and 100 training digits (each naively reduced to 50 dimensions), we found that our approach consistently outperformed standard WDRO in classification accuracy. For the final version, we will extend this test to larger dimensions and training set sizes, and include the results in our Section 5.
>
> **Uniform performance over Wasserstein ball:** On Line 29, the sentence "$\hat{\theta} \in \Theta$ performs uniformly well over the Wasserstein ball" was intended to be a qualitative description of Eq. 1. Indeed, the minimizer $\hat{\theta}$ is selected to incur low risk $\mathbb{E}_{Z \sim \nu}[\ell(\theta,Z)]$ w.r.t. any distribution $\nu$ lying within Wasserstein distance $\rho$ of the observed data distribution $\tilde{\mu}$. We will adjust the phrasing to emphasize that we are optimizing for such a uniform risk bound, rather than suggesting that the performance of $\hat{\theta}$ is similar for all such $\nu$ (assuming this was the cause for confusion).
>
> **Definition of $\Sigma_\mu$:** $\Sigma_\mu \in \R^{d \times d}$ denotes the covariance matrix of a probability measure $\mu \in \mathcal{P}(\R^d)$. We will include this definition in the final version.
>
> **Strength of Assumption 2:** Generally, any continuous function can be approximated arbitrarily well by a maximum of finitely many concave functions. However, the number of needed functions can be arbitrarily large, which raises efficiency concerns in practice.
> For instance, the $\ell\_1$-norm $\|z\|\_1 = \max_{\sigma \in \{\pm 1\}^d} \sum\_{i=1}^d \sigma\_i z\_i$ requires $2^d$ concave functions, whereas the $\ell\_\infty$-norm $\|z\|\_\infty = \max_{i \in [d], \sigma \in \{\pm 1\}} \sigma z\_i$ requires only $2d$. We will add a discussion of this limitation to the final version. The question of how to efficiently perform outlier-robust WDRO for loss functions requiring $\exp(d)$ concave pieces is an interesting avenue for future research. In such cases, it may be appropriate to apply gradient methods directly to our dual form (Eq. 6).

---

> > ### Comment · Reviewer_vYVj · 2023-08-11
> >
> > I have read authors rebuttal and I believe their arguments are justified. I thank for authors for their work and would like to keep my rating as it is.

---

> > > ### Author Response · Authors · 2023-08-11
> > >
> > > Thank you for your kind response. We were wondering if there are any other additions to the text that the reviewer would like to see that could further improve their assessment of the work?

---

> > > > ### Comment · Reviewer_vYVj · 2023-08-20
> > > >
> > > > I just realized that I did not respond to this comment and I apologize. The main reason I would like to keep my rating is due the fact that I believe there is more room for improvement on the experiments. In the context of outlier detection, I believe there could have been more done with MNIST data such as various digits (more than two) as outlier and playing with the training flip rate. Moreover, another alternative is to use two or more digits as normal and picking one as outlier could also be used for comparison as well. I am very satisfied with the discussion so far and it looked like most of the reviewer also provide similar ratings. So, I would still like to keep mine if that would not majorly effect area chairs final decision.

---

> > ### Comment · Reviewer_ZiUC · 2023-08-15
> > **There is a more precise relation between adversarial attacks and Wasserstein perturbations.**
> >
> > Upon reading the review of vYVj and your response, I would like to point out that there is a more precise relation between adversarial attacks and Wasserstein perturbations besides the intuition provided in your response. Reference [*] establishes the equivalence between robustness against adversarial attacks with perturbation radius $\epsilon$ and distributional robustness in a ball of radius $\epsilon$ w.r.t. $W_\infty$ metric. I wonder if your strong duality result holds as $p\to \infty$.
> >
> > [*] Pydi, M. S., & Jog, V. (2021). The many faces of adversarial risk. Advances in Neural Information Processing Systems, 34, 10000-10012.

---

> > > ### Author Response · Authors · 2023-08-16
> > >
> > > Indeed, there is an equivalence between point-wise adversarial attacks and $\mathsf{W}\_\infty$ perturbations for standard WDRO. Fix observed data $\tilde{\mu}\_n = \frac{1}{n} \sum\_{i=1}^n \delta\_{\tilde{z}\_i}$, and write $\mathcal{B}\_\infty := \\{ \nu \in \mathcal{P}(\mathcal{Z}) : \mathsf{W}\_\infty(\nu,\tilde{\mu}\_n) \leq \rho \\}$. By Lemma EC2 of [A], $\mathcal{B}\_\infty$ admits the equivalent representation
> > > $$
> > > \mathcal{B}\_\infty = \left\\{\ \frac{1}{n}\sum\_{i=1}^n \delta\_{z\_i} : \\|z\_i - \tilde{z}\_i\\| \leq \rho, z_i \in \mathcal{Z} \right\\},
> > > $$
> > > and we have
> > > $$
> > > \sup\_{\nu \in \mathcal{B}\_\infty} \mathbb{E}\_\nu[\ell(Z)] = \mathbb{E}\_{\tilde{\mu}\_n}[\bar{\ell}(Z)],
> > > $$
> > > where $\bar{\ell}(z) := \sup\_{z' \in \mathcal{Z}, \\|z' - z\\| \leq \rho} \ell(z')$.
> > >
> > > Our theory also extends naturally to this $p \to \infty$ limit. For the robust Wasserstein ball $\mathcal{B}\_\infty^\varepsilon := \\{ \nu \in \mathcal{Z} : \mathsf{W}\_\infty^\varepsilon(\nu,\tilde{\mu}\_n) \leq \rho \\}$, we can similarly prove
> > > $$
> > > \sup\_{\nu \in \mathcal{B}\_\infty^\varepsilon} \mathbb{E}\_\nu[\ell(Z)] = \mathrm{CVaR}\_{\tilde{\mu}\_n}^{1-\varepsilon}[\bar{\ell}(Z)],
> > > $$
> > > where $\mathrm{CVaR}$ is the conditional variance at risk appearing in our corrected dual.
> > >
> > > Enforcing our moment constraints, we can prove
> > > $$
> > > \sup\_{\nu \in \mathcal{B}\_\infty^\varepsilon \cap \mathcal{G}\_2(\sigma,z\_0)} \mathbb{E}\_\nu[\ell(Z)] = \inf\_{\lambda \geq 0} \lambda \sigma^2 + \mathrm{CVaR}\_{\tilde{\mu}\_n}^{1-\varepsilon}[\bar{\ell}\_2(Z)],
> > > $$
> > > where $\bar{\ell}\_2(z) := \sup\_{z' \in \mathcal{Z} : \\|z' - z\\| \leq \rho} \ell(z') - \lambda\\|z' - z\_0\\|^2$, and
> > > $$
> > > \sup\_{\nu \in \mathcal{B}\_\infty^\varepsilon \cap \mathcal{G}\_\mathrm{cov}(\sigma,z\_0)} \mathbb{E}\_\nu[\ell(Z)] = \inf\_{\Lambda \succeq 0} z\_0^\top \Lambda z\_0 + \sigma^2 \mathrm{Tr}(\Lambda) + \mathrm{CVaR}\_{\tilde{\mu}\_n}^{1-\varepsilon}[\bar{\ell}\_\mathrm{cov}(Z)],
> > > $$
> > > where $\bar{\ell}\_\mathrm{cov}(z) := \sup\_{z' \in \mathcal{Z} : \\|z' - z\\| \leq \rho} \ell(z') - z'^\top \Lambda z'$.
> > >
> > > Note that both cases recover the standard dual as $\varepsilon \to 0$ and $\sigma \to \infty$.
> > >
> > >
> > > [A] : Rui Gao, Xi Chen, and Anton J. Kleywegt. Wasserstein Distributionally Robust Optimization and Variation Regularization. Operations Research, 2022, to be published. doi:10.1287/opre.2022.2383

---

### Author Rebuttal · Authors · 2023-08-10

$\newcommand{\cG}{\mathcal{G}}\newcommand{\ep}{\varepsilon}\newcommand{\RWp}{\mathsf{W}_p^\ep}\newcommand{\E}{\mathbb{E}}\newcommand{\R}{\mathbb{R}}\newcommand{\sg}{\sigma}\newcommand{\mr}{\mathrm}$**Common Response:**

We thank the reviewers for their time and feedback. Below we provide a common response to shared concerns.

**1. Parameter selection:**
Selection of $\rho$, $\sg$, and $\ep$ appearing in Eq. 8 and its tractable dual reformulation is a key practical consideration. First, we observe that knowledge of valid upper bounds on these parameters is sufficient to attain excess risk bounds scaling in terms of said upper bounds. This approach avoids meticulously tuning the parameters but may result in suboptimal risk. Below, we discuss methods for parameter selection that attain optimal minimax risk.

First, we note that knowledge of $\rho$ is necessary to attain meaningful risk bounds against adversarial Wasserstein perturbations, even for standard WDRO (i.e., $\ep = 0$, $\sg \to \infty$). See the beginning of our response to Reviewer zQim for a proof of this claim. In the popular setting where $\rho$ models only sampling error (i.e., $\rho_0 = 0$ in the setting of Section 3.2), without adversarial perturbations, knowledge of one of $\sg$ or $\ep$ is enough to tune $\rho$. If $\sg$ is known, then one may select $\rho = O(\sg n^{-1/d})$, as discussed in Proposition 4 (where $\sg \leq \sqrt{d}$ by the covariance bound). If $\sg$ is unknown but $\ep$ is known, we can show that the bootstrapped estimate $\hat{\rho} = 2\mathsf{W}\_p^{2\ep}(\tilde{\mu}\_{S}, \tilde{\mu}\_{[n] \setminus S})$ gives near optimal guarantees, where $S$ is a uniformly random $n/2$-sized subset of $[n]$.

Next, we address tuning $\sg$ and $\ep$, assuming henceforth that $\rho$ is known or was tuned as described above, based on knowledge of one of them. In general, knowing at least one of $\sg$ or $\ep$ is necessary to obtain non-trivial risk bounds, even in the easier problem without Wasserstein perturbations (i.e. $\rho = 0$). Otherwise, it is information theoretically impossible to meaningfully distinguish inliers from outliers (see Exercise 1.7b of "Algorithmic High-Dimensional Robust Statistics" by Diakonikolas and Kane, 2022, for a discussion of this issue in the setting of robust mean estimation).

The first step of our approach is to obtain an accurate robust mean estimate, which we discuss next. If $\ep$ is known but $\sg$ is unknown, then the mean estimation algorithms we employ are fully specified and we may proceed with outlier-robust WDRO. In the opposite case, when $\sg$ is known but $\ep$ is unknown, we employ a halving trick, dividing a guess $\hat{\ep}$ by two until an appropriate $\hat{\ep}$-trimmed variance of the observed data is $O(\sg)$.

Having an accurate robust mean estimate, to run the main robust WDRO procedure, we can learn the missing parameter via an analogous halving trick. Namely, we use a binary search to set the missing parameter as small as possible, such that the primal problem (Eq. 5) is feasible (corresponding to the tractable dual (Eq. 6) being bounded). The number of search steps scales logarithmically with the ratio of the initial guess to the true value.

We will add a remark and a Supplement section to expand on the above in the final version.

**2. Comparison to WDRO w/ expanded radius:**
We agree that the discussion in Remark 1 should be significantly expanded, which we will do in the revision to account for the points below. As mentioned in the remark, the main issue with running vanilla WDRO with an expanded radius is that the Wasserstein ball must be centered around the output of a complicated minimum distance estimation (MDE) procedure, namely $\hat{\mu} = \mr{argmin}_{\nu \in \cG} \mathsf{W}_1^\ep(\tilde{\mu} \| \nu)$. While [28] provides a statistical analysis for this estimate, the procedure they propose for finite-sample computation is a heuristic one based on Wasserstein GANs, which lacks formal guarantees.

An efficient alternative to the MDE-based approach is possible for the class $\cG\_\mr{cov}$ using the popular iterative filtering method [8], but the resulting (sharp) risk bounds are suboptimal. Specifically, we can prove that iterative filtering returns a distribution $\check{\mu}$ with $\mathsf{W}\_1(\check{\mu},\mu) \lesssim \sqrt{d}\rho + \sqrt{d\ep}$ (and that this guarantee is tight). Performing vanilla WDRO around $\check{\mu}$ with radius $\sqrt{d}\rho + \sqrt{d\ep}$ yields excess risk $\\|\ell\\|\_\mr{Lip} (\sqrt{d}\rho + \sqrt{d\ep})$. This bound is suboptimal in dependence on $\rho$ by a $\sqrt{d}$ factor (compare to our Corollary 1), which becomes prohibitive as dimension grows. We are working to prove a stronger lower bound, showing that vanilla WDRO around $\check{\mu}$ with any radius leads to suboptimal excess risk, and will report any findings in the final version.

Even if we ignore the tractability of finding $\hat{\mu}$, we are unaware of any theory which would allow matching our risk bounds from Section 4 for $k$-dimensional features using standard WDRO. Although the MDE estimate satisfies the stronger (and also tight) approximation guarantee $\mathsf{W}\_1(\hat{\mu},\mu) \lesssim \rho + \sqrt{d\ep}$, running standard WDRO with this radius leads to the same risk bounds from Corollary 3, but with $\sqrt{d\ep}$ instead of $\sqrt{k\ep}$. Again, as $k\ll d$, this results in a significant worsening of the bound.

Lastly, although somewhat subjective, our approach can be viewed as more holistic and interpretable than WDRO with expanded radius. The outlier-robust Wasserstein distance is tailored to account for the considered adversarial model, the strong duality clearly reveals the effect of the different parameters, and the characterization of worst-case distribution (see **Contextualization w/in related work** in response to Reviewer ZiUC) further elucidates the problem structure.

---

### Decision · Program_Chairs · 2023-09-21

**Decision:**

Accept (poster)

**Comment:**

The collective feedback from the reviews is positive, underscoring the significance of the problem tackled, commendable theoretical insights, and the clarity of the writing. The authors' rebuttal seems to have aptly addressed prevailing methodological concerns, including matters of parameter selection, an error in the dual formula, and comparisons with existing literature. The inclusion of new numerical results during the rebuttal phase is a praiseworthy addition. I recommend acceptance.